# A Gang of Adversarial Bandits

**Mark Herbster\*, Stephen Pasteris\***
Department of Computer Science
University College London
London WC1E 6BT
{m.herbster,s.pasteris}@cs.ucl.ac.uk

**Fabio Vitale**
University of Lille
59653 Villeneuve d'Ascq CEDEX
France
fabio.vitale2@univ-lille.fr

**Massimiliano Pontil**
CSML, Istituto Italiano di Tecnologia and
Department of Computer Science
University College London
massimiliano.pontil@iit.it

## Abstract

We consider running multiple instances of multi-armed bandit (MAB) problems in parallel. A main motivation for this study are online recommendation systems, in which each of $N$ users is associated with a MAB problem and the goal is to exploit users' similarity in order to learn users' preferences to $K$ items more efficiently. We consider the adversarial MAB setting, whereby an adversary is free to choose which user and which loss to present to the learner during the learning process. Users are in a social network and the learner is aided by a-priori knowledge of the strengths of the social links between all pairs of users. It is assumed that if the social link between two users is strong then they tend to share the same action. The regret is measured relative to an arbitrary function which maps users to actions. The smoothness of the function is captured by a resistance-based dispersion measure $\Psi$. We present two learning algorithms, GABA-I and GABA-II which exploit the network structure to bias towards functions of low $\Psi$ values. We show that GABA-I has an expected regret bound of $\mathcal{O}(\sqrt{\ln(NK/\Psi)\Psi KT})$ and per-trial time complexity of $\mathcal{O}(K\ln(N))$, whilst GABA-II has a weaker $\mathcal{O}(\sqrt{\ln(N/\Psi)\ln(NK/\Psi)\Psi KT})$ regret, but a better $\mathcal{O}(\ln(K)\ln(N))$ per-trial time complexity. We highlight improvements of both algorithms over running independent standard MABs across users.

## 1 Introduction

During the last decade multi-armed bandits (MAB) have received a great deal of attention in machine learning and related fields, due to their wide practical and theoretical importance. The central problem is to design a decision strategy whereby a learner explores sequentially the environment in order to find the best item (arm) within a prescribed set. At each step in the exploration the learner chooses an arm, after which feedback (typically a loss or reward corresponding to the selected arm) is observed from the environment. Then the next decision is made by the learner based on past interactions, and the process repeats. The goal is to design efficient exploration strategies which incur a small cumulative loss in comparison to the cumulative loss that would have been obtained by always

---

\* Equal contribution.

35th Conference on Neural Information Processing Systems (NeurIPS 2021).

selecting the best arm in hindsight. Applications of MAB are numerous, including recommender systems [1], clinical trials [2], and adaptive routing [3], among others.

In this paper we study the problem in which the learner is facing several MAB problems that are related according to a prescribed interaction graph. A main motivation behind this problem are online recommendation systems, whereby each of several users is associated with a MAB problem (task), where the arms correspond to a finite set of products, and the graph represents a social network among users. The goal is to exploit users' similarity in order to improve the efficiency of learning users' preferences via online exploration of products. In the standard full information setting, a lot of work has been done showing that techniques from multitask learning are effective in reducing the amount of data needed to learn each of the individual tasks, both in the statistical and adversarial settings, see [4, 5, 6, 7, 8, 9, 10, 11] and references therein. Graphs have been used to model task relationships, with different tasks' parameters encouraged to be close according to the graph topology. In contrast, multitask learning in the bandit setting is much less explored.

The algorithms that we present exploit the *network homophily* principle [12] which formulates that users that are connected in the network have similar preferences, that is, they tend to share preferred recommendations. We will show that our algorithms exploit graph structure and enjoy potentially much smaller regret bounds than the cumulative regret of standard MAB run independently on each user. Since the original graph may be dense, we exploit a randomized sparsification technique to develop fast prediction algorithms. Our approach builds upon previous work on online learning over graphs [13, 14] to generate a perfect full oriented binary tree, whose leaves are in one-to-one correspondence with the nodes of the original graph. This construction approximately preserves the relevant graph properties in expectation, and provides the starting point for designing our efficient algorithms. A further ingredient in our algorithm is provided by the method of *specialists* [15, 16]. Our learning strategies combine the above ingredients to devise efficient online algorithms under partial feedback.

**Contributions.** We introduce two Gang of Adversarial BAndit algorithms, GABA-I and GABA-II that learn jointly MAB models for $N$ users over $K$ possible actions. Both algorithms are designed to exploit network structure while being (extremely) computationally efficient. We derive expected (over the algorithms' randomizations) regret bounds. The bounds scale with the *dispersion* measure $\Psi \in [1, N]$ of the best actions over the graph. For GABA-I the bound[1] is of order of $\mathcal{O}(\sqrt{\ln(NK/\Psi)\Psi K T})$, where $T$ is the number of trials, and has a per-trial time complexity of $\mathcal{O}(K \ln(N))$. On the other hand GABA-II has a weaker expected regret bound of $\mathcal{O}(\sqrt{\ln(N/\Psi)\ln(NK/\Psi)\Psi K T})$ but is faster, having a per-trial time complexity of $\mathcal{O}(\ln(K)\ln(N))$. Thus the GABA-I algorithm improves on algorithms that treat each user independently, as in the best case the regret improves from $\mathcal{O}(\sqrt{N})$ to $\mathcal{O}(\sqrt{\ln N})$ and in the worst case the regret degrades by at most a constant factor. GABA-II has slightly weaker regret bounds; however, it is more computationally efficient.

**Outline of Main Results.** The social network graph $\mathcal{G}$ is determined by a set of undirected links between users $\{\omega_{u,v}\}_{u<v}^N$ where $\omega_{u,v} \in [0, \infty)$ indicates the magnitude of the link between user $u$ and $v$. For all $t \in [T]$ we have a user $u_t \in [N]$ and a loss vector $\boldsymbol{\ell}_t \in [0, 1]^K$ which are selected by *Nature* before learning begins and are unknown to *Learner*; i.e., Nature is a *deterministic oblivious adversary* (see e.g., [17, Section 5.1]). Learning then proceeds in trials $t = 1, 2, \ldots, T$. On trial $t$:

1. Nature reveals user $u_t \in [N]$ to Learner,
2. Learner selects action $a_t \in [K]$,
3. Nature reveals loss $\ell_{t,a_t} \in [0, 1]$ to Learner.

Before reflecting on the $N$-user case we review the well-known results for the single user ($N = 1$). The seminal EXP3 algorithm [18] obtains the following *(uniform)* regret bound[2],

$$\mathbb{E}\Big[ \sum_{t \in [T]} \ell_{t,a_t} \Big] - \min_{a \in [K]} \sum_{t \in [T]} \ell_{t,a} \in \mathcal{O}\left( \sqrt{K \ln(K) T} \right), \tag{1}$$

---

[1]The bounds of GABA-I and GABA-II however depend on oracular knowledge of optimal tuning parameters. We discuss this as well as a means of lessening this dependency following Corollary 5.

[2]An algorithm was given in [19] that removed the $\ln K$ term from the regret.

where the expectation is with respect to the internal randomization of the EXP3 algorithm. In the $N$-user setting, by running a copy of EXP3 independently for each user, we may obtain a uniform regret bound of (see e.g., [20])

$$\mathbb{E}\Big[\sum_{t\in[T]}\ell_{t,a_t}\Big] - \min_{y:[N]\to[K]}\sum_{t\in[T]}\ell_{t,y(u_t)} \in \mathcal{O}\left(\sqrt{K\ln(K)NT}\right), \qquad (2)$$

i.e., for each user $u$ the best action is $y(u)$ and we now pay an additional constant factor of $\sqrt{N}$ in our regret. In this work we exploit the social network structure to prove a *non-uniform* regret bound for the GABA-I algorithm (see Corollary 4) of

$$R(y) := \mathbb{E}\Big[\sum_{t\in[T]}\ell_{t,a_t}\Big] - \sum_{t\in[T]}\ell_{t,y(u_t)} \in \mathcal{O}\left(\sqrt{K\ln\left(\frac{KN}{\Psi(y)}\right)\Psi(y)T}\right), \qquad (3)$$

for any mapping of users to actions $y : [N] \to [K]$. The non-uniform regret now depends on $\Psi(y) \in [1, N]$ (see (5)) which measures dispersion of users' 'best' actions across the network. Thus, by taking network structure into account, we may upper bound the scaling in the regret with respect to the number of users by $\mathcal{O}(\sqrt{\ln(\frac{eN}{\Psi(y)})\Psi(y)})$. When the best action across the network is nearly uniform then the dispersion $\Psi(y) \in \mathcal{O}(1)$, in contrast when the dispersion is maximal then $\Psi(y) = N$ thus in the best case the regret with respect to the number of users improves from $\mathcal{O}(\sqrt{N})$ to $\mathcal{O}(\sqrt{\ln N})$ and in the worst case the regret only increases by a constant factor. The first algorithm GABA-I obtains the regret (3) while requiring $\mathcal{O}(K\ln N)$ time to predict and update. The second algorithm GABA-II's regret (see Corollary 5) is larger by a $\mathcal{O}(\sqrt{\ln N/\Psi(y)})$ factor but now prediction is at an even faster $\mathcal{O}(\ln(K)\ln(N))$ time per trial, that is, prediction time improves exponentially with respect to the cardinality of the action set $[K]$. Thus both algorithms support very large user networks and the second algorithm allows efficient prediction with very large action sets.

**Related Work.** We mention here some of the key papers that are more closely related to ours and refer the reader to the technical appendices for an expanded literature review. There has been much work in the heterogenous multi-user setting for linear-stochastic bandits. Out of these works, those closest to us are when the users are in a known social network and it is assumed that neighbouring users respond to context vectors in a similar way [21, 22, 23, 24, 25, 26] but as far as we are aware no works on this model have so far been done in the adversarial setting. Other works on this topic include those in which it is assumed that there is an unknown clustering of the users, where users in the same cluster are homogenous [27, 28, 29, 30, 31, 32, 33, 34, 35]; as well as other models [36, 37, 38, 39, 40, 41, 42, 43]. There are also works on networked, homogenous multi-user bandit models with limited communication between users [44, 45, 46, 47, 48, 49, 50]. Related to the multi-user setting are works on transfer learning and meta-learning with linear-stochastic bandits [51, 52]. Whilst our work assumes a known network structure over the users, there is a wide literature on bandit problems in which the actions are structured in a network and it is assumed that neighbouring actions give similar losses [53, 54, 55, 56, 57, 58, 59, 60], as well as other networked-action models [61, 62, 63]. In addition to the seminal paper on adversarial bandits [18], our work utilises ideas from several different papers [13, 14, 15, 16, 64, 65].

**Notation.** Given a set $X$ we define $2^X$ to be its power-set, that is: the set of all subsets of $X$. For any positive integer $m$, we define $[m] := \{1, 2, \ldots, m\}$. For any predicate PRED , $[\![\text{PRED}]\!] := 1$ if PRED is true and equals 0 otherwise. Given vectors $\boldsymbol{x}, \boldsymbol{x}' \in \mathbb{R}^K$ we define $\boldsymbol{x} \cdot \boldsymbol{x}'$ to be their inner product (i.e., equal to $\sum_{i\in[K]} x_i x_{i'}$) and we define $\boldsymbol{x} \odot \boldsymbol{x}'$ to be their component-wise product (i.e., $(x \odot x')_i := x_i x'_i$ for all $i \in [K]$). We define '$\mathbf{1}$' to be the $K$-dimensional vector in which each component is equal to $1$. Given a full oriented binary tree $\mathcal{B}$ we denote the set of its vertices also by $\mathcal{B}$. Given a non-leaf vertex $n \in \mathcal{B}$ let $\triangleleft(n)$ and $\triangleright(n)$ be its left child and right child respectively. Given a non-root vertex $n \in \mathcal{B}$ let $\uparrow(n)$ be its parent. Given a vertex $n \in \mathcal{B}$ let $\Uparrow(n)$ and $\Downarrow(n)$ be its set of ancestors and leaf-descendants (i.e. descendants which are leaves) respectively. Given a vertex $n \in \mathcal{B}$ we define $\blacktriangleleft(n)$ and $\blacktriangleright(n)$ as the left-most and right-most descendants (which are leaves) of $n$ respectively. Finally, we denote the user graph by $\mathcal{G}$, which is an undirected connected graph with edge weights $\{\omega_{u,v} : 1 \le u < v \le N\}$. For convenience we assume $N$ is a power of two.[3]

---

[3]This assumption does not limit our results because to run our algorithms one can always add dummy vertices without altering input weights, so as to force $N$ to be a power of two.

## 2    Modeling a Social Network as a Resistive Network

In this section we introduce the tools necessary to formalize our complexity measures, as well as the ones to implement our algorithms.

### 2.1    Conceptual Tools

To minimize the incurred loss, Learner can exploit the similarity between any pair of users defined by the weights $\omega_{u,v}$ of user graph edges for all $u, v \in [N]$. The function $y : [N] \to [K]$ is completely unknown to Learner, and can be viewed as labeling each user with its best/favorite action. Within this context, our homophilic bias can be stated as follows: users strongly connected w.r.t. the link weights $\omega$, tend to be associated with the same label.

The complexity measure used for this problem is the *robustified resistance weighted cutsize* $\Psi(y)$, which we now define formally. Within the graph-based learning context, the *cutsize* is defined as the number of edges connecting users with different labels, i.e., $\sum_{u<v} [\![\omega_{u,v} \neq 0]\!][\![y(u) \neq y(v)]\!]$, and the *weighted cutsize* is defined as the sum of the edge weights $\omega_{u,v}$ over all pairs of users $u$ and $v$ having different labels, i.e., $\sum_{u<v} \omega_{u,v}[\![y(u) \neq y(v)]\!]$ [14]. The *effective resistance* between two given nodes $u$ and $v$ of a graph is a commonly used measure that expresses the degree of the connection strength between $u$ and $v$ (see, e.g., [66]). More precisely, viewing the graph as an electrical circuit, where each edge weight $\omega_{u,v}$ corresponds to a $\frac{1}{\omega_{u,v}}$ resistor, the effective resistance between $u$ and $v$ is the power required to hold between them a unit voltage difference for a unit time. Informally, the more there are paths between two nodes $u$ and $v$ that are short, edge-disjoint and formed by edges with large weights, the lower is $r(u, v)$ because the amount of flow between the two considered nodes is larger. A formal definition of effective resistance $r(u, v)$ between users $u$ and $v$ is

$$r(u, v) := \frac{1}{\min_{\boldsymbol{x} \in \mathbb{R}^N} \{ \sum_{i<j}^N \omega_{i,j}(x_i - x_j)^2 : x_u - x_v = 1 \}} \ .$$

Interestingly enough, for all $u, v \in [N]$, $r(u, v)$ is exactly equal to the probability that the edge $\{u, v\}$ is included in a uniformly generated random spanning tree of the given user graph $\mathcal{G}$ (see, e.g., [66]).

The *resistance weighted cutsize* $\Phi(y)$ [67] is the weighted sum of the effective resistances $r(u, v)$ between any two nodes $u$ and $v$ with different labels. i.e.,

$$\Phi(y) := \sum_{u<v}^N \omega_{u,v} r(u, v) [\![y(u) \neq y(v)]\!] \tag{4}$$

and then its robustifcation is defined as

$$\Psi(y) := 1 + \min_{z:[N] \to [K]} \left( \Phi(z) + \sum_{u \in [N]} [\![z(u) \neq y(u)]\!] \right). \tag{5}$$

The first quantity (4) can be viewed as a dispersion measure based on the above mentioned homophilic tendency. It has several advantages compared to the weighted cutsize in measuring the degree of homophily violation [67]. The most significant property is that it is *locally* density-dependent because the contribution to $\Phi(y)$ of each edge $(u, v)$ such that $y(u) \neq y(v)$ is inversely proportional to how strongly $u$ and $v$ are connected in their user graph local area. Indeed, because of the effective resistance, the potential contribution to $\Phi(y)$ of the edges in dense areas is smaller than the ones of the edges in sparse areas. In fact if the graph is well-clustered i.e., it can be partitioned into dense clusters (many intra-cluster edges) and fewer inter-cluster edges and the labeling $y$ respects these clusters then in many cases $\Phi(y) \ll N$. As an archetypical instance consider the following proposition where the clusters are represented by cliques.

**Proposition 1.** *Consider an unweighted graph $\mathcal{G}$ partitioned into $G$ clusters and a labeling function $y(\cdot)$, where each cluster is an $n$-clique and, if $u, v$ are vertices in the cluster, then $y(u) = y(v)$. For any pair of such clusters $C, C' \subset \mathcal{G}$, suppose that there are $\frac{n-1}{G}$ edges connecting the nodes of $C$ with the nodes of $C'$. Then we have $\Phi(y) \in \mathcal{O}(G)$.*

> **CONSTRUCTBST-$\mathcal{C}$** (User graph: $\mathcal{G}$)
>
> 1. Sample a uniform random spanning tree $\mathcal{T}$ from the user graph.
>
> 2. Perform a depth-first visit of $\mathcal{T}$ to provide an order of the users. Without loss of generality assume that, for all $u \in [N]$, we have that user $u$ is the $u$-th vertex visited.
>
> 3. Construct a perfect full oriented binary tree $\mathcal{C}$ of depth $h := \log_2(N)$ whose $u$-th leftmost leaf of its graphical representation is user $u$.[a]
>
> ──────────────
>
> [a] In this context, by *oriented* we mean that the leaves of $\mathcal{C}$ are numbered sequentially from the leftmost to the rightmost one so that, for each internal vertex of $\mathcal{C}$, both its left and right subtree contain subsets of leaves uniquely determined by the depth-first visit of $\mathcal{T}$.

Figure 1: Binary Support Tree Construction Algorithm

Thus in this archetypical case our regret bounds now scale strongly with the number of clusters of users $G$ (see (3)) rather than with the number of users $N$ (comparing to the baseline (2)).

The second quantity (5) is an extension of $\Phi(y)$ to deal with adversarial label perturbation, viz., capturing the regularity of all labelings $y$ such that $\Phi(y)$ can be dramatically reduced by simply changing the labels of a relatively small number of users. To give an insight into the advantages of $\Psi(y)$ w.r.t. $\Phi(y)$ regarding its noise-tolerance property, consider an input star graph with all edge weights equal to $1$ and where all vertex labels are equal except for the one of the central node $u$. It is natural to consider this labeling regular w.r.t. our bias, because it is sufficient to change only $y(u)$ to obtain a cutsize equal to $0$. This is precisely the labeling property that is captured by $\Psi(y)$, which is equal to the minimum, over all labelings $z$, of the sum of $\Phi(z)$ and the number of vertices for which $y$ and $z$ differ (plus 1). In this case we have therefore $\Psi(y) = 1 + \Phi(y^*) + 1 = 2$, where $y^*$ is the labeling obtained by changing $y(u)$ to make it equal to all other labels, so that $\Phi(y^*) = 0$, whereas $\Phi(y) = N - 1$.

## 2.2 An Embedding to Enable Fast Computation

A uniformly generated random spanning tree (RST) is defined as a spanning tree selected with a probability proportional to the product of the weights of all its edges (see, e.g., [66]). It represents a fundamental tool in several mathematical fields, e.g., combinatorial geometry, algebraic graph theory, stationary Markov chains [68], and can be viewed as a way to summarize the topological information of the input network. When the input graph is weighted as in our case, it can be generated in time almost linear in the number of edges [69, 70].

In a preliminary phase, our algorithms operate as follows (see Fig. 1). A RST $\mathcal{T}$ of the input social network is drawn (step 1). Thus, an order of the $N$ users is determined through a depth-first visit of $\mathcal{T}$ (step 2). From here on we assume, without loss of generality, that user $u \in [N]$ is the $u$-th vertex visited. This step is necessary to make the algorithms noise-tolerant, and is strictly related to the improvement of the complexity measure $\Psi(y)$ over $\Phi(y)$. Finally, a full perfect binary tree, called the Binary Support Tree (BST), and having the users, ordered from left to right, as leaves (step 3) is constructed. The BST forms the geometry that underlies the data-structures of our algorithms.

We conclude this section by showing a result which will be useful in the analysis of our algorithms, and stems directly *only* from the user order determined by the depth-first visit of $\mathcal{T}$. If we consider the line graph $\mathcal{L}$ connecting the users $u$ with $u + 1$ for all $u \in [N - 1]$, we have that, as stated in the following theorem, the cutsize of $\mathcal{L}$ is at most twice the robustified resistance weighted cutsize of the input user graph. This result can be viewed as the multi-class extension of part 2 of Theorem 6 in [67].

**Lemma 2** ([67, Theorem 6]). *For any given input user graph, we have*

$$\mathbb{E}\left[ \sum_{u \in [N-1]} [\![ y(u) \neq y(u+1) ]\!] \right] \leq 2\Psi(y) ,$$

*where the expectation is over the draw of the uniform random spanning tree $\mathcal{T}$.*

$\boxed{\begin{array}{l}\textbf{SPECIALISTEXP} \text{ (Learning rate } \eta > 0\text{; Distribution } w_1 : \mathbb{S} \to [0,1] \text{ s.t. } \sum_{s\in\mathbb{S}} w_1(s) = 1.\text{)}\\[4pt]\hline\\\textbf{For } t = 1, \ldots, T \textbf{ do}\\\quad 1.\ \forall a \in [K], \quad p_{t,a} \leftarrow \sum_{s \in \mathbb{S} : s(u_t) = a} w_t(s);\\[4pt]\quad 2.\ \textbf{Predict } a_t \text{ by drawing from } [K] \text{ with probability } \mathbb{P}\left[a_t = a\right] := p_{t,a}/\|\boldsymbol{p}_t\|_1;\\[4pt]\quad 3.\ \textbf{Receive } \ell_{t,a_t}\\[4pt]\quad 4.\ \lambda_t \leftarrow \exp(-\eta \ell_{t,a_t} \|\boldsymbol{p}_t\|_1/p_{t,a_t}); \quad z_t \leftarrow \|\boldsymbol{p}_t\|_1/(\|\boldsymbol{p}_t\|_1 - (1 - \lambda_t)p_{t,a_t});\\[4pt]\quad 5.\ \forall s \in \mathbb{S}: \qquad\qquad w_{t+1}(s) \leftarrow \begin{cases} w_t(s) & s(u_t) = \square \\ w_t(s)z_t & s(u_t) \neq a_t \\ w_t(s)z_t\lambda_t & s(u_t) = a_t \end{cases}\end{array}}$

Figure 2: SPECIALISTEXP Algorithm

# 3 Predicting with Specialists

We build on the *prediction with expert advice framework* [71, 72, 73, 74], specifically that with bandit feedback: pioneered by the EXP4 algorithm [18]. This type of online algorithm maintains a distribution over a set of predictors ("experts"). After the predictors predict they incur a loss and the distribution is updated accordingly. Although, except in special cases, this procedure does not have a natural Bayesian interpretation, probabilistic methods still may be transferred into the expert advice framework. In particular we will exploit an analogue of message-passing as used in graphical models [75] to predict very efficiently over exponentially-sized sets of predictors. Broadly speaking we would like build a graphical model that is isomorphic to the user graph $\mathcal{G}$. However it is well-known that exact prediction with graphical models that contain cycles is NP-hard [76]. Thus a benefit of the embedding to a BST (see Section 2.2) is that it enables fast and exact computation as the graph is now cycle-free and Lemma 2 ensures that the embedding only modestly increases our regret bounds. Surprisingly, we improve in terms of computation over standard message passing techniques, i.e., if we embedded to a "line" graph we would require $\mathcal{O}(KN)$ time to predict [75] per trial or using the method of [77] $\mathcal{O}(K^3 \log N)$ time. However, we will require only $\mathcal{O}(K \log N)$ and $\mathcal{O}(\log K \log N)$ for the GABA-I and GABA-II algorithms respectively (see Figures 3 and 4). To accomplish this technically we adapt the method of *specialists* [15, 16].

A specialist is a prediction function $s : [N] \to \{1, 2, \ldots, K, \square\}$ from a context space to an extended output space with *abstentions*. For us the context space is just the set of users $[N]$; and the extended output space is $\{1, 2, \ldots, K, \square\}$ where $[K]$ corresponds to predicted actions, but '$\square$' indicates that the specialist abstains from predicting an action. Thus a specialist *specializes* its prediction to part of the context space. We denote the set of all specialists as $\mathbb{S} := \{1, \ldots, K, \square\}^{[N]}$. As a single specialist only predicts over part of the context space, we need a set of specialists $\mathcal{S} \subseteq \mathbb{S}$ if we wish to define a function that predicts an *action* for every context. A specialist set $\mathcal{S} \subseteq \mathbb{S}$ is *well-formed* if for each $u \in [N]$ there exists a unique specialist $s \in \mathcal{S}$ such that $s(u) \in [K]$. For such a user $u$ and specialist $s$ we then define $\mathcal{S}^\dagger(u) := s(u)$ so that $\mathcal{S}^\dagger$ is a function from $[N]$ into $[K]$. Finally a specialist *model* is defined by giving a distribution $w_1 : \mathbb{S} \to [0,1]$ s.t. $\sum_{s\in\mathbb{S}} w_1(s) = 1$. To predict with specialists we adapt [15] to the EXP3/4 [18] setting giving the SPECIALISTEXP algorithm (see Figure 2). We then bound the regret by combining the analysis of [15, 18] into the following theorem.

**Theorem 3.** *The expected regret of* SPECIALISTEXP *with initial specialist distribution* $w_1 : \mathbb{S} \to [0,1]$ *and learning rate* $\eta > 0$ *is bounded above by*

$$\mathbb{E}\left[\sum_{t \in [T]} \ell_{t,a_t} - \ell_{t,\mathcal{S}^\dagger(u_t)}\right] \leq \frac{1}{\eta}\sum_{s \in \mathcal{S}} \ln\left(\frac{1}{w_1(s)|\mathcal{S}|}\right) + \frac{\eta KT}{2} \qquad (6)$$

*for all well-formed specialist sets* $\mathcal{S} \subseteq \mathbb{S}$.

In the following we give the two distributions that define the two specialist models corresponding to GABA-I and GABA-II in (7) and (9), and in the supplementary material we detail how these distributions lead to the regret bounds in Corollaries 4 and 5.

We now give the distribution $w_1(\cdot)$ over $\mathbb{S}$ that defines the GABA-I model. The model has a single parameter $\phi \in (0, 1)$ and we give the following helper functions to define the distribution,

$$
\begin{aligned}
\text{valid1}(s) &:= [\![\forall u, v \in [N] : s(u) = s(v) \text{ or } s(u) = \square \text{ or } s(v) = \square]\!] \\
\text{cut}(s) &:= \sum_{u \in [N-1]} [\![s(u) \neq s(u+1)]\!] \\
\text{startfactor}(s) &:= \frac{K-1}{K} [\![s(1) \neq \square]\!] + \frac{1}{K} [\![s(1) = \square]\!].
\end{aligned}
$$

The function $\text{valid1}(\cdot)$ determines the support of $w_1(\cdot)$ which are the specialists that predict a unique action or abstain, hence the cardinality of the support of $w_1(\cdot)$ is $K \times (2^N - 1) + 1$. The remaining two functions quantitatively determine probability mass of a specialist as:

$$
w_1(s) := \text{valid1}(s) \times \frac{1}{K} \times \text{startfactor}(s) \times (1-\phi)^{N-1-\text{cut}(s)} \phi^{\text{cut}(s)} \quad (\forall s \in \mathbb{S}). \tag{7}
$$

We note that this specialist selection is similar to that of the Markov circadian specialists in [16] except that the nodes of the Markov chain are now users instead of trials.

**Corollary 4.** *The expected regret of* SPECIALISTEXP *with distribution* $w_1(\cdot)$ *as defined by* (7) *with parameter* $\phi = 4\Psi(y)/(K(N-1))$, *learning rate* $\eta = \sqrt{\frac{10\Psi(y)\ln(KN/\Psi(y))}{KT}}$ *and with* $\Psi(y) \leq (N-1)/4$ *is bounded above by:*

$$
\mathbb{E}\left[\sum_{t \in [T]} \ell_{t,a_t} - \ell_{t,y(u_t)}\right] \in \mathcal{O}\left(\sqrt{K\ln\left(\frac{KN}{\Psi(y)}\right)\Psi(y)T}\right) \tag{8}
$$

*for any mapping of users to actions* $y : [N] \to [K]$.

We now give the distribution $w_1(\cdot)$ over $\mathbb{S}$ that defines the GABA-II model. Whereas for GABA-I the cardinality of the support was exponential in $N$, for GABA-II the cardinality is just $K(2N - 1)$. The supported specialists in GABA-II predict a unique action over a contiguous $l, \ldots, r$ and abstain everywhere else, thus they are of the form:

$$
s_a^{l,r}(u) := \begin{cases} a & u \in \{l, \ldots, r\} \\ \square & u \notin \{l, \ldots, r\} \end{cases},
$$

but not all contiguous segments are supported. The segments supported are those that correspond to the set of all leaf-descendents of a node in the BST (see Section 2.2). As an example if $N = 4$ the supported $(l, r)$ segments are $\{(1, 1), (2, 2), (3, 3), (4, 4), (1, 2), (3, 4), (1, 4)\}$. Expressing this algebraically leads to a relatively complex "validity" function

$$
\text{valid2}(s) := [\![\exists a \in [K]; l, r \in [N]; i, j \in [\log_2 N] : 1+r-l = 2^i \text{ and } l = 2^i(j-1)+1 \text{ and } s = s_a^{l,r}]\!],
$$

and then the distribution is defined as,

$$
w_1(s) := \text{valid2}(s) \times \frac{1}{K(2N-1)} \quad (\forall s \in \mathbb{S}). \tag{9}
$$

We note that this selection of specialists is a simple multi-action extension of those defined in [65].

**Corollary 5.** *The expected regret of* SPECIALISTEXP *with distribution* $w_1(\cdot)$ *as defined by* (9) *with learning rate* $\eta = \sqrt{\frac{8\Psi(y)\log_2(eN/\Psi(y))\ln(3KN/2\Psi(y))}{KT}}$ *and with* $\Psi(y) \leq N/2$ *is bounded above by:*

$$
\mathbb{E}\left[\sum_{t \in [T]} \ell_{t,a_t} - \ell_{t,y(u_t)}\right] \in \mathcal{O}\left(\sqrt{K\ln\left(\frac{N}{\Psi(y)}\right)\ln\left(\frac{KN}{\Psi(y)}\right)\Psi(y)T}\right) \tag{10}
$$

*for any mapping of users to actions* $y : [N] \to [K]$.

## 4 The GABA Algorithms

We now introduce the GABA algorithms. Both algorithms are based on the BST $\mathcal{C}$ (see Section 2.2).

## 4.1 GABA-I

Since we have an exponential number of non-zero weight specialists in GABA-I a direct implementation of SPECIALISTEXP would take per-trial time and space exponential in $N$. We now describe how GABA-I implements SPECIALISTEXP, bringing the per-trial time down to $\mathcal{O}(K \ln(N))$ and the space down to $\mathcal{O}(KN)$. The implementation works by, for each action independently, performing online belief propagation [64] over the tree $\mathcal{C}$. We note that each of these $K$ online belief propagations is over two states $\{0, 1\}$ and hence takes a per-trial time of only $\mathcal{O}(\ln(N))$. We now detail this procedure:

GABA-I maintains a vector valued function $\boldsymbol{\alpha}_t : \mathcal{C} \times \{0, 1\} \times \{0, 1\} \to \mathbb{R}^K$ which, for all $i, j \in \{0, 1\}$ and $t \in [T]$, has the following properties:

$$\forall u \in [N] \setminus \{u_t\}, \quad \boldsymbol{\alpha}_{t+1}(u, i, j) = \boldsymbol{\alpha}_t(u, i, j) \tag{11}$$

and for all internal vertices $n$ of $\mathcal{C}$ we have:

$$\boldsymbol{\alpha}_t(n, i, j) = \sum_{k \in \{0, 1\}} \boldsymbol{\alpha}_t(\triangleleft(n), i, k) \odot \boldsymbol{\alpha}_t(\triangleright(n), k, j) \tag{12}$$

On trial $t$ GABA-I computes $\boldsymbol{p}_t$ by sending vector valued messages down the path in $\mathcal{C}$ from the root to $u_t$. Specifically, we construct the left and right message functions $\boldsymbol{\beta}_t^{\Leftarrow}, \boldsymbol{\beta}_t^{\Rightarrow} : \Uparrow(u_t) \times \{0, 1\} \to \mathbb{R}^K$ as follows. Each (non-root, proper) ancestor $n$ of $u_t$ receives, for $i \in \{0, 1\}$, $K$ dimensional vector messages $\boldsymbol{\beta}_t^{\Leftarrow}(\uparrow(n), i)$ and $\boldsymbol{\beta}_t^{\Rightarrow}(\uparrow(n), i)$ from its parent and then constructs its own messages $\boldsymbol{\beta}_t^{\Leftarrow}(n, i)$ from $\boldsymbol{\beta}_t^{\Leftarrow}(\uparrow(n), j)$ and $\boldsymbol{\alpha}_t(\triangleleft(\uparrow(n)), j, i)$ and messages $\boldsymbol{\beta}_t^{\Rightarrow}(n, i)$ from $\boldsymbol{\beta}_t^{\Rightarrow}(\uparrow(n), j)$ and $\boldsymbol{\alpha}_t(\triangleright(\uparrow(n)), i, j)$, for all $i, j \in \{0, 1\}$. It then sends these messages to its child that is next on the path to $u_t$. Once $u_t$ has received the messages from its parent it combines them with $\boldsymbol{\alpha}_t(u_t, 1, i)$ (for $i \in \{0, 1\}$) to create $\boldsymbol{p}_t$.

On the receipt of $\ell_{t, a_t}$ we update the function $\boldsymbol{\alpha}_t$ to $\boldsymbol{\alpha}_{t+1}$ noting that by (11) and (12) we need only modify the values $\boldsymbol{\alpha}_t(n, i, j)$ when $n$ is an ancestor of $u_t$.

## 4.2 GABA-II

For GABA-II we have $\mathcal{O}(K \ln(N))$ non-zero weight specialists that don't abstain on any given trial so a direct implementation of SPECIALISTEXP would take a per-trial time of $\mathcal{O}(K \ln(N))$. We now show how GABA-II implements SPECIALISTEXP, which takes the per-trial time down to $\mathcal{O}(\ln(K) \ln(N))$ whilst maintaining the space complexity of $\mathcal{O}(KN)$.

We first note that SPECIALISTEXP maintains a weight for each specialist. For any vertex $n$ of $\mathcal{C}$ and any action $a$, the weight, on trial $t$, of the specialist that predicts $a$ whenever $u_t$ is its descendant and abstains otherwise, is kept, by GABA-II in the following factored form:

$$\frac{\mu_t(n) \theta_t(n, a)}{K(2N - 1)} \tag{13}$$

where $\mu_{t+1}(n) := \mu_t(n)$ whenever $n \notin \Uparrow(u_t)$, and $\theta_{t+1}(n, a) := \theta_t(n, a)$ whenever $n \notin \Uparrow(u_t)$ or $a \neq a_t$.

In addition to the tree $\mathcal{C}$, GABA-II also works with an oriented full binary tree $\mathcal{B}$ whose leaves are the actions (in this overview we assume that the cardinality of the action set is an integer power of two, although this is not required by GABA-II). For any vertex $n$ of $\mathcal{C}$ the function $\theta_t(n, \cdot)$ is extended onto all internal vertices of $\mathcal{B}$ by the following inductive relationship:

$$\theta_t(n, m) := \theta_t(n, \triangleleft(m)) + \theta_t(n, \triangleright(m)) \tag{14}$$

To sample the action $a_t$ GABA-II first samples an ancestor $\delta_t$ of $u_t$ with probability $\mathbb{P}[\delta_t = n] \propto \mu_t(n) \theta_t(n, r)$ where $r$ is the root of $\mathcal{B}$. GABA-II then uses the function $\theta_t(\delta_t, \cdot)$ to sample action $a_t$ with probability $\mathbb{P}[a_t = a \mid \delta_t = n] = \theta_t(n, a)/\theta_t(n, r)$ in $\mathcal{O}(\ln(K))$ time. The law of total probability and (13) can then be used to show that $\mathbb{P}[a_t = a] \propto p_{t, a}$ where $p_{t, a}$ is as defined in SPECIALISTEXP.

On the receipt of $\ell_{t, a_t}$ we update the functions $\mu_t$ and $\theta_t$ to $\mu_{t+1}$ and $\theta_{t+1}$ noting that by the equalities between these functions and (14) we need only modify the values $\mu_t(n)$ and $\theta_t(n, m)$ when $n$ is an ancestor of $u_t$ and $m$ is an ancestor of $a_t$.

**GABA-I** (Learning rate : $\eta > 0$; Model parameter: $\phi \in (0,1)$)

0. Construct binary support tree $\mathcal{C}$ via CONSTRUCTBST-$\mathcal{C}$ algorithm (see Figure 1).

1. $\forall$ leaf $n \in \mathcal{C}, \forall i, j \in \{0,1\}, \quad \boldsymbol{\alpha}_1(n,i,j) \leftarrow [\![i \neq j]\!]\phi\mathbf{1} + [\![i = j]\!](1 - \phi)\mathbf{1};$

2. **For** $d = 1, 2, \ldots, h-1, \forall n \in \mathcal{C}$ at depth $h-d, \forall i, j \in \{0,1\}$, **do**
   $\boldsymbol{\alpha}_1(n,i,j) \leftarrow \sum_{k \in \{0,1\}} \boldsymbol{\alpha}_1(\triangleleft(n),i,k) \odot \boldsymbol{\alpha}_1(\triangleright(n),k,j);$

**For** $t = 1, 2 \ldots T$, **do**

3. $\forall d \in [h] \cup \{0\} \quad \nu_{t,d} \leftarrow$ ancestor of $u_t$ at depth $d$ in $\mathcal{C}$;

4. $\forall i \in \{0,1\}, \quad \boldsymbol{\beta}_t^{\Leftarrow}(\nu_{t,0},i) \leftarrow (1 + [\![i=0]\!](K-2))\mathbf{1}/K; \quad \forall i \in \{0,1\}, \quad \boldsymbol{\beta}_t^{\Rightarrow}(\nu_{t,0},i) \leftarrow \mathbf{1};$

5. **For** $d = 1, 2, \ldots, h$, **do**
   (a) **if** $\nu_{t,d} = \triangleleft(\nu_{t,d-1})$ **then** $\forall i \in \{0,1\}$
       i. $\boldsymbol{\beta}_t^{\Leftarrow}(\nu_{t,d},i) \leftarrow \boldsymbol{\beta}_t^{\Leftarrow}(\nu_{t,d-1},i);$
       ii. $\boldsymbol{\beta}_t^{\Rightarrow}(\nu_{t,d},i) \leftarrow \sum_{j \in \{0,1\}} \boldsymbol{\alpha}_t(\triangleright(\nu_{t,d-1}),i,j) \odot \boldsymbol{\beta}_t^{\Rightarrow}(\nu_{t,d-1},j);$
   (b) **if** $\nu_{t,d} = \triangleright(\nu_{t,d-1})$ **then** $\forall i \in \{0,1\}$
       i. $\boldsymbol{\beta}_t^{\Rightarrow}(\nu_{t,d},i) \leftarrow \boldsymbol{\beta}_t^{\Rightarrow}(\nu_{t,d-1},i);$
       ii. $\boldsymbol{\beta}_t^{\Leftarrow}(\nu_{t,d},i) \leftarrow \sum_{j \in \{0,1\}} \boldsymbol{\beta}_t^{\Leftarrow}(\nu_{t,d-1},j) \odot \boldsymbol{\alpha}_t(\triangleleft(\nu_{t,d-1}),j,i);$

6. $\bar{\boldsymbol{p}}_t \leftarrow (1/K) \sum_{i \in \{0,1\}} \boldsymbol{\beta}_t^{\Leftarrow}(\nu_{t,h},1) \odot \boldsymbol{\alpha}_t(\nu_{t,h},1,i) \odot \boldsymbol{\beta}_t^{\Rightarrow}(\nu_{t,h},i);$

7. **Predict** $a_t \in [K]$ with probability $\mathbb{P}[a_t = a] = \bar{p}_{t,a}/\|\bar{\boldsymbol{p}}_t\|_1;$

8. **Receive** $\ell_{t,a_t}$

9. $\forall a \in [K], c_{t,a} \leftarrow \exp(-\eta[\![a = a_t]\!]\ell_{t,a_t}\|\bar{\boldsymbol{p}}_t\|_1/\bar{p}_{t,a}); \quad \boldsymbol{\pi}^t \leftarrow (\|\bar{\boldsymbol{p}}_t\|_1 \boldsymbol{c}_t)/(\bar{\boldsymbol{p}}_t \cdot \boldsymbol{c}_t);$

10. $\forall i \in \{0,1\}, \quad \boldsymbol{\alpha}_{t+1}(\nu_{t,h},1,i) \leftarrow \boldsymbol{\pi}^t \odot \boldsymbol{\alpha}_t(\nu_{t,h},1,i);$

11. $\forall i \in \{0,1\}, \quad \boldsymbol{\alpha}_{t+1}(\nu_{t,h},0,i) \leftarrow \boldsymbol{\alpha}_t(\nu_{t,h},0,i);$

12. $\forall n \in \mathcal{C} \setminus \{\nu_{t,d} \mid d \in [h] \cup \{0\}\}, \forall i, j \in \{0,1\}, \quad \boldsymbol{\alpha}_{t+1}(n,i,j) \leftarrow \boldsymbol{\alpha}_t(n,i,j);$

13. **For** $d = 1, 2, \ldots, h-1$, **do** $\forall i, j \in \{0,1\}$
    $\boldsymbol{\alpha}_{t+1}(\nu_{t,(h-d)},i,j) \leftarrow \sum_{k \in \{0,1\}} \boldsymbol{\alpha}_{t+1}(\triangleleft(\nu_{t,(h-d)}),i,k) \odot \boldsymbol{\alpha}_{t+1}(\triangleright(\nu_{t,(h-d)}),k,j);$

Figure 3: GABA-I Algorithm

**GABA-II** (Learning rate: $\eta > 0$)

0. Construct binary support tree $\mathcal{C}$ via CONSTRUCTBST-$\mathcal{C}$ algorithm (see Figure 1).

1. Construct a full perfect oriented binary tree $\mathcal{B}$ with height $g := \lceil \log_2(K) \rceil$, whose first $K$ leaves represent the actions $[K]$; Set $r$ to be the root of $\mathcal{B}$;

2. $\forall$ vertex $n \in \mathcal{C}$:
   (a) $\mu_1(n) \leftarrow 1; \quad \forall$ leaf $m \in \mathcal{B}, \quad$ **if** $m \in [K]$ **then** $\theta_1(n,m) \leftarrow 1$; **else** $\theta_1(n,m) \leftarrow 0;$
   (b) $\forall d \in \{1, 2, \ldots, g\}, \forall m \in \mathcal{B}$ at depth $g-d, \quad \theta_1(n,m) := \theta_1(n,\triangleleft(m)) + \theta_1(n,\triangleright(m));$

**For** $t = 1, 2 \ldots T$, **do**

3. Draw $\delta_t$ from $\Uparrow(u_t)$ with prob. $\mathbb{P}[\delta_t = n] \propto \mu_t(n)\theta_t(n,r); \quad \zeta_{t,0} \leftarrow r;$

4. **For** $d = 0, \ldots, g-1$: draw $\zeta_{t,d+1}$ from $\{\triangleleft(\zeta_{t,d}), \triangleright(\zeta_{t,d})\}$ with prob. $\mathbb{P}[\zeta_{t,d+1} = m] \propto \theta_t(\delta_t,m);$

5. **Predict** $a_t \leftarrow \zeta_{t,g};$

6. **Receive** $\ell_{t,a_t}$

7. $\psi_t \leftarrow \sum_{n \in \Uparrow(u_t)} \mu_t(n)\theta(n,r); \quad \varrho_t \leftarrow \sum_{n \in \Uparrow(u_t)} \mu_t(n)\theta(n,a_t); \quad \bar{\lambda}_t \leftarrow \exp(-\eta\ell_{t,a_t}\psi_t/\varrho_t);$

8. $\forall n \in \Uparrow(u_t)$:
   (a) $\mu_{t+1}(n) \leftarrow \mu_t(n)\psi_t/(\psi_t - (1 - \bar{\lambda}_t)\varrho_t); \quad \theta_{t+1}(n,a_t) \leftarrow \bar{\lambda}_t\theta_t(n,a_t);$
   (b) $\forall m \in \mathcal{B} \setminus \Uparrow(a_t), \quad \theta_{t+1}(n,m) := \theta_t(n,m);$
   (c) **For** $d = 1, 2, \ldots g$: $\quad \theta_{t+1}(n, \zeta_{t,(g-d)}) \leftarrow \theta_{t+1}(n, \triangleleft(\zeta_{t,(g-d)})) + \theta_{t+1}(n, \triangleright(\zeta_{t,(g-d)}));$

9. $\forall n \in \mathcal{C} \setminus \Uparrow(u_t), \quad \mu_{t+1}(n) := \mu_t(n); \quad \forall m \in \mathcal{B}, \theta_{t+1}(n,m) := \theta_t(n,m);$

Figure 4: GABA-II Algorithm

### 4.3 Parameter Tuning

A limitation of the GABA-I regret bound is that it is dependent on knowing the optimal values of the parameters $\phi$ and $\eta$, and for GABA-II on the parameter $\eta$. In the following, we will 1) sketch how to autotune $\phi$ at little cost and 2) autotune $\eta$, however at essentially the cost of moving $\Psi(y)$ outside of the square root.

We first sketch how to automatically tune the parameter $\phi$ that appears in GABA-I. Assume, without loss of generality, that $N$ is an integer power of 2. The idea of our tuning method is that since $\phi$ is unknown we will "mix" over possible values of $\phi \in [0, 1]$. In fact, at little cost in regret it is sufficient to just mix over the exponentially increasing values of $\phi = 2/N, 4/N, 8/N, \ldots, N/N$. Thus each specialist is split into $\log_2 N$ specialists, so that the new distribution over specialists is

$$w_1(s_\phi) := \frac{1}{\log_2 N} \times \text{valid1}(s) \times \frac{1}{K} \times \text{startfactor}(s) \times (1 - \phi)^{N-1-\text{cut}(s)} \phi^{\text{cut}(s)},$$

where $\sum_{s \in \mathbb{S}, \phi \in \{2/N, 4/N, \ldots, 1\}} w_1(s_\phi) = 1$. Implementing this efficiently is similar to the implementation of GABA-I, except that we now have $\log_2 N$ copies of the BST $\mathcal{C}_\phi$, each initialized with a different value of $\phi$. On each trial the computed values from the $\log_2 N$ copies of the BST $\mathcal{C}_\phi$ are summed to find the prediction vector. After receipt of the loss, all copies of the BST are updated as in GABA-I. The regret bound of this autotuning with respect to $\phi$ is equal, up to an $\mathcal{O}(\sqrt{\log(\log(N))})$ factor, to that of GABA-I with the optimal $\phi$, but comes at the cost of an additional $\mathcal{O}(\log(N))$ factor in the computation time.

Now that we have shown how to automatically tune $\phi$ in GABA-I we are left with the learning rate $\eta$ in both algorithms. We first note that, with any $\eta$, the regret of both algorithms is $\Upsilon/\eta + \eta KT/2$, where $\Upsilon$ is the robustified resistance weighted cutsize $\Psi(y)$ multiplied by logarithmic terms (one in GABA-I and two in GABA-II). By setting $\eta = \sqrt{2/KT}$ we get a regret of $(\Upsilon + 1)\sqrt{KT/2}$. In addition, if $T$ is unknown then a doubling trick can be performed with this result to get a regret bound of $\mathcal{O}(\Upsilon\sqrt{KT})$ with no parameters needed. We compare this to the regret bound of $\mathcal{O}(\sqrt{\Upsilon KT})$ that comes from the optimal tuning of $\eta$. It remains an open problem to bring $\Upsilon$ inside the square-root.

Even with the above knowledge-free tuning of $\eta$, our methods improve over the *baseline* comparator of running an *independent* EXP3 algorithm for each of the $N$ users in many natural scenarios. Recall that in this case the induced regret is then $\widetilde{\mathcal{O}}(\sqrt{NKT})$ (see (2)). Consider a very large social network where the bandit problem is to show 1-of-$K$ advertisements (for simplicity assume $K \in \mathcal{O}(1)$) at the nodes (users). Now consider the case that each user is served at most *one* advert, i.e., there is at most a single trial for any given user. Since $N \geq T$ the bound of the baseline is now the vacuous regret $\Theta(T)$. We can intuitively see that this analysis is correct since the baseline algorithm is now just picking a single "uniformly at random" advertisement from $[K]$ for each user independently. However, observe that when $\Psi(y) \in \tilde{o}(\sqrt{T})$ we get $\widetilde{\mathcal{O}}(\Psi(y)\sqrt{T}) \subseteq \tilde{o}(T)$, which is non-vacuous. Intuitively, GABA-I/II may achieve this result since algorithmically they are exploiting the network structure.

## 5 Conclusion

We considered a contextual, non-stochastic bandit problem in which the finite set of contexts (a.k.a users) form a social network and the inductive bias is that if the social link between two users is strong then actions that perform well for one of these users are likely to perform well for the other. We gave two highly efficient algorithms for this problem, both with good regret bounds. Since this work is theoretical in nature we cannot foresee any potential negative societal impacts.

In the future it may be interesting to investigate extensions of our algorithms to the stochastic setting, as well as continuous bandit settings. Finally, it would be valuable to study potential applications of our algorithms, with large scale recommender systems being a natural candidate. On the theory side our bounds are based on an exponential potential function. Improved adversarial regret bounds were proven for an alternate potential function in [19] and it is an open question if our techniques can be extended to that potential.

**Acknowledgements.** Mark Herbster was supported by the U.S. Army Research Laboratory and the U.K. Ministry of Defence under Agreement Number W911NF-16-3-0001. Stephen Pasteris and Massimiliano Pontil were supported in part by EPSRC Grant N. EP/P009069/1 and SAP SE.

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
