\mathrm{valid1}(s) \times \frac{1}{K} \times \mathrm{startfactor}(s) \times (1-\phi)^{N-1-\mathrm{cut}(s)} \phi^{\mathrm{cut}(s)},$$

where $\sum_{s \in \mathbb{S}, \phi \in \{2/N, 4/N, \ldots, 1\}} w_1(s_\phi) = 1$. Implementing this efficiently is similar to the implementation of GABA-I, except that we now have $\log_2 N$ copies of the BST $\mathcal{C}_\phi$, each initialized with a different value of $\phi$. On each trial the computed values from the $\log_2 N$ copies of the BST $\mathcal{C}_\phi$ are summed to find the prediction vector. After receipt of the loss, all copies of the BST are updated as in GABA-I. The regret bound of this autotuning with respect to $\phi$ is equal, up to an $\mathcal{O}(\sqrt{\log(\log(N))})$ factor, to that of GABA-I with the optimal $\phi$, but comes at the cost of an additional $\mathcal{O}(\log(N))$ factor in the computation time.

Now that we have shown how to automatically tune $\phi$ in GABA-I we are left with the learning rate $\eta$ in both algorithms. We first note that, with any $\eta$, the regret of both algorithms is $\Upsilon/\eta + \eta KT/2$, where $\Upsilon$ is the robustified resistance weighted cutsize $\Psi(y)$ multiplied by logarithmic terms (one in GABA-I and two in GABA-II). By setting $\eta = \sqrt{2/KT}$ we get a regret of $(\Upsilon + 1)\sqrt{KT/2}$. In addition, if $T$ is unknown then a doubling trick can be performed with this result to get a regret bound of $\mathcal{O}(\Upsilon\sqrt{KT})$ with no parameters needed. We compare this to the regret bound of $\mathcal{O}(\sqrt{\Upsilon KT})$ that comes from the optimal tuning of $\eta$. It remains an open problem to bring $\Upsilon$ inside the square-root.

Even with the above knowledge-free tuning of $\eta$, our methods improve over the *baseline* comparator of running an *independent* EXP3 algorithm for each of the $N$ users in many natural scenarios. Recall that in this case the induced regret is then $\widetilde{\mathcal{O}}(\sqrt{NKT})$ (see (2)). Consider a very large social network where the bandit problem is to show 1-of-$K$ advertisements (for simplicity assume $K \in \mathcal{O}(1)$) at the nodes (users). Now consider the case that each user is served at most *one* advert, i.e., there is at most a single trial for any given user. Since $N \geq T$ the bound of the baseline is now the vacuous regret $\Theta(T)$. We can intuitively see that this analysis is correct since the baseline algorithm is now just picking a single "uniformly at random" advertisement from $[K]$ for each user independently. However, observe that when $\Psi(y) \in \tilde{o}(\sqrt{T})$ we get $\widetilde{\mathcal{O}}(\Psi(y)\sqrt{T}) \subseteq \tilde{o}(T)$, which is non-vacuous. Intuitively, GABA-I/II may achieve this result since algorithmically they are exploiting the network structure.

## 5 Conclusion

We considered a contextual, non-stochastic bandit problem in which the finite set of contexts (a.k.a users) form a social network and the inductive bias is that if the social link between two users is strong then actions that perform well for one of these users are likely to perform well for the other. We gave two highly efficient algorithms for this problem, both with good regret bounds. Since this work is theoretical in nature we cannot foresee any potential negative societal impacts.

In the future it may be interesting to investigate extensions of our algorithms to the stochastic setting, as well as continuous bandit settings. Finally, it would be valuable to study potential applications of our algorithms, with large scale recommender systems being a natural candidate. On the theory side our bounds are based on an exponential potential function. Improved adversarial regret bounds were proven for an alternate potential function in [19] and it is an open question if our techniques can be extended to that potential.

**Acknowledgements.**  Mark Herbster was supported by the U.S. Army Research Laboratory and the U.K. Ministry of Defence under Agreement Number W911NF-16-3-0001. Stephen Pasteris and Massimiliano Pontil were supported in part by EPSRC Grant N. EP/P009069/1 and SAP SE.

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

# Technical Appendices

In the following appendices we provide proofs and additional background for all results in the body as well as an extended literature review. In the following appendix we provide a synopsis of our results and analyses as well as a brief guide to the remaining appendices.

## A   Synopsis

In this work we provided two novel algorithms GABA-I and GABA-II (see Figures 3 and 4, respectively). We gave their regret bounds in Corollaries 4 and 5. We lightly discussed their time complexities in the main body of the paper and we elaborated on that discussion in the lead-in to Propositions 24 and 38. In this section we provide a very brief overview of the supporting analysis.

---

Step 1:   Embed user graph $\mathcal{G}$ to a random spanning tree $\mathcal{T}$ (see Figure 1).

Step 2:   Embed random spanning tree $\mathcal{T}$ to a linear order $\mathcal{L}$ (see Figure 1).

Step 3:   Choose a prior $w_1(\cdot)$ for GABA-I (7) or GABA-II (9) aligning with the ordering $\mathcal{L}$.

> *Observation:* Corollaries 4 and 5 rely only on the above steps.

Step 4:   Construct from linear order $\mathcal{L}$ the binary support tree $\mathcal{C}$ (see Figure 1).

> *Observation:* The tree $\mathcal{C}$ is the "skeleton" of the data structures that underpins the algorithms GABA-I and GABA-II.

---

Figure 5: A Schematic Overview

There are two technical tools that underly the regret analysis and the fast algorithms. The first tool is a sequence of embedding steps that produce a "binary support tree" (BST) $\mathcal{C}$ and a linear embedding $\mathcal{L}$ from the user graph $\mathcal{G}$. The second tool is the SPECIALISTEXP algorithm (see Figure 2) and its regret bound in Theorem 3. We give a four step schema in Figure 5 which shows how these tools interact. The linear embedding provided by Steps 1 and 2 reduces (with approximation guarantees) the user graph to a linear ordering. This construction is used at Step 3 to map the priors in Equations (7) and (9) to the ordering. Then, using these priors with Theorem 3, the proofs of Corollaries 4 and 5 follow in conjunction with the approximation guarantees given in Lemmas 2 and 36, the latter lemma is needed for the analysis of GABA-II only. By using the priors "directly" we would have per trial prediction and update times of $\Theta(K^N)$ and $\Theta(KN)$ for the GABA-I and GABA-II models, respectively. This motivates Step 4 which gives the basic data structure, a BST, upon which more elaborate data structures are then built that enable the algorithms GABA-I/II to speed-up prediction[4] to $\mathcal{O}(K \ln N)$ and $\mathcal{O}(\ln(K) \ln(N))$ time, respectively.

For GABA-I, from a bird's eye perspective we may understand the computational motivation behind the BST as that since we have a linear ordering over $N$ users then to perform belief propagation directly it will require $\Theta(N)$ time per action per trial. However, on the BST no two users are more than $2 \log N$ vertices apart. Thus by caching partial computations "in" the BST we may "online belief propagate" in $\Theta(\log N)$ time per action per trial. For GABA-II, however, the algorithm does not resemble belief propagation, rather it may be interpreted as a simpler "message passing" algorithm where there is a natural mapping between the vertices of the BST and the specialists themselves. The proofs of our results are contained in the remaining appendices whose structure we outline below.

As background we expand on our literature review in Appendix B. In Appendix C we provide a proof to Proposition 1. We then give notational conventions that hold in the remaining appendices in Appendix D. In Appendix E we give the pseudocode and overview the analysis of the PROTOGABA algorithm. This algorithm is essentially a special case of SPECIALISTEXP. However, the notational conventions of the PROTOGABA algorithm will prove to be more natural for the data structures

---

[4]Note that the GABA-I/II algorithms do not take as explicit input the prior distributions. Rather they are prediction-equivalent to SPECIALISTEXP with the correct prior.

used in the GABA-I/II algorithms. The pseudocode and more detailed overviews of the GABA-I/II algorithms are given in Appendices F and G respectively. We then provide all (remaining) proofs in the appendices H, I and J.

## B  Expanded Literature Review

In order to give more background we expand on literature review in the the main body.

There has been much work in the multi-user setting for linear-stochastic bandits. In this setting (the pure stochastic setting being a special case) each user has a weight vector and on each trial we need to choose an action from a given finite set of context vectors which varies over trials: the loss on a trial is equal to the inner product of the selected context vector and the weight vector of the current user, plus some zero-mean random noise. Out of these works, those closest to us are when the users are in a known social network and it is assumed that neighbouring users in this network are likely to have similar weight vectors [21, 22, 23, 24, 25, 26] but as far as we are aware no works on this model have so far been done in the adversarial setting. Other works on this topic include those in which no social network is given but it is assumed that there is an unknown clustering of the users, sometimes dependent on the context, and all users in a cluster have the same weight vector or respond to the context in the same way [27, 28, 29, 30, 31, 32, 33, 34, 35]; those in which the norm of the difference of the weight vectors between any two users is bounded [36, 37], sometimes by the weight of the corresponding edge in a social network [38]; those in which the user weight vectors are equal to an unknown global vector plus a user-specific random vector [39]; those in which the user weight vectors evolve over time by linearly incorporating weights of neighbouring users [40]; those in which each user's weight vector is a linear combination of a set of unknown vectors [41]; those in which the users are unrelated but the actions have, in addition to the observed context vector, hidden features/components [42]; and those in which, when an action is selected for a user, the loss is observed for all its neighbours also [43]. Another multi-user topic is in the works [78, 79] in which users either like or dislike items (the suggestion of an item being the action) but we can't suggest the same item to a user twice: when a new item is selected its like-dislike vector (over users) is drawn uniformly at random from some probability distribution which has constraints on it. There is also a multi-user adversarial bandit problem which is a special case of the combinatorial bandit problem [80, 81, 82], where on each trial actions are selected for all users but only the total loss is observed. The above works all assume that the bandit problem for different users is different (i.e. not all users have the same weight vectors or best actions) and the algorithm is centralised: another line of research is multi-user bandit problems (both stochastic and adversarial) in which the users are collaboratively trying to solve the same problem but the algorithm is distributed in that there is limited communication between users over the social network [44, 45, 46, 47, 48, 49, 50]. Related to the multi-user setting are works on transfer learning and meta-learning with linear-stochastic bandits [51, 52].

Whilst our work assumes a known network structure over the users, there is a wide literature on bandit problems in which the actions are structured in a network and it is assumed that neighbouring actions give similar losses [53, 54, 55, 56, 57, 58, 59, 60] or that when an action is selected, the losses of its neighbouring actions are revealed [61, 62] sometimes contributing to the incurred loss [63]. Related to this is the work of [83] in which contexts and actions are vectors, with each component being of a finite set of values. In this work, each component of the context and action vector corresponds to a node of a given graph and the expected reward of a context/action pair is defined via a graphical model on the graph. There is also the work [84] in the stochastic setting where we have users and actions and there is a known metric space over all feasible user/action pairs such that the difference between the mean rewards of two user/action pairs is bounded by the distance between them.

Since, in our work, the best action varies over the users and is assumed to be similar for neighbouring users, our work is related to non-stationary linear-stochastic bandits where the user's (often a single user) weight vector varies over the trials but is assumed to be similar for neighbouring trials [85, 86, 87, 88]

We now cite the works that haven given us the mathematical tools to formulate our algorithms. Both our main algorithms GABA-I and GABA-II are efficient implementations of instances of our underlying algorithm SPECIALISTEXP which is dependent on a weighted set of subsets of users and is a combination of ideas from [18, 15, 16, 89]. Also, in order to define the weighted set of subsets

that is input to PROTOGABA , as well as performing the implementations themselves, both GABA-I and GABA-II linearize the social network as was introduced, in the machine learning context, in [13, 14]. GABA-I then utilises ideas from [16, 64] whilst GABA-II utilises ideas from [65].

## C  Proof of Proposition 1

First we recall Proposition 1 and then prove it.

**Proposition 1.** *Consider an unweighted graph $\mathcal{G}$ partitioned into $G$ clusters and a labeling function $y(\cdot)$, where each cluster is an $n$-clique and, if $u, v$ are vertices in the cluster, then $y(u) = y(v)$. For any pair of such clusters $C, C' \subset \mathcal{G}$, suppose that there are $\frac{n-1}{G}$ edges connecting the nodes of $C$ with the nodes of $C'$. Then we have $\Phi(y) \in \mathcal{O}(G)$.*

*Proof.* Let $\mathcal{C} := \{C_1, C_2, \ldots, C_G\}$ be the set of all clusters. For each inter-cluster cut-edge $\{v_i, v_j\}$, where $v_i \in C_i$ and $v_j \in C_j$, we show how to find $\Theta(n)$ edge-disjoint paths of constant length in $\mathcal{G}$ connecting $v_i$ with $v_j$. Such paths can be partitioned into $G - 2$ equally sized sets, where each of such sets contains $\frac{n-1}{G}$ paths passing through one of the clusters in $\mathcal{C} \setminus \{C_i, C_j\}$. This way, by applying the rule of resistors in parallel combined with Rayleigh's monotonicity, we have that the effective resistance between $u$ and $v$ is upper bounded by $\mathcal{O}\left(\frac{1}{n}\right)$. On the other hand, we have $\frac{n-1}{G}\binom{G}{2}$-many inter-cluster edges in $\mathcal{G}$, which gives us a total effective resistance $\Phi(y)$ bounded by $\mathcal{O}(G)$.

For the sake of simplicity, assume that there are at least 3 clusters[5]. Consider any cluster $C_k \in \mathcal{C} \setminus \{C_i, C_j\}$. Let $V_{i,k}$ and $V_{j,k}$ be the subset of nodes of $C_k$ that are incident to edges connecting $C_k$ with $C_i$ and $C_k$ with $C_j$ respectively. For each node $u \in V_{i,k} \cup V_{j,k}$, let $d_u^i$ and $d_u^j$ be the number of edges connecting $u$ with the nodes of $C_i$ and the nodes of $C_j$ respectively.

We first show that we can connect the nodes of $V_{i,k}$ with the nodes of $V_{j,k}$ through $\frac{n-1}{G}$ edge disjoint-paths of length at most 2 lying all within $C_k$ [6]. Each node $u \in V_{i,k}$ is connected with the nodes $V_{j,k}$ through $d_u^i$-many edge-disjoint paths of length *at most* 2 within $C_k$. Symmetrically, these edge-disjoint paths connect the nodes of $V_{i,k}$ with the nodes of $V_{j,k}$ in such a way that each node $v \in V_{j,k}$ is a terminal node of $d_u^j$-many of such edge-disjoint paths.

In the simplest case scenario, if $|V_{i,k}| = \frac{n-1}{G}$, $|V_{i,j}| = \frac{n-1}{G}$ and $V_{i,k} \cap V_{j,k} = \emptyset$, then we can immediately find $\frac{n-1}{G}$ edges (edge disjoint paths of length 1) lying within $C_k$. If $V_{i,k} \cap V_{j,k} \neq \emptyset$, we can create $\min(d_u^i, d_u^j)$-many paths of length 0 for each node $u \in V_{i,k} \cap V_{j,k}$. If we have instead that $|V_{i,k}| < \frac{n-1}{G}$ or $|V_{i,j}| < \frac{n-1}{G}$, then we can always exploit the clique structure of the cluster to create new paths of length 2 lying within $C_k$. To see how, consider for instance the extreme case where $|V_{i,k}| = 1$, $|V_{j,k}| = 1$ and $V_{i,k} \cap V_{j,k} = \emptyset$. The clique structure ensures that we can connect the nodes in $V_{i,k}$ and $V_{j,k}$ through 1 edge plus $\frac{n-1}{G} - 1$ edge-disjoint paths of length 2 all within $C_k$. For the paths of length 2 we use $\frac{n-1}{G} - 1$ *other* nodes of $C_k$ that are not terminal nodes of such paths (i.e., they are the nodes in the middle of these paths). This possibility is guaranteed by the fact $n \geq 1 + \frac{n-1}{G} = 2 + \frac{n-1}{G} - 1$. Finally, if $|V_{i,k}| < \frac{n-1}{G}$ or $|V_{i,j}| < \frac{n-1}{G}$ and $V_{i,k} \cap V_{j,k} \neq \emptyset$, then, as we mentioned above, we can exploit each node $u \in V_{i,k} \cap V_{j,k}$ to create $\min(d_u^i, d_u^j)$-many paths of length 0, which makes the problem easier. Notice that, whenever $V_{i,k} \cap V_{j,k} \neq \emptyset$, the problem can be reduced to finding just $\left(\frac{n-1}{G} - \sum_{u \in V_{i,k} \cap V_{j,k}} \min(d_u^i, d_u^j)\right)$-many edge-disjoint paths within $C_k$, connecting the nodes in $V_{i,k}$ and $V_{j,k}$

Now we show how to connect $v_i$ (resp. $v_j$) with the nodes in $V_{i,k}$ (resp. $V_{j,k}$) by $\frac{n-1}{G}$ edge-disjoint paths of length at most three, for each $k$ in turn (so that all paths are edge-disjoint). Without loss of generality, we will focus on $C_i$ and $v_i$. We proceed incrementally by connecting one by one each node $u \in V_{i,k}$ (for all $k \in [G] \setminus \{i, j\}$ in turn) to $v_i$ through $d_u^i$ paths via the following algorithm. For all edges $\{w, u\}$ where $w \in C_i$ we create a path as follows:

---

[5]If there are only 2 clusters, then it can be shown that $\Phi(y) \in \mathcal{O}(1)$ by using a very similar argument.

[6]In the special case a node of $C_k$ is incident to an edge connecting $C_i$ with $C_k$, and one connecting $C_k$ with $C_j$, then we view it by convenience as a node of a path of length 0 belonging to $C_k$. Clearly such special path is edge-disjoint from any other path lying in $C_k$ because it does not have any edge.

- If $w = v_i$ then our path is $(v_i, u)$.

- If $w \neq v_i$ choose a node $v \in C_i \setminus \{v_i\}$ which hasn't been a "middle node" of any path so far. We have two cases:

    - If $w = v$ then our path is $(v_i, v, u)$
    - If $w \neq v$ then our path is $(v_i, v, w, u)$

  The node $v$ is called the "middle node" of our path, so can't be selected as the middle node of any future path.

It is easy to verify that all the paths formed exist and are edge disjoint, and that we have $n - 1$ possibilities for middle nodes. Note that we need to use at most $\frac{n-1}{G}$ middle nodes for each cluster $C_k \in \mathcal{C} \setminus \{C_i, C_j\}$. Since the total number of clusters is $G$, we then need to have $(G-2)\frac{n-1}{G} \leq n-1$, which is always true.

Thus we can ensure that, for all inter-cluster edges $\{v_i, v_j\}$ where $v_i \in V_i$ and $v_j \in V_j$, there exist $(G - 2)\frac{n-1}{G} = \Theta(n)$ edge-disjoint paths formed by, for all $k$, concatenating the edge-disjoint paths from node $v_i$ into $V_{i,k}$, the edge-disjoint paths lying within $C_k$, and finally the edge-disjoint paths from node $v_j$ into $V_{j,k}$. The total length of each of such paths cannot therefore exceed $3 + 2 + 3 = 8$, which concludes the proof. $\qquad\square$

## D   Conventions

In the following we present some conventions which lighten the notation of the main body as well as introduce the subroutine structure shared by the three algorithms PROTOGABA and GABA-I/II.

In order to make our pseudocode clearer, the learning algorithms we present each have three subroutines: **Initialization**, **Prediction**$(\cdot)$, which returns an element of $[K]$, and **Update**$(\cdot)$. The learning protocol (i.e. the sequence of subroutine calls) is given in Figure 6.

---

**Learning Protocol**

  1. **Initialization**
  2. For all $t \in [T]$ in order:
      (a) $a_t \leftarrow \mathbf{Prediction}_t(u_t)$
      (b) $\mathbf{Update}_t(\ell_{t,a_t})$

where $\{u_t \mid t \in [T]\} \subseteq [N]$ and $\{\boldsymbol{\ell}_t \mid t \in [T]\} \subseteq [0,1]^K$ are arbitrary, fixed a-priori, and unknown to the algorithm.

---

Figure 6: Learning Algorithm Protocol

We recall that $y$ is an arbitrary function from the users $[N]$ to the actions $[K]$. We abbreviate the Robustified Resistance Weighted Cutsize (see (5)) $\Psi(y)$ to $\Psi$ and we recall the expected regret notation (see (3)),

$$R(y) := \mathbb{E}\left[\sum_{t \in [T]} (\ell_{t,a_t} - \ell_{t,y(u_t)})\right],$$

which we abbreviate to $R$.

## E   SPECIALISTEXP and PROTOGABA

We first introduce the algorithm PROTOGABA of which the GABA-I/II algorithms implement instances of. PROTOGABA takes as input a set $\mathcal{E}$, of subsets of $[N]$, and a probability distribution $\sigma$ over $\mathcal{E}$, and runs SPECIALISTEXP with initial weights defined as follows:

- Given $U \in \mathcal{E}$ and $a \in [K]$, the specialist that predicts $a$ whenever $u_t \in U$ and abstains otherwise is given initial weight $\sigma(U)/K$.

The PROTOGABA algorithm takes a parameter $\eta \in \mathbb{R}^+$ and a set $\mathcal{E} \subseteq 2^{[N]}$ along with a function $\sigma : \mathcal{E} \to \mathbb{R}^+$ satisfying:

$$\sum_{U \in \mathcal{E}} \sigma(U) = 1$$

**Initialization** :

      1. For all $U \in \mathcal{E}$ and all $a \in [K]$:

         (a) $\kappa_1(a, U) \leftarrow \sigma(U)/K$.

**Prediction**$_t(u_t)$ :

      1. For all $a \in [K]$ set $p_{t,a} \leftarrow \sum_{U \in \mathcal{E}: u_t \in U} \kappa_t(a, U)$.

      2. Draw $a_t$ from $[K]$ with probability $\mathbb{P}[a_t = a] \propto p_{t,a}$.

      3. Return $a_t$.

**Update**$_t(\ell_{t,a_t})$ :

      1. $\lambda_t \leftarrow \exp(-\eta \ell_{t,a_t} \|\boldsymbol{p}_t\|_1 / p_{t,a_t})$.

      2. $z_t \leftarrow \|\boldsymbol{p}_t\|_1 / (\|\boldsymbol{p}_t\|_1 - (1 - \lambda_t)p_{t,a_t})$.

      3. For all $U \in \mathcal{E}$ and all $a \in [K]$:

         (a) If $u_t \notin U$ then $\kappa_{t+1}(a, U) := \kappa_t(a, U)$.

         (b) If $u_t \in U$:

            i. If $a \neq a_t$ then $\kappa_{t+1}(a, U) \leftarrow z_t \kappa_t(a, U)$.

            ii. If $a = a_t$ then $\kappa_{t+1}(a, U) \leftarrow z_t \lambda_t \kappa_t(a, U)$.

Figure 7: PROTOGABA Subroutines

- If there does not exist such a $U$ and $a$ then the specialist has initial weight $0$.

The subroutines of PROTOGABA are given in Figure 7. We shall now bound the regret of PROTO-GABA, noting that all results are proved (in order) in Appendix H. Note that Theorem 3 is proved in exactly the same way but replacing $\mathcal{A}$ by $\mathcal{S}$ and replacing the set $[K] \times \mathcal{E}$ by the non-zero weight specialists.

The regret bound of PROTOGABA depends on any set $\mathcal{A} \subseteq [K] \times \mathcal{E}$ satisfying the following conditions:

**Definition 6.** *Take an arbitrary set $\mathcal{A} \subseteq [K] \times \mathcal{E}$ such that:*

- *For all $u \in [N]$ there exists a unique pair $(a, U) \in \mathcal{A}$ with $u \in U$.*

- *For all $u \in [N]$ and $(a, U) \in \mathcal{A}$ with $u \in U$, we have that $a = y(u)$.*

We begin our analysis by bounding the expected "progress" on each trial:

**Lemma 7.** *For all $t \in [T]$ we have:*

$$\mathbb{E}\left[ \sum_{(a,U) \in \mathcal{A}} \ln\left( \frac{\kappa_{t+1}(a, U)}{\kappa_t(a, U)} \right) \right] \geq \eta \left( \mathbb{E}[\ell_{t,a_t}] - \ell_{t,y(u_t)} \right) - \frac{K\eta^2}{2}.$$

We will utilise the following inequality:

**Lemma 8.** *We have:*

$$\sum_{(a,U) \in \mathcal{A}} \ln\left( |\mathcal{A}| \kappa_{T+1}(a, U) \right) \leq 0.$$

Lemmas 7 and 8 lead to the following bound on the expected regret:

**Theorem 9.** *We have:*

$$R \leq \frac{1}{\eta} \sum_{(a,U) \in \mathcal{A}} \ln \left( \frac{K}{|\mathcal{A}|\sigma(U)} \right) + \frac{\eta K T}{2}.$$

## F  GABA-I

We now introduce and analyse our algorithm GABA-I which has a per trial time-complexity of $\mathcal{O}(K \ln(N))$, a space complexity of $\mathcal{O}(KN)$, and a regret bound of $\mathcal{O}\left( \sqrt{\ln\left(\frac{KN}{\Psi}\right) \Psi K T} \right)$. The subroutines of GABA-I are given in Figure 8. We shall now outline the proof of the bound the on regret of GABA-I, noting that all results are proved (in order) in Appendix I. For simplicity assume, without loss of generality, that for all $u \in [N]$ we have that $u$ is the $u$-th leftmost leaf of $\mathcal{C}$.

We will show that GABA-I implements PROTOGABA with the following choice of $\mathcal{E}$ and $\sigma$:

**Definition 10.** *For all $i, j \in \{0, 1\}$ we define:*

$$\iota_j := \frac{1}{K}[\![j = 1]\!] + \left( 1 - \frac{1}{K} \right) [\![j = 0]\!]$$

*and:*

$$\tau_{i,j} := \phi[\![i \neq j]\!] + (1 - \phi)[\![i = j]\!].$$

*For* GABA-I *we set:*

$$\mathcal{E} := 2^{[N]}$$

*and for all $U \in \mathcal{E}$ we set:*

$$\sigma(U) := \iota_{[\![1 \in U]\!]} \prod_{u \in [N-1]} \tau_{[\![u \in U]\!],[\![u+1 \in U]\!]}.$$

The following lemma states that, as required, $\sigma$ is a probability distribution over $\mathcal{E}$:

**Lemma 11.** *We have that $\sigma$ is a probability distribution, in that:*

$$\sum_{U \in \mathcal{E}} \sigma(U) = 1.$$

We will show that the function $\kappa_t$, constructed in PROTOGABA, is equal to the function $\bar{\kappa}_t$ that is defined as follows:

**Definition 12.** *For all $t \in [T+1]$ we define a function $\bar{\kappa}_t : [K] \times \mathcal{E} \to \mathbb{R}^+$ inductively as follows. For all $(a, U) \in [K] \times \mathcal{E}$:*

- $\bar{\kappa}_1(a, U) := \sigma(U)/K$                                  `

- For all $t \in [T]$   $\bar{\kappa}_{t+1}(a, U) = \begin{cases} \bar{\kappa}_t(a, U) & \text{if } u_t \notin U \\ \pi_{t,a}\bar{\kappa}_t(a, U) & \text{otherwise.} \end{cases}$

To quantify the vector valued functions $\boldsymbol{\alpha}_t$, $\boldsymbol{\beta}_t^{\Leftarrow}$ and $\boldsymbol{\beta}_t^{\Rightarrow}$ we need the following definitions:

**Definition 13.** *For all $t \in [T]$, $u \in [N]$ and $a \in [K]$ define:*

$$\xi_{t,u,a} := \prod_{s \in [t-1]:u_s = u} \pi_{s,a}.$$

**Definition 14.** *Given $u, v \in [N+1]$ with $u \leq v$ let $I(u, v)$ be the set of functions that map the set $\{w \in \mathbb{N} \mid u \leq w \leq v\}$ to $\{0, 1\}$.*

**Definition 15.** *Given $f \in I(u, v)$ for some $u, v \in [N+1]$ with $u \leq v$, and some $a \in [K]$, let:*

$$\Omega_{t,a}(f) := \prod_{w \in \mathbb{N}:u \leq w < v} \tau_{f(w),f(w+1)}([\![f(w) = 0]\!] + [\![f(w) = 1]\!]\xi_{t,w,a}).$$

**GABA-I Subroutines**

GABA-I takes parameters $\eta \in \mathbb{R}^+$ and $\phi \in [0, 1]$.

**Initialization** :

1. Construct BST $\mathcal{C}$ as in Section 2.2.
2. For all leaves $n \in \mathcal{C}$:
   (a) $\forall i, j \in \{0, 1\}, \boldsymbol{\alpha}_1(n, i, j) \leftarrow [\![i \neq j]\!]\phi\mathbf{1} + [\![i = j]\!](1 - \phi)\mathbf{1}$.
3. For all $d \in [h - 1]$ in order:
   (a) For all vertices $n \in \mathcal{C}$ at depth $h - d$:
      i. $\forall i, j \in \{0, 1\}, \boldsymbol{\alpha}_1(n, i, j) \leftarrow \sum_{k \in \{0,1\}} \boldsymbol{\alpha}_1(\triangleleft(n), i, k) \odot \boldsymbol{\alpha}_1(\triangleright(n), k, j)$.

**Prediction**$_t(u_t)$ :

1. For all $d \in [h] \cup \{0\}$ let $\nu_{t,d}$ be the ancestor (in $\mathcal{C}$) of $u_t$ at depth $d$.
2. $\forall i \in \{0, 1\}, \boldsymbol{\beta}_t^{\Leftarrow}(\nu_{t,0}, i) \leftarrow (1 + [\![i = 0]\!](K - 2))\mathbf{1}/K$.
3. $\forall i \in \{0, 1\}, \boldsymbol{\beta}_t^{\Rightarrow}(\nu_{t,0}, i) \leftarrow \mathbf{1}$.
4. For all $d \in [h]$ in order:
   (a) If $\nu_{t,d} = \triangleleft(\nu_{t,d-1})$ then:
      i. $\forall i \in \{0, 1\}, \boldsymbol{\beta}_t^{\Leftarrow}(\nu_{t,d}, i) \leftarrow \boldsymbol{\beta}_t^{\Leftarrow}(\nu_{t,d-1}, i)$.
      ii. $\forall i \in \{0, 1\}, \boldsymbol{\beta}_t^{\Rightarrow}(\nu_{t,d}, i) \leftarrow \sum_{j \in \{0,1\}} \boldsymbol{\alpha}_t(\triangleright(\nu_{t,d-1}), i, j) \odot \boldsymbol{\beta}_t^{\Rightarrow}(\nu_{t,d-1}, j)$.
   (b) If $\nu_{t,d} = \triangleright(\nu_{t,d-1})$ then:
      i. $\forall i \in \{0, 1\}, \boldsymbol{\beta}_t^{\Rightarrow}(\nu_{t,d}, i) \leftarrow \boldsymbol{\beta}_t^{\Rightarrow}(\nu_{t,d-1}, i)$.
      ii. $\forall i \in \{0, 1\}, \boldsymbol{\beta}_t^{\Leftarrow}(\nu_{t,d}, i) \leftarrow \sum_{j \in \{0,1\}} \boldsymbol{\beta}_t^{\Leftarrow}(\nu_{t,d-1}, j) \odot \boldsymbol{\alpha}_t(\triangleleft(\nu_{t,d-1}), j, i)$.
5. $\bar{\boldsymbol{p}}_t \leftarrow (1/K) \sum_{i \in \{0,1\}} \boldsymbol{\beta}_t^{\Leftarrow}(\nu_{t,h}, 1) \odot \boldsymbol{\alpha}_t(\nu_{t,h}, 1, i) \odot \boldsymbol{\beta}_t^{\Rightarrow}(\nu_{t,h}, i)$.
6. Draw $a_t$ from $[K]$ with probability $\mathbb{P}[a_t = a] \propto \bar{p}_{t,a}$.
7. Return $a_t$.

**Update**$_t(\ell_{t,a_t})$ :

1. $\forall a \in [K], c_{t,a} \leftarrow \exp\left(-\eta[\![a = a_t]\!]\ell_{t,a_t}\|\bar{\boldsymbol{p}}_t\|_1/\bar{p}_{t,a}\right)$.
2. $\boldsymbol{\pi}^t \leftarrow (\|\bar{\boldsymbol{p}}_t\|_1 \boldsymbol{c}_t)/(\bar{\boldsymbol{p}}_t \cdot \boldsymbol{c}_t)$.
3. $\forall i \in \{0, 1\}, \boldsymbol{\alpha}_{t+1}(\nu_{t,h}, 1, i) \leftarrow \boldsymbol{\pi}^t \odot \boldsymbol{\alpha}_t(\nu_{t,h}, 1, i)$.
4. $\forall i \in \{0, 1\}, \boldsymbol{\alpha}_{t+1}(\nu_{t,h}, 0, i) := \boldsymbol{\alpha}_t(\nu_{t,h}, 0, i)$.
5. For all $n \in \mathcal{C} \setminus \{\nu_{t,d} \mid d \in [h] \cup \{0\}\}$:
   (a) $\forall i, j \in \{0, 1\}, \boldsymbol{\alpha}_{t+1}(n, i, j) := \boldsymbol{\alpha}_t(n, i, j)$.
6. For all $d \in [h - 1]$ in order:
   (a) $\forall i, j \in \{0, 1\}$ set $\boldsymbol{\alpha}_{t+1}(\nu_{t,(h-d)}, i, j) \leftarrow \sum_{k \in \{0,1\}} \boldsymbol{\alpha}_{t+1}(\triangleleft(\nu_{t,(h-d)}), i, k) \odot \boldsymbol{\alpha}_{t+1}(\triangleright(\nu_{t,(h-d)}), k, j)$.

Figure 8: GABA-I Subroutines

To prove lemmas 17 and 18 we will need the following lemma:

**Lemma 16.** *Given $i, j, k \in \{0, 1\}$, $u, v, w \in [N]$ with $u < v < w$, and $a \in [K]$ we have:*

$$\left(\sum_{f \in F} \Omega_{t,a}(f)\right)\left(\sum_{f \in G} \Omega_{t,a}(f)\right) = \sum_{f \in H} \Omega_{t,a}(f)$$

*where:*

- $F := \{f \in I(u, v) \mid f(u) = i, f(v) = j\}$
- $G := \{f \in I(v, w) \mid f(v) = j, f(w) = k\}$
- $H := \{f \in I(u, w) \mid f(u) = i, f(v) = j, f(w) = k\}.$

We now quantify the vector valued function $\boldsymbol{\alpha}_t$:

**Lemma 17.** *For all $t \in [T] \cup \{0\}$, all non-root vertices $n$ of $\mathcal{C}$, all $i, j \in \{0, 1\}$ and all $a \in [K]$ we have:*
$$\alpha_t(n, i, j)_a = \sum_{f \in A_t(n,i,j)} \Omega_{t,a}(f)$$

*where:*

$$A_t(n, i, j) := \{f \in I(\blacktriangleleft(n), \blacktriangleright(n) + 1) \mid f(\blacktriangleleft(n)) = i, f(\blacktriangleright(n) + 1) = j\}.$$

Lemma 17 now allows us to quantify the vector valued functions $\boldsymbol{\beta}_t^{\Leftarrow}$ and $\boldsymbol{\beta}_t^{\Rightarrow}$. We note that the following lemma is proved via induction over $d$.

**Lemma 18.** *For all $t \in [T]$, $d \in [h] \cup \{0\}$, $i \in \{0, 1\}$ and $a \in [K]$ we have:*

$$\beta_t^{\Rightarrow}(\nu_{t,d}, i)_a = \sum_{f \in B_t^{\Rightarrow}(\nu_{t,d},i)} \Omega_{t,a}(f)$$

*where:*

$$B_t^{\Rightarrow}(\nu_{t,d}, i) := \{f \in I(\blacktriangleright(\nu_{t,d}) + 1, N + 1) \mid f(\blacktriangleright(\nu_{t,d}) + 1) = i\}$$

*and:*

$$\beta_t^{\Leftarrow}(\nu_{t,d}, i)_a = \sum_{f \in B_t^{\Leftarrow}(\nu_{t,d},i)} \iota_{f(1)} \Omega_{t,a}(f)$$

*where:*

$$B_t^{\Leftarrow}(\nu_{t,d}, i) := \{f \in I(1, \blacktriangleleft(\nu_{t,d})) \mid f(\blacktriangleleft(\nu_{t,d})) = i\}.$$

Lemmas 17 and 18 allow us to write the vector $\bar{p}_t$ in terms of the function $\bar{\kappa}_t$:

**Lemma 19.** *For all $t \in [T]$ and $a \in [K]$ we have:*

$$\bar{p}_{t,a} = \sum_{U \in \mathcal{E}: u_t \in U} \bar{\kappa}_t(a, U)$$

Lemma 19 implies that GABA-I does indeed implement PROTOGABA:

**Lemma 20.** GABA-I *implements* PROTOGABA *with $\mathcal{E}$ and $\sigma$ defined as in Definition 10.*

We choose, as required in the analysis of PROTOGABA, the set $\mathcal{A}$ to be as follows:

**Definition 21.** *We define:*

$$\mathcal{A} := \{(a, \{u \in [N] \mid y(u) = a\}) \mid a \in [K]\}.$$

The following lemma states that $\mathcal{A}$ is valid:

**Lemma 22.** *We have that:*

- *For all $u \in [N]$ there exists a unique pair $(a, U) \in \mathcal{A}$ with $u \in U$.*
- *For all $u \in [N]$ and $(a, U) \in \mathcal{A}$ with $u \in U$, we have that $a = y(u)$.*

Lemmas 20 and 22 allow us to invoke Theorem 9. Noting that $|\mathcal{A}| = K$, Lemma 2 allows us to bound the expectation of the summation of $\ln(1/\sigma(a, U))$ appearing in Theorem 9, leading to our main result:

**Theorem 23.** *Given $\Psi \leq (N-1)/4$ and the parameters are tuned as:*

$$\phi := 4\Psi/(K(N-1))$$

*and:*

$$\eta := \sqrt{\frac{10\Psi \ln(KN/\Psi)}{KT}}$$

*we have:*

$$R \in \mathcal{O}\left(\sqrt{\ln\left(\frac{KN}{\Psi}\right) \Psi K T}\right).$$

We now argue the per-trial time complexity of GABA-I. In the prediction algorithm each of the $\mathcal{O}(\ln(N))$ proper ancestors $n$ of $u_t$ constructs four vector valued messages $\boldsymbol{\beta}_t^{\Leftarrow}(n, 0), \boldsymbol{\beta}_t^{\Leftarrow}(n, 1), \boldsymbol{\beta}_t^{\Rightarrow}(n, 0), \boldsymbol{\beta}_t^{\Rightarrow}(n, 1)$, each taking a time of $\mathcal{O}(K)$ to construct. In the update algorithm the values $\boldsymbol{\alpha}_t(n, i, j)$ are updated to $\boldsymbol{\alpha}_{t+1}(n, i, j)$, only being modified when $n$ is one of the $\mathcal{O}(\ln(N))$ ancestors of $u_t$. Since each such modification takes a time of $\mathcal{O}(K)$ we then have the following proposition:

**Proposition 24.** GABA-I *takes a per-trial time of $\mathcal{O}(K \ln(N))$.*

# G  GABA-II

We now introduce and analyse our algorithm GABA-II which has a per trial time-complexity of $\mathcal{O}(\ln(K)\ln(N))$, a space complexity of $\mathcal{O}(KN)$, and a regret bound of $\mathcal{O}\left(\sqrt{\ln\left(\frac{N}{\Psi}\right)\ln\left(\frac{KN}{\Psi}\right)\Psi K T}\right)$. The subroutines of GABA-II are given in Figure 9. We shall now outline the proof of the bound on the regret of GABA-II, noting that all results are proved (in order) in Appendix J. For simplicity assume, without loss of generality, that for all $u \in [N]$ we have that $u$ is the $u$-th leftmost leaf of $\mathcal{C}$.

We will show that GABA-II implements PROTOGABA with the following choice of $\mathcal{E}$ and $\sigma$:

**Definition 25.** *For GABA-II we set:*

$$\mathcal{E} := \{\Downarrow(n) \mid n \in \mathcal{C}\}$$

*and for all $U \in \mathcal{E}$ we set:*

$$\sigma(U) := \frac{1}{2N-1}.$$

Note that as $|\mathcal{E}| = |\mathcal{C}| = 2N - 1$ it is clear that $\sigma$ is a probability distribution over $\mathcal{E}$, as required.

We will show that, for all $n \in \mathcal{C}$ and $a \in [K]$, the value $\kappa_t(a, \Downarrow(n))$, constructed in PROTOGABA, is equal to the value $\bar{\kappa}_t(a, n)$ that is defined as follows:

**Definition 26.** *For all $t \in [T]$ we define:*

$$\bar{z}_t := \psi_t/(\psi_t - (1 - \bar{\lambda}_t)\varrho_t).$$

**Definition 27.** *We define the functions $\bar{\kappa}_t : [K] \times \mathcal{C} \to [0, 1]$ inductively as follows. For all $(a, n) \in [K] \times \mathcal{C}$:*

- $\bar{\kappa}_1(a, n) := 1/K(2N-1)$

- *For all $t \in [T]$:*

  - *If $n \notin \Uparrow(u_t)$ then $\bar{\kappa}_{t+1}(a, n) = \bar{\kappa}_t(a, n)$*
  - *If $n \in \Uparrow(u_t)$ and $a \neq a_t$ then $\bar{\kappa}_{t+1}(a, n) := \bar{z}_t \bar{\kappa}_t(a, n)$*
  - *If $n \in \Uparrow(u_t)$ and $a = a_t$ then $\bar{\kappa}_{t+1}(a, n) := \bar{\lambda}_t \bar{z}_t \bar{\kappa}_t(a, n).$*

We will define a vector $\bar{\boldsymbol{p}}_t$ from $\bar{\kappa}_t$ in the same way that $\boldsymbol{p}_t$ is defined from $\kappa_t$ in PROTOGABA:

**GABA-II Subroutines**

GABA-II takes a parameter $\eta \in \mathbb{R}^+$.

**Intialization** :

1. Construct BST $\mathcal{C}$ as in Section 2.2 .

2. Construct a full, balanced, oriented binary tree $\mathcal{B}$ of depth $g := \lceil \log_2(K) \rceil$ whose first $K$ leaves are the actions $[K]$.

3. Set $r$ to be the root of $\mathcal{B}$.

4. For all vertices $n \in \mathcal{C}$:

   (a) $\mu_1(n) \leftarrow 1$.

   (b) For all leaves $m \in \mathcal{B}$:

      i. If $m \in [K]$ then $\theta_1(n, m) \leftarrow 1$.
      
      ii. If $m \notin [K]$ then $\theta_1(n, m) \leftarrow 0$.

   (c) For all $d \in [g]$ in order:

      i. For all vertices $m \in \mathcal{B}$ at depth $g - d$ set $\theta_1(n, m) \leftarrow \theta_1(n, \triangleleft(m)) + \theta_1(n, \triangleright(m))$.

**Prediction$_t(u_t)$** :

1. Draw $\delta_t$ from $\Uparrow(u_t)$ with probability $\mathbb{P}[\delta_t = n] \propto \mu_t(n)\theta_t(n, r)$.

2. $\zeta_{t,0} \leftarrow r$.

3. For all $d \in [g - 1] \cup \{0\}$ in order:

   (a) Draw $\zeta_{t,d+1}$ from $\{\triangleleft(\zeta_{t,d}), \triangleright(\zeta_{t,d})\}$ with probability $\mathbb{P}[\zeta_{t,d+1} = m] \propto \theta_t(\delta_t, m)$.

4. $a_t \leftarrow \zeta_{t,g}$.

5. Return $a_t$.

**Update$_t(\ell_{t,a_t})$** :

1. $\psi_t \leftarrow \sum_{n \in \Uparrow(u_t)} \mu_t(n)\theta(n, r)$.

2. $\varrho_t \leftarrow \sum_{n \in \Uparrow(u_t)} \mu_t(n)\theta(n, a_t)$.

3. $\bar{\lambda}_t \leftarrow \exp(-\eta \ell_{t,a_t} \psi_t / \varrho_t)$.

4. For all $n \in \Uparrow(u_t)$:

   (a) $\mu_{t+1}(n) \leftarrow \mu_t(n)\psi_t / (\psi_t - (1 - \bar{\lambda}_t)\varrho_t)$.

   (b) $\theta_{t+1}(n, a_t) \leftarrow \bar{\lambda}_t \theta_t(n, a_t)$.

   (c) For all $m \in \mathcal{B} \setminus \Uparrow(a_t)$:

      i. $\theta_{t+1}(n, m) := \theta_t(n, m)$.

   (d) For all $d \in [g]$ in order:

      i. $\theta_{t+1}(n, \zeta_{t,(g-d)}) \leftarrow \theta_{t+1}(n, \triangleleft(\zeta_{t,(g-d)})) + \theta_{t+1}(n, \triangleright(\zeta_{t,(g-d)}))$.

5. For all $n \in \mathcal{C} \setminus \Uparrow(u_t)$:

   (a) $\mu_{t+1}(n) := \mu_t(n)$.

   (b) $\forall m \in \mathcal{B}, \theta_{t+1}(n, m) := \theta_t(n, m)$.

Figure 9: GABA-II Subroutines

**Definition 28.** *For all $t \in [T]$ and $a \in [K]$ we define:*

$$\bar{p}_{t,a} := \sum_{n \in \mathcal{C}: n \in \Uparrow(u_t)} \bar{\kappa}_t(a, n).$$

The following lemma states that GABA-II maintains a factorisation of the values $\bar{\kappa}_t(a, n)$:

**Lemma 29.** *Given $t \in [T]$, $n \in \mathcal{C}$ and $a \in [K]$ we have:*

$$\bar{\kappa}_t(a, n) = \frac{\mu_t(n)\theta_t(n, a)}{(2N - 1)K}.$$

The following lemma comes from how we update the values $\theta(n, m)$:

**Lemma 30.** *Given $t \in [T]$, $n \in \mathcal{C}$ and $m \in \mathcal{B}$ we have:*

$$\theta_t(n, m) = \sum_{a \in \Downarrow(m) \cap [K]} \theta_t(n, a).$$

Lemmas 29 and 30 then allow us to quantify the probability distribution that $a_t$ is selected from:

**Lemma 31.** *For all $t \in [T]$ and $a \in [K]$ we have:*

$$\mathbb{P}\left[a_t = a\right] = \frac{\bar{p}_{t,a}}{\|\bar{\boldsymbol{p}}_t\|_1}.$$

Lemmas 29 and 30 also allow us to quantify $\bar{\lambda}_t$ and $\bar{z}_t$ in terms of $\bar{\boldsymbol{p}}_t$:

**Lemma 32.** *For all $t \in [T]$ we have:*

$$\bar{\lambda}_t = \exp\left(\frac{-\eta\ell_{t,a_t}\|\bar{\boldsymbol{p}}_t\|_1}{\bar{p}_{t,a_t}}\right)$$

*and:*

$$\bar{z}_t = \frac{\|\bar{\boldsymbol{p}}_t\|_1}{\|\bar{\boldsymbol{p}}_t\|_1 - (1 - \bar{\lambda}_t)\bar{p}_{t,a_t}}.$$

Lemmas 31 and 32 imply that GABA-II does indeed implement PROTOGABA:

**Lemma 33.** GABA-II *implements* PROTOGABA *with $\mathcal{E}$ and $\sigma$ defined as in Definition 25*

We choose, as required in the analysis of PROTOGABA, the set $\mathcal{A}$ to be as follows:

**Definition 34.** *Let $\mathcal{A}^\dagger$ be the set of all $(a, n) \in [K] \times \mathcal{C}$ such that:*

- *For all $u \in \Downarrow(n)$ we have $y(u) = a$.*

- *$n$ is the root of $\mathcal{C}$ or there exists $v \in \Downarrow(\uparrow(n))$ with $y(v) \neq a$.*

*Then define:*

$$\mathcal{A} = \{(a, \Downarrow(n)) \mid (a, n) \in \mathcal{A}^\dagger\}.$$

The following lemma states that $\mathcal{A}$ is valid:

**Lemma 35.** *We have that:*

- *For all $u \in [N]$ there exists a unique pair $(a, U) \in \mathcal{A}$ with $u \in U$.*

- *For all $u \in [N]$ and $(a, U) \in \mathcal{A}$ with $u \in U$, we have that $a = y(u)$.*

Lemma 2 allows us to bound the expected cardinality of $\mathcal{A}$:

**Lemma 36.** *We have:*

$$\mathbb{E}\left[|\mathcal{A}|\right] \leq 4\Psi \log_2\left(\frac{eN}{\Psi}\right).$$

Lemmas 33 and 35 allow us to invoke Theorem 9 which, combined with Lemma 36 gives us our final result:

**Theorem 37.** *Given $\Psi \leq N/2$ and setting:*

$$\eta := \sqrt{\frac{8\Psi \log_2 (eN/\Psi) \ln (3KN/2\Psi)}{KT}}$$

*we have:*

$$R \in \mathcal{O}\left(\sqrt{\ln\left(\frac{N}{\Psi}\right) \ln\left(\frac{KN}{\Psi}\right) \Psi KT}\right).$$

We now argue the per trial time complexity of GABA-II. In the prediction algorithm $\delta_t$ is first sampled from the ancestors of $u_t$, taking a time of $\mathcal{O}(\ln(N))$. Then, in selecting $a_t$, a path of length $\mathcal{O}(\ln(K))$ in $\mathcal{B}$ is sampled, with each vertex taking a time of $\mathcal{O}(1)$ to sample. In the update algorithm $\psi_t$ and $\varrho_t$ take a time of $\mathcal{O}(\ln(N))$ to compute and then the values $\mu_t(n)$ and $\theta_t(n,m)$ are only modified if $n$ is one of the $\mathcal{O}(\ln(N))$ ancestors of $u_t$ and $m$ is one of the $\mathcal{O}(\ln(K))$ ancestors of $a_t$, with each update taking $\mathcal{O}(1)$ time. This gives us the following proposition:

**Proposition 38.** GABA-II *takes a per-trial time of $\mathcal{O}(\ln(K)\ln(N))$.*

## H  PROTOGABA Proofs

### H.1  Proof of Lemma 7

**Result**. For all $t \in [T]$ we have:

$$\mathbb{E}\left[\sum_{(a,U)\in\mathcal{A}} \ln\left(\frac{\kappa_{t+1}(a,U)}{\kappa_t(a,U)}\right)\right] \geq \eta\left(\mathbb{E}\left[\ell_{t,a_t}\right] - \ell_{t,y(u_t)}\right) - \frac{K\eta^2}{2}.$$

**Proof:**

**Definition 39.** *Let $\tilde{\boldsymbol{p}}_t := \boldsymbol{p}_t/\|\boldsymbol{p}_t\|_1$ and for all $a \in [K]$ let $c_{t,a} := [\![a_t = a]\!]\lambda_t + [\![a_t \neq a]\!]$*

**Lemma 40.** *We have:*

$$\sum_{(a,U)\in\mathcal{A}} \ln\left(\frac{\kappa_{t+1}(a,U)}{\kappa_t(a,U)}\right) = \ln\left(\frac{c_{t,y(u_t)}}{\boldsymbol{c}_t \cdot \tilde{\boldsymbol{p}}_t}\right).$$

*Proof.* Let $(y(u_t), U')$ be the unique pair $(a,U) \in \mathcal{A}$ with $u_t \in U$. For all $(a,U) \in \mathcal{A}\backslash\{(y(u_t), U')\}$ we have $u_t \notin U$ so $\kappa_{t+1}(a,U) = \kappa_t(a,U)$ and hence $\ln(\kappa_{t+1}(a,U)/\kappa_t(a,U)) = 0$ so:

$$\sum_{(a,U)\in\mathcal{A}} \ln\left(\frac{\kappa_{t+1}(a,U)}{\kappa_t(a,U)}\right) = \ln\left(\frac{\kappa_{t+1}(y(u_t), U')}{\kappa_t(y(u_t), U')}\right)$$

$$= \ln\left(\frac{\kappa_t(y(u_t), U')([\![a_t = y(u_t)]\!]\lambda_t + [\![a_t \neq y(u_t)]\!])z_t}{\kappa_t(y(u_t), U')}\right)$$

$$= \ln(([\![a_t = y(u_t)]\!]\lambda_t + [\![a_t \neq y(u_t)]\!])z_t)$$

$$= \ln(c_{t,y(u_t)}z_t)$$

$$= \ln\left(c_{t,y(u_t)}\frac{\|\boldsymbol{p}_t\|_1}{\|\boldsymbol{p}_t\|_1 - (1-\lambda_t)p_{t,a_t}}\right)$$

$$= \ln\left(c_{t,y(u_t)}\frac{\|\boldsymbol{p}_t\|_1}{\lambda_t p_{t,a_t} + \sum_{a\in[K]\backslash\{a_t\}} p_{t,a}}\right)$$

$$= \ln\left(c_{t,y(u_t)}\frac{\|\boldsymbol{p}_t\|_1}{c_{t,a_t}p_{t,a_t} + \sum_{a\in[K]\backslash\{a_t\}} c_{t,a}p_{t,a}}\right)$$

$$= \ln\left(c_{t,y(u_t)}\frac{\|\boldsymbol{p}_t\|_1}{\boldsymbol{c}_t \cdot \boldsymbol{p}_t}\right)$$

$$= \ln\left(\frac{c_{t,y(u_t)}}{\boldsymbol{c}_t \cdot \tilde{\boldsymbol{p}}_t}\right)$$

as required. $\qquad\square$

**Lemma 41.** *We have:*

$$\mathbb{E}\left[\ln\left(\frac{c_{t,y(u_t)}}{\boldsymbol{c}_t \cdot \tilde{\boldsymbol{p}}_t}\right)\right] \geq \eta\left(\mathbb{E}\left[\ell_{t,a_t}\right] - \ell_{t,y(u_t)}\right) - \frac{K\eta^2}{2}.$$

*Proof.* Noting that, for all $a \in [K]$:

$$c_{t,a} = \exp\left(-\frac{\eta[\![a = a_t]\!]\ell_{t,a_t}}{\tilde{p}_{t,a}}\right)$$

we have:

$$\ln\left(\frac{c_{t,y(u_t)}}{\tilde{\boldsymbol{p}}_t \cdot \boldsymbol{c}_t}\right)$$

$$= \ln(c_{t,y(u_t)}) - \ln(\tilde{\boldsymbol{p}}_t \cdot \boldsymbol{c}_t)$$

$$= \ln(c_{t,y(u_t)}) - \ln\left(\sum_{a\in[K]} \tilde{p}_{t,a} c_{t,a}\right)$$

$$= -\frac{\eta[\![y(u_t) = a_t]\!]\ell_{t,a_t}}{\tilde{p}_{t,y(u_t)}} - \ln\left(\sum_{a\in[K]} \tilde{p}_{t,a}\exp\left(\frac{-\eta[\![a = a_t]\!]\ell_{t,a_t}}{\tilde{p}_{t,a}}\right)\right)$$

$$\geq -\frac{\eta[\![y(u_t) = a_t]\!]\ell_{t,a_t}}{\tilde{p}_{t,y(u_t)}} - \ln\left(\sum_{a\in[K]} \tilde{p}_{t,a}\left(1 - \frac{\eta[\![a = a_t]\!]\ell_{t,a_t}}{\tilde{p}_{t,a}} + \frac{\eta^2[\![a = a_t]\!](\ell_{t,a_t})^2}{2(\tilde{p}_{t,a})^2}\right)\right) \quad (15)$$

$$= -\frac{\eta[\![y(u_t) = a_t]\!]\ell_{t,a_t}}{\tilde{p}_{t,y(u_t)}} - \ln\left(\sum_{a\in[K]} \tilde{p}_{t,a} - \sum_{a\in[K]} \eta[\![a = a_t]\!]\ell_{t,a_t} + \sum_{a\in[K]} \frac{\eta^2[\![a = a_t]\!](\ell_{t,a_t})^2}{2\tilde{p}_{t,a}}\right)$$

$$= -\frac{\eta[\![y(u_t) = a_t]\!]\ell_{t,a_t}}{\tilde{p}_{t,y(u_t)}} - \ln\left(1 - \sum_{a\in[K]} \eta[\![a = a_t]\!]\ell_{t,a_t} + \sum_{a\in[K]} \frac{\eta^2[\![a = a_t]\!](\ell_{t,a_t})^2}{2\tilde{p}_{t,a}}\right)$$

$$= -\frac{\eta[\![y(u_t) = a_t]\!]\ell_{t,a_t}}{\tilde{p}_{t,y(u_t)}} - \ln\left(1 - \eta\ell_{t,a_t} + \frac{\eta^2(\ell_{t,a_t})^2}{2\tilde{p}_{t,a_t}}\right)$$

$$\geq -\frac{\eta[\![y(u_t) = a_t]\!]\ell_{t,a_t}}{\tilde{p}_{t,y(u_t)}} + \eta\ell_{t,a_t} - \frac{\eta^2(\ell_{t,a_t})^2}{2\tilde{p}_{t,a_t}} \quad (16)$$

$$= \eta\ell_{t,a_t} - \frac{\eta[\![y(u_t) = a_t]\!]\ell_{t,a_t}}{\tilde{p}_{t,y(u_t)}} - \frac{\eta^2(\ell_{t,a_t})^2}{2\tilde{p}_{t,a_t}}$$

where inequalities (15) and (16) are since $\exp(-x) \leq 1 - x + x^2/2$ for $x \geq 0$ and $\ln(1 + x) \leq x$ respectively. This implies:

$$\mathbb{E}\left[\ln\left(\frac{c_{t,y(u_t)}}{\tilde{\boldsymbol{p}}_t \cdot \boldsymbol{c}_t}\right)\right] \geq \sum_{a\in[K]} \mathbb{P}\left[a_t = a\right]\left(\eta\ell_{t,a} - \frac{\eta[\![y(u_t) = a]\!]\ell_{t,a}}{\tilde{p}_{t,y(u_t)}} - \frac{\eta^2(\ell_{t,a})^2}{2\tilde{p}_{t,a}}\right)$$

$$= \sum_{a\in[K]} \tilde{p}_{t,a}\left(\eta\ell_{t,a} - \frac{\eta[\![y(u_t) = a]\!]\ell_{t,a}}{\tilde{p}_{t,y(u_t)}} - \frac{\eta^2(\ell_{t,a})^2}{2\tilde{p}_{t,a}}\right)$$

$$= \eta\sum_{a\in[K]} \tilde{p}_{t,a}\ell_{t,a} - \sum_{a\in[K]} \frac{\eta\tilde{p}_{t,a}[\![y(u_t) = a]\!]\ell_{t,a}}{\tilde{p}_{t,y(u_t)}} - \sum_{a\in[K]} \frac{\eta^2(\ell_{t,a})^2}{2}$$

$$= \eta\sum_{a\in[K]} \tilde{p}_{t,a}\ell_{t,a} - \frac{\eta\tilde{p}_{t,y(u_t)}\ell_{t,y(u_t)}}{\tilde{p}_{t,y(u_t)}} - \sum_{a\in[K]} \frac{\eta^2(\ell_{t,a})^2}{2}$$

$$= \eta \sum_{a\in[K]} \tilde{p}_{t,a}\ell_{t,a} - \eta\ell_{t,y(u_t)} - \sum_{a\in[K]} \frac{\eta^2(\ell_{t,a})^2}{2}$$

$$= \eta \sum_{a\in[K]} \mathbb{P}\left[a_t = a\right] \ell_{t,a_t} - \eta\ell_{t,y(u_t)} - \sum_{a\in[K]} \frac{\eta^2(\ell_{t,a})^2}{2}$$

$$= \eta\mathbb{E}\left[\ell_{t,a_t}\right] - \eta\ell_{t,y(u_t)} - \sum_{a\in[K]} \frac{\eta^2(\ell_{t,a})^2}{2}$$

$$\geq \eta\mathbb{E}\left[\ell_{t,a_t}\right] - \eta\ell_{t,y(u_t)} - \sum_{a\in[K]} \frac{\eta^2}{2}$$

$$= \eta\mathbb{E}\left[\ell_{t,a_t}\right] - \eta\ell_{t,y(u_t)} - \frac{K\eta^2}{2}$$

as required. $\square$

Lemmas 40 and 41 imply the result. ∎

### H.2 Proof of Lemma 8

**Result**. We have:

$$\sum_{(a,U)\in\mathcal{A}} \ln\left(|\mathcal{A}|\kappa_{T+1}(a,U)\right) \leq 0\,.$$

**Proof:**

**Definition 42.** *Given a finite set $X$ let $\Delta_X$ be the set of functions $f$ from $X$ into $\mathbb{R}^+$ such that $\sum_{x\in X} f(x) = 1$.*

**Lemma 43.** *For all $t \in [T+1]$ we have $\kappa_t \in \Delta_{[K]\times\mathcal{E}}$.*

*Proof.* We prove via induction over $t$. For $t = 1$ the result holds as:

$$\sum_{(a,U)\in[K]\times\mathcal{E}} \kappa_1(a,U) = \sum_{(a,U)\in[K]\times\mathcal{E}} \frac{\sigma(U)}{K}$$

$$= \frac{1}{K} \sum_{a\in[K]} \sum_{U\in\mathcal{E}} \sigma(U)$$

$$= \frac{1}{K} \sum_{a\in[K]} 1$$

$$= 1\,.$$

Now suppose it holds for $t = s$ (for some $s \in [T]$). We now show it holds for $t = s + 1$, completing the proof. We have:

$$\sum_{(a,U)\in[K]\times\mathcal{E}:u_s\in U} \kappa_{s+1}(a,U)$$

$$= \sum_{U\in\mathcal{E}:u_s\in U} \left( \kappa_{s+1}(a_s, U) + \sum_{a\in[K]\setminus\{a_s\}} \kappa_{s+1}(a,U) \right)$$

$$= \sum_{U\in\mathcal{E}:u_s\in U} \left( \lambda_s z_s \kappa_s(a_s, U) + \sum_{a\in[K]\setminus\{a_s\}} z_s \kappa_s(a,U) \right)$$

$$= z_s \left( \lambda_s \sum_{U\in\mathcal{E}:u_s\in U} \kappa_s(a_s, U) + \sum_{a\in[K]\setminus\{a_s\}} \sum_{U\in\mathcal{E}:u_s\in U} \kappa_s(a,U) \right)$$

$$= z_s \left( \lambda_s p_{s,a_s} + \sum_{a \in [K] \setminus \{a_s\}} p_{s,a} \right)$$

$$= z_s \left( \lambda_s p_{s,a_s} + (\|\boldsymbol{p}_s\|_1 - p_{s,a_s}) \right)$$

$$= z_s \left( \|\boldsymbol{p}_s\|_1 - (1 - \lambda_s) p_{s,a_s} \right)$$

$$= z_s \frac{\|\boldsymbol{p}_s\|_1}{z_s}$$

$$= \|\boldsymbol{p}_s\|_1 ,$$

so:

$$\sum_{(a,U) \in [K] \times \mathcal{E}} \kappa_{s+1}(a, U)$$

$$= \sum_{(a,U) \in [K] \times \mathcal{E} : u_s \in U} \kappa_{s+1}(a, U) + \sum_{(a,U) \in [K] \times \mathcal{E} : u_s \notin U} \kappa_{s+1}(a, U)$$

$$= \sum_{(a,U) \in [K] \times \mathcal{E} : u_s \in U} \kappa_{s+1}(a, U) + \sum_{(a,U) \in [K] \times \mathcal{E} : u_s \notin U} \kappa_s(a, U)$$

$$= \|\boldsymbol{p}_s\|_1 + \sum_{(a,U) \in [K] \times \mathcal{E} : u_s \notin U} \kappa_s(a, U)$$

$$= \sum_{a \in [K]} p_{s,a} + \sum_{(a,U) \in [K] \times \mathcal{E} : u_s \notin U} \kappa_s(a, U)$$

$$= \sum_{a \in [K]} \sum_{U \in \mathcal{E} : u_s \in U} \kappa_s(a, U) + \sum_{(a,U) \in [K] \times \mathcal{E} : u_s \notin U} \kappa_s(a, U)$$

$$= \sum_{(a,U) \in [K] \times \mathcal{E}} \kappa_s(a, U)$$

$$= 1$$

as required. $\square$

**Definition 44.** *We define the function $\kappa^* \in \Delta_{[K] \times \mathcal{E}}$ by:*

- *For all $(a, U) \in \mathcal{A}$ we have $\kappa^*(a, U) := 1/|\mathcal{A}|$.*

- *For all $(a, U) \in ([K] \times \mathcal{E}) \setminus \mathcal{A}$ we have $\kappa^*(a, U) := 0$.*

**Lemma 45.** *We have:*

$$\sum_{(a,U) \in [K] \times \mathcal{E}} \kappa^*(a, U) \ln \left( \frac{\kappa^*(a, U)}{\kappa_{T+1}(a, U)} \right) \geq 0 .$$

*Proof.* Given a finite set $X$ and functions $f, f' \in \Delta_X$, the value $\sum_{x \in X} f(x) \ln \left( \frac{f(x)}{f'(x)} \right)$ is their relative entropy and is hence positive. The result then follows by Lemma 43 (i.e. that $\kappa_{T+1} \in \Delta_{[K] \times \mathcal{E}}$) and the fact that $\kappa^* \in \Delta_{[K] \times \mathcal{E}}$ $\square$

By taking limits we have $\kappa^*(a, U) \ln \left( \kappa^*(a, U) / \kappa_{T+1}(a, U) \right) = 0$ whenever $(a, U) \notin \mathcal{A}$. By definition of $\kappa^*$ we then have:

$$- \sum_{(a,U) \in \mathcal{A}} \ln \left( |\mathcal{A}| \kappa_{T+1}(a, U) \right) = |\mathcal{A}| \sum_{(a,U) \in \mathcal{A}} \frac{1}{|\mathcal{A}|} \ln \left( \frac{1/|\mathcal{A}|}{\kappa_{T+1}(a, U)} \right)$$

$$= \frac{1}{|\mathcal{A}|} \sum_{(a,U) \in [K] \times \mathcal{E}} \kappa^*(a, U) \ln \left( \frac{\kappa^*(a, U)}{\kappa_{T+1}(a, U)} \right)$$

which, by Lemma 45 is bounded below by zero. This implies the result. $\blacksquare$

### H.3 Proof of Theorem 9

**Result**. We have:

$$R \leq \frac{1}{\eta} \sum_{(a,U) \in \mathcal{A}} \ln \left( \frac{K}{|\mathcal{A}|\sigma(U)} \right) + \frac{\eta K T}{2} \,.$$

**Proof:**

For all $(a, U) \in \mathcal{A}$ we have:

$$\ln(|\mathcal{A}|\kappa_{T+1}(a,U)) - \ln(|\mathcal{A}|\kappa_1(a,U)) = \sum_{t \in [T]} (\ln(|\mathcal{A}|\kappa_{t+1}(a,U)) - \ln(|\mathcal{A}|\kappa_t(a,U)))$$

$$= \sum_{t \in [T]} \ln \left( \frac{|\mathcal{A}|\kappa_{t+1}(a,U)}{|\mathcal{A}|\kappa_t(a,U)} \right)$$

$$= \sum_{t \in [T]} \ln \left( \frac{\kappa_{t+1}(a,U)}{\kappa_t(a,U)} \right)$$

so:

$$\sum_{(a,U) \in \mathcal{A}} (\ln(|\mathcal{A}|\kappa_{T+1}(a,U)) - \ln(|\mathcal{A}|\kappa_1(a,U))) = \sum_{t \in [T]} \sum_{(a,U) \in \mathcal{A}} \ln \left( \frac{\kappa_{t+1}(a,U)}{\kappa_t(a,U)} \right) \,.$$

Applying Lemma 7 then gives us:

$$\mathbb{E} \left[ \sum_{(a,U) \in \mathcal{A}} (\ln(|\mathcal{A}|\kappa_{T+1}(a,U)) - \ln(|\mathcal{A}|\kappa_1(a,U))) \right] \geq \sum_{t \in [T]} \left( \eta \left( \mathbb{E}\left[\ell_{t,a_t}\right] - \ell_{t,y(u_t)} \right) - \frac{K\eta^2}{2} \right)$$

$$= \sum_{t \in [T]} \left( \eta \mathbb{E}\left[\ell_{t,a_t} - \ell_{t,y(u_t)}\right] - \frac{K\eta^2}{2} \right)$$

$$= \eta \sum_{t \in [T]} \mathbb{E}\left[\ell_{t,a_t} - \ell_{t,y(u_t)}\right] - \frac{TK\eta^2}{2}$$

$$= \eta \mathbb{E}\left[ \sum_{t \in [T]} (\ell_{t,a_t} - \ell_{t,y(u_t)}) \right] - \frac{TK\eta^2}{2} \,,$$

so:

$$\mathbb{E}\left[ \sum_{t \in [T]} (\ell_{t,a_t} - \ell_{t,y(u_t)}) \right] \leq \frac{1}{\eta}\mathbb{E}\left[ \sum_{(a,U) \in \mathcal{A}} \ln(|\mathcal{A}|\kappa_{T+1}(a,U)) \right] - \frac{1}{\eta}\mathbb{E}\left[ \sum_{(a,U) \in \mathcal{A}} \ln(|\mathcal{A}|\kappa_1(a,U)) \right] + \frac{TK\eta}{2} \,.$$

Applying Lemma 8 then gives us:

$$\mathbb{E}\left[ \sum_{t \in [T]} (\ell_{t,a_t} - \ell_{t,y(u_t)}) \right] \leq -\frac{1}{\eta}\mathbb{E}\left[ \sum_{(a,U) \in \mathcal{A}} \ln(|\mathcal{A}|\kappa_1(a,U)) \right] + \frac{TK\eta}{2} \,,$$

and the definition of $\kappa_1$ then gives us:

$$\mathbb{E}\left[ \sum_{t \in [T]} (\ell_{t,a_t} - \ell_{t,y(u_t)}) \right] \leq \frac{1}{\eta}\mathbb{E}\left[ \sum_{(a,U) \in \mathcal{A}} \ln \left( \frac{K}{|\mathcal{A}|\sigma(U)} \right) \right] + \frac{\eta K T}{2}$$

as required. ∎

# I  GABA-I Proofs

## I.1  Proof of Lemma 11

**Result.** We have that $\sigma$ is a probability distribution, in that:

$$\sum_{U \in \mathcal{E}} \sigma(U) = 1\,.$$

**Proof:**

**Definition 46.** *For $f : [N] \to \{0,1\}$ define:*

$$\hat{\sigma}(f) := \iota_{f(1)} \prod_{u \in [N-1]} \tau_{f(u),f(u+1)}\,.$$

**Definition 47.** *For $U \in \mathcal{E}$ let $\gamma_U : [N] \to \{0,1\}$ be such that for all $u \in [N]$ we have $\gamma_U(u) := [\![ u \in U ]\!]$.*

Since $\tau_{0,0} + \tau_{0,1} = 1$, $\tau_{1,0} + \tau_{1,1} = 1$ and $\tau_{i,j} \geq 0$ for all $i,j \in \{0,1\}$ we have that $\boldsymbol{\tau}$ is the transistion matrix of a Markov chain. Since also $\iota_0 + \iota_1 = 1$ and $\iota_i \geq 0$ for all $i \in \{0,1\}$, we then have, for all $f : [N] \to \{0,1\}$, that $\hat{\sigma}(f)$ is the probability of $f$ in a Markov chain. This implies that $\sum_{f:[N]\to\{0,1\}} \hat{\sigma}(f) = 1$ so:

$$\begin{aligned}
\sum_{U \in \mathcal{E}} \sigma(U) &= \sum_{U \in \mathcal{E}} \iota_{[\![ 1 \in U ]\!]} \prod_{u \in [N-1]} \tau_{[\![ u \in U ]\!],[\![ u+1 \in U ]\!]} \\
&= \sum_{U \in \mathcal{E}} \iota_{\gamma_U(1)} \prod_{u \in [N-1]} \tau_{\gamma_U(u),\gamma_U(u+1)} \\
&= \sum_{U \in 2^{[N]}} \iota_{\gamma_U(1)} \prod_{u \in [N-1]} \tau_{\gamma_U(u),\gamma_U(u+1)} \\
&= \sum_{f:[N]\to\{0,1\}} \iota_{f(1)} \prod_{u \in [N-1]} \tau_{f(u),f(u+1)} \\
&= \sum_{f:[N]\to\{0,1\}} \hat{\sigma}(f) \\
&= 1
\end{aligned}$$

as required.  ∎

## I.2  Proof of Lemma 16

**Result.** Given $i,j,k \in \{0,1\}$, $u,v,w \in [N]$ with $u < v < w$, and $a \in [K]$ we have:

$$\left( \sum_{f \in F} \Omega_{t,a}(f) \right) \left( \sum_{f \in G} \Omega_{t,a}(f) \right) = \sum_{f \in H} \Omega_{t,a}(f)\,,$$

where:

- $F = \{ f \in I(u,v) \mid f(u) = i\,, f(v) = j \}$
- $G = \{ f \in I(v,w) \mid f(v) = j\,, f(w) = k \}$
- $H = \{ f \in I(u,w) \mid f(u) = i\,, f(v) = j\,, f(w) = k \}\,.$

**Proof:**

Given $(f,f') \in F \times G$ let $\xi_{f,f'}$ be the function in $H$ defined by $\xi_{f,f'}(u') := f(u')$ for $u \leq u' \leq v$ and $\xi_{f,f'}(u') := f'(u')$ for $v \leq u' \leq w$. Note first that by definition of $\Omega_{t,a}$ we have, for all

$(f, f') \in F \times G$, that $\Omega_{t,j}(f)\Omega_{t,a}(f') = \Omega_{t,j}(\xi_{f,f'})$. Note also that the function $(f, f') \to \xi_{f,f'}$ is a bijection from $F \times G$ into $H$. Hence, we have:

$$\left(\sum_{f \in F} \Omega_{t,a}(f)\right)\left(\sum_{f \in G} \Omega_{t,a}(f)\right) = \left(\sum_{f \in F} \Omega_{t,a}(f)\right)\left(\sum_{f' \in G} \Omega_{t,a}(f')\right)$$

$$= \sum_{f \in F}\sum_{f' \in G} \Omega_{t,a}(f)\Omega_{t,a}(f')$$

$$= \sum_{(f,f') \in F \times G} \Omega_{t,a}(f)\Omega_{t,a}(f')$$

$$= \sum_{(f,f') \in F \times G} \Omega_{t,a}(\xi_{f,f'})$$

$$= \sum_{f^\dagger \in H} \Omega_{t,a}(f^\dagger)$$

as required. ■

### I.3 Proof of Lemma 17

**Result**. For all $t \in [T] \cup \{0\}$, all non-root vertices $n$ of $\mathcal{C}$, all $i, j \in \{0, 1\}$ and all $a \in [K]$ we have:

$$\alpha_t(n, i, j)_a = \sum_{f \in A_t(n,i,j)} \Omega_{t,a}(f),$$

where:

$$A_t(n, i, j) := \{f \in I(\blacktriangleleft(n), \blacktriangleright(n) + 1) \mid f(\blacktriangleleft(n)) = i, f(\blacktriangleright(n) + 1) = j\}.$$

**Proof:**

**Lemma 48.** *For all $t \in [T] \cup \{0\}$, $u \in [N]$, $i, j \in \{0, 1\}$ and $a \in [K]$, we have:*

$$\alpha_t(u, i, j)_a = \tau_{i,j}(\llbracket i = 0 \rrbracket + \llbracket i = 1 \rrbracket \xi_{t,u,a}).$$

*Proof.* We prove by induction on $t$. By the initialization algorithm we have that:

$$\alpha_0(u, i, j)_a = \tau_{i,j}$$
$$= \tau_{i,j}(\llbracket i = 0 \rrbracket + \llbracket i = 1 \rrbracket)$$
$$= \tau_{i,j}(\llbracket i = 0 \rrbracket + \llbracket i = 1 \rrbracket \xi_{0,u,a})$$

so the result holds for $t = 0$. Now suppose the result holds for $t = s$ (for some $s \in [T] \cup \{0\}$). By the update algorithm on trial $s$ and the inductive hypothesis we then have:

- If $u \neq u_s$ then:

$$\alpha_{s+1}(u, i, j)_a := \alpha_s(u, i, j)_a$$
$$= \tau_{i,j}(\llbracket i = 0 \rrbracket + \llbracket i = 1 \rrbracket \xi_{s,u,a})$$
$$= \tau_{i,j}\llbracket i = 0 \rrbracket + \tau_{i,j}\llbracket i = 1 \rrbracket \prod_{q \in [s-1]:u=u_q} \pi_{q,a}$$
$$= \tau_{i,j}\llbracket i = 0 \rrbracket + \tau_{i,j}\llbracket i = 1 \rrbracket \prod_{q \in [s]:u=u_q} \pi_{q,a}$$
$$= \tau_{i,j}\llbracket i = 0 \rrbracket + \tau_{i,j}\llbracket i = 1 \rrbracket \xi_{s+1,u,a}$$

- If $u = u_s$ then:

$$\alpha_{s+1}(u, i, j)_a$$

$$:= [\![i=0]\!]\alpha_s(u,i,j)_a + [\![i=1]\!]\pi_{s,a}\alpha_s(u,i,j)_a$$
$$= [\![i=0]\!]\tau_{i,j}([\![i=0]\!] + [\![i=1]\!]\xi_{s,u,a}) + [\![i=1]\!]\pi_{s,a}\tau_{i,j}([\![i=0]\!] + [\![i=1]\!]\xi_{s,u,a})$$
$$= [\![i=0]\!]\tau_{i,j} + [\![i=1]\!]\pi_{s,a}\tau_{i,j}\xi_{s,u,a}$$
$$= [\![i=0]\!]\tau_{i,j} + [\![i=1]\!]\tau_{i,j}\pi_{s,a} \prod_{q\in[s-1]:u=u_q} \pi_{q,a}$$
$$= [\![i=0]\!]\tau_{i,j} + [\![i=1]\!]\tau_{i,j} \prod_{q\in[s]:u=u_q} \pi_{q,a}$$
$$= \tau_{i,j}[\![i=0]\!] + \tau_{i,j}[\![i=1]\!]\xi_{s,u,a}$$

as required. $\square$

**Lemma 49.** *For all $t \in [T] \cup \{0\}$, all internal, non-root vertices $n$ of $\mathcal{C}$, all $i,j \in \{0,1\}$ and all $a \in [K]$ we have:*
$$\alpha_t(n,i,j)_a = \sum_{k\in\{0,1\}} \alpha_t(\lhd(n),i,k)_a \alpha_t(\rhd(n),k,j)_a\,.$$

*Proof.* We prove by induction on $t$. For $t=0$ the result is direct from the initialization algorithm. Now suppose the result holds for $t=s$ (for some $s \in [T-1] \cup \{0\}$). We now show that it holds for $t=s+1$, completing the proof. First, suppose $n \neq \nu_{s,d}$ for all $d \in [h]$. Since, in this case, we must also have that $\lhd(n), \rhd(n) \neq \nu_{s,d}$ for all $d \in [h]$ we then have, by the update algorithm, that, for $k \in \{0,1\}$, $\alpha_{s+1}(\lhd(n),i,k)_a := \alpha_s(\lhd(n),i,k)_a$ and $\alpha_{s+1}(\rhd(n),k,j)_a := \alpha_s(\rhd(n),k,j)_a$ so hence, by the inductive hypothesis and the update algorithm:

$$\alpha_{s+1}(n,i,j)_a := \alpha_s(n,i,j)_a$$
$$= \sum_{k\in\{0,1\}} \alpha_s(\lhd(n),i,k)_a \alpha_s(\rhd(n),k,j)_a$$
$$= \sum_{k\in\{0,1\}} \alpha_{s+1}(\lhd(n),i,k)_a \alpha_{s+1}(\rhd(n),k,j)_a$$

as required. On the other hand, it is clear from the update algorithm that if $n = \nu_{t,d}$ for some $d \in [h]$ then:
$$\alpha_{s+1}(n,i,j)_a := \sum_{k\in\{0,1\}} \alpha_{s+1}(\lhd(n),i,k)_a \alpha_{s+1}(\rhd(n),k,j)_a$$

as required. $\square$

With lemmas 48 and 49 at hand we prove the result by reverse induction on the depth of $n$ (i.e. from depth $h$ to depth 1). We first consider the case that $n$ is of depth $h$. In this case we have that $n$ is a leaf and have $\blacktriangleleft(n) = \blacktriangleright(n) = n$. This means that $I(\blacktriangleleft(n), \blacktriangleright(n)+1)$ is the set of functions from $\{n, n+1\}$ into $\{0,1\}$ and hence the set $\{f \in I(\blacktriangleleft(n), \blacktriangleright(n)+1) \mid f(\blacktriangleleft(n)) = i\,, f(\blacktriangleright(n)+1) = j\}$ contains a single function $f$ with $f(n) = i$ and $f(n+1) = j$. We then have, by Lemma 48, that:

$$\Omega_{t,a}(f) = \tau_{f(n),f(n+1)}([\![f(n)=0]\!] + [\![f(n)=1]\!]\xi_{t,n,a})$$
$$= \tau_{i,j}([\![i=0]\!] + [\![i=1]\!]\xi_{t,n,a})$$
$$= \alpha_t(n,i,j)_a$$

as required. So the inductive hypothesis holds for $n$ at depth $h$. Now suppose the inductive hypothesis holds for $n$ at depth $d$ (for some $d \in [h]$). We now show it holds for $n$ at depth $d-1$ which will complete the proof. For $k \in \{0,1\}$ let:

$$F_k := A_t(\lhd(n),i,k) = \{f \in I(\blacktriangleleft(\lhd(n)), \blacktriangleright(\lhd(n))+1) \mid f(\blacktriangleleft(\lhd(n))) = i\,, f(\blacktriangleright(\lhd(n))+1) = k\}\,.$$

and let:

$$G_k := A_t(\rhd(n),k,j) = \{f \in I(\blacktriangleleft(\rhd(n)), \blacktriangleright(\rhd(n))+1) \mid f(\blacktriangleleft(\rhd(n))) = k\,, f(\blacktriangleright(\rhd(n))+1) = j\}\,.$$

Since $\lhd(n)$ and $\rhd(n)$ are at depth $d$ we have, by the inductive hypothesis and Lemma 49 that:

$$\alpha_t(n,i,j)_a = \sum_{k\in\{0,1\}} \alpha_t(\lhd(n),i,k)_a \alpha_t(\rhd(n),k,j)_a$$

$$= \sum_{k \in \{0,1\}} \left( \sum_{f \in F_k} \Omega_{t,a}(f) \right) \left( \sum_{f \in G_k} \Omega_{t,a}(f) \right).$$

Since $\blacktriangleright(\triangleleft(n)) + 1 = \blacktriangleleft(\triangleright(n))$ we then have, by Lemma 16, that:

$$\alpha_t(n, i, j)_a = \sum_{k \in \{0,1\}} \sum_{f \in I(\blacktriangleleft(\triangleleft(n)), \blacktriangleright(\triangleright(n))+1) : f(\blacktriangleleft(\triangleleft(n)))=i, f(\blacktriangleleft(\triangleright(n)))=k, f(\blacktriangleright(\triangleright(n))+1)=j} \Omega_{t,a}(f)$$

$$= \sum_{f \in I(\blacktriangleleft(\triangleleft(n)), \blacktriangleright(\triangleright(n))+1) : f(\blacktriangleleft(\triangleleft(n)))=i, f(\blacktriangleright(\triangleright(n))+1)=j} \Omega_{t,a}(f)$$

$$= \sum_{f \in I(\blacktriangleleft(n), \blacktriangleright(n)+1) : f(\blacktriangleleft(n))=i, f(\blacktriangleright(n)+1)=j} \Omega_{t,a}(f)$$

as required. ∎

### I.4 Proof of Lemma 18

**Result.** For all $t \in [T]$, $d \in [h] \cup \{0\}$, $i \in \{0, 1\}$ and $a \in [K]$ we have:

$$\beta_t^{\Rightarrow}(\nu_{t,d}, i)_a = \sum_{f \in B_t^{\Rightarrow}(\nu_{t,d}, i)} \Omega_{t,a}(f),$$

where:

$$B_t^{\Rightarrow}(\nu_{t,d}, i) := \{f \in I(\blacktriangleright(\nu_{t,d}) + 1, N + 1) \mid f(\blacktriangleright(\nu_{t,d}) + 1) = i\},$$

and:

$$\beta_t^{\Leftarrow}(\nu_{t,d}, i)_a = \sum_{f \in B_t^{\Leftarrow}(\nu_{t,d}, i)} \iota_{f(1)} \Omega_{t,a}(f),$$

where:

$$B_t^{\Leftarrow}(\nu_{t,d}, i) := \{f \in I(1, \blacktriangleleft(\nu_{t,d})) \mid f(\blacktriangleleft(\nu_{t,d})) = i\}.$$

**Proof:**

**Lemma 50.** *For all $t \in [T]$, $d \in [h] \cup \{0\}$, $i \in \{0, 1\}$ and $a \in [K]$ we have:*

$$\beta_t^{\Rightarrow}(\nu_{t,d}, i)_a = \sum_{f \in I(\blacktriangleright(\nu_{t,d})+1, N+1) : f(\blacktriangleright(\nu_{t,d})+1)=i} \Omega_{t,a}(f).$$

*Proof.* We prove by induction over $d$. In the case that $d = 0$ we have that $\nu_{t,d}$ is the root so $\blacktriangleright(\nu_{t,d}) + 1 = N + 1$. From the algorithm we see that $\beta_t^{\Rightarrow}(\nu_{t,0}, i)_a = 1$ so we have, since $\{u \in \mathbb{N} \mid N+1 \le u < N+1\} = \emptyset$ and $\{f \in I(N+1, N+1) \mid f(N+1) = i\}$ contains the single function $f'$ which maps $N + 1$ to $i$, that:

$$\beta_t^{\Rightarrow}(\nu_{t,0}, i)_a = 1$$
$$= \Omega_{t,a}(f')$$
$$= \sum_{f \in I(N+1, N+1) : f(N+1)=i} \Omega_{t,a}(f)$$
$$= \sum_{f \in I(\blacktriangleright(\nu_{t,0})+1, N+1) : f(\blacktriangleright(\nu_{t,0})+1)=i} \Omega_{t,a}(f)$$

as required. Now suppose the result holds for some $d = q$ (for some $q \in [h-1] \cup \{0\}$) we now show that it holds for $d = q + 1$ which will complete the proof. We have two cases:

- If $\nu_{t,q+1} = \triangleright(\nu_{t,q})$ then from the algorithm we have $\beta_t^{\Rightarrow}(\nu_{t,q+1}, i)_a := \beta_t^{\Rightarrow}(\nu_{t,q}, i)_a$ so since in this case $\blacktriangleright(\nu_{t,q+1}) = \blacktriangleright(\nu_{t,q})$ we have, by the inductive hypothesis:

$$\beta_t^{\Rightarrow}(\nu_{t,q+1}, i)_a := \beta_t^{\Rightarrow}(\nu_{t,q}, i)_a$$

$$= \sum_{f \in I(\blacktriangleright(\nu_{t,q})+1,N+1):f(\blacktriangleright(\nu_{t,q})+1)=i} \Omega_{t,a}(f)$$

$$= \sum_{f \in I(\blacktriangleright(\nu_{t,q+1})+1,N+1):f(\blacktriangleright(\nu_{t,q+1})+1)=i} \Omega_{t,a}(f)$$

as required.

- If $\nu_{t,q+1} = \triangleleft(\nu_{t,q})$ then for all $j, k \in \{0,1\}$ define:

$$\begin{aligned} F_j :&= A_t(\triangleright(\nu_{t,q}), i, j) \\ &= \{f \in I(\blacktriangleleft(\triangleright(\nu_{t,q})), \blacktriangleright(\triangleright(\nu_{t,q}))+1) \mid f(\blacktriangleleft(\triangleright(\nu_{t,q}))) = i\,, f(\blacktriangleright(\triangleright(\nu_{t,q}))+1) = j\}\,, \end{aligned}$$

and:

$$G_{j,k} := \{f \in I(\blacktriangleright(\nu_{t,q})+1, N+1) \mid f(\blacktriangleright(\nu_{t,q})+1) = j\,, f(N+1) = k\}\,.$$

From the algorithm we have:

$$\beta_t^{\Rightarrow}(\nu_{t,q+1}, i)_a := \sum_{j \in \{0,1\}} \alpha_t(\triangleright(\nu_{t,q}), i, j)_a \beta_t^{\Rightarrow}(\nu_{t,q}, j)_a\,.$$

From Lemma 17 we have, for all $j \in \{0,1\}$ that:

$$\alpha_t(\triangleright(\nu_{t,q}), i, j)_a = \sum_{f \in F_j} \Omega_{t,a}(f)\,,$$

and from the inductive hypothesis we have, for all $j \in \{0,1\}$ that:

$$\begin{aligned} \beta_t^{\Rightarrow}(\nu_{t,q}, j)_a &= \sum_{f \in I(\blacktriangleright(\nu_{t,q})+1,N+1):f(\blacktriangleright(\nu_{t,q})+1)=j} \Omega_{t,a}(f) \\ &= \sum_{f \in G_{j,0} \cup G_{j,1}} \Omega_{t,a}(f) \\ &= \sum_{k \in \{0,1\}} \sum_{f \in G_{j,k}} \Omega_{t,a}(f)\,, \end{aligned}$$

so since $\blacktriangleright(\triangleright(\nu_{t,q})) + 1 = \blacktriangleright(\nu_{t,q}) + 1$ we have, by Lemma 16, that:

$$\begin{aligned} &\beta_t^{\Rightarrow}(\nu_{t,q+1}, i)_a \\ &:= \sum_{j \in \{0,1\}} \alpha_t(\triangleright(\nu_{t,q}), i, j)_a \beta_t^{\Rightarrow}(\nu_{t,q}, j)_a \\ &= \sum_{j \in \{0,1\}} \left( \sum_{f \in F_j} \Omega_{t,a}(f) \right) \left( \sum_{k \in \{0,1\}} \sum_{f \in G_{j,k}} \Omega_{t,a}(f) \right) \\ &= \sum_{j \in \{0,1\}} \sum_{k \in \{0,1\}} \left( \sum_{f \in F_j} \Omega_{t,a}(f) \right) \left( \sum_{f \in G_{j,k}} \Omega_{t,a}(f) \right) \\ &= \sum_{j \in \{0,1\}} \sum_{k \in \{0,1\}} \sum_{f \in I(\blacktriangleleft(\triangleright(\nu_{t,q})),N+1):f(\blacktriangleleft(\triangleright(\nu_{t,q})))=i,f(\blacktriangleright(\nu_{t,q})+1)=j,f(N+1)=k} \Omega_{t,a}(f) \\ &= \sum_{f \in I(\blacktriangleleft(\triangleright(\nu_{t,q})),N+1):f(\blacktriangleleft(\triangleright(\nu_{t,q})))=i} \Omega_{t,a}(f) \\ &= \sum_{f \in I(\blacktriangleright(\triangleleft(\nu_{t,q}))+1,N+1):f(\blacktriangleright(\triangleleft(\nu_{t,q}))+1)=i} \Omega_{t,a}(f) \\ &= \sum_{f \in I(\blacktriangleright(\nu_{t,q+1})+1,N+1):f(\blacktriangleright(\nu_{t,q+1})+1)=i} \Omega_{t,a}(f) \end{aligned}$$

as required.

$\square$

**Lemma 51.** *For all $t \in [T]$, $d \in [h] \cup \{0\}$, $i \in \{0,1\}$ and $a \in [K]$ we have:*

$$\beta_t^{\Leftarrow}(\nu_{t,d}, i)_a = \sum_{f \in I(1, \blacktriangleleft(\nu_{t,d})): f(\blacktriangleleft(\nu_{t,d}))=i} \iota_{f(1)} \Omega_{t,a}(f) \,.$$

*Proof.* We prove by induction over $d$. In the case that $d = 0$ we have that $\nu_{t,d}$ is the root so $\blacktriangleleft(\nu_{t,d}) = 1$. From the algorithm we see that $\beta_t^{\Rightarrow}(\nu_{t,0}, i)_a = \iota_i$ so we have, since $\{u \in \mathbb{N} \mid 1 \le u < 1\} = \emptyset$ and $\{f \in I(1,1) \mid f(1) = i\}$ contains the single function $f'$ which maps $1$ to $i$, that:

$$\begin{aligned}
\beta_t^{\Leftarrow}(\nu_{t,0}, i)_a &:= \iota_i \\
&= \iota_i \Omega_{t,a}(f') \\
&= \sum_{f \in I(1,1): f(1)=i} \iota_i \Omega_{t,a}(f) \\
&= \sum_{f \in I(1, \blacktriangleleft(\nu_{t,0})): f(\blacktriangleleft(\nu_{t,0}))=i} \iota_i \Omega_{t,a}(f)
\end{aligned}$$

as required. Now suppose the result holds for $d = q$ (for some $q \in [h-1] \cup \{0\}$). We now show that it holds for $d = q+1$ which will complete the proof. We have two cases:

- If $\nu_{t,q+1} = \triangleleft(\nu_{t,q})$ then from the algorithm we have $\beta_t^{\Leftarrow}(\nu_{t,q+1}, i)_a := \beta_t^{\Leftarrow}(\nu_{t,q}, i)_a$ so since in this case $\blacktriangleleft(\nu_{t,q+1}) = \blacktriangleleft(\nu_{t,q})$ we have, by the inductive hypothesis:

$$\begin{aligned}
\beta_t^{\Leftarrow}(\nu_{t,q+1}, i)_a &:= \beta_t^{\Leftarrow}(\nu_{t,q}, i)_a \\
&= \sum_{f \in I(1, \blacktriangleleft(\nu_{t,q})): f(\blacktriangleleft(\nu_{t,q}))=i} \iota_{f(1)} \Omega_{t,a}(f) \\
&= \sum_{f \in I(1, \blacktriangleleft(\nu_{t,q+1})): f(\blacktriangleleft(\nu_{t,q+1}))=i} \iota_{f(1)} \Omega_{t,a}(f)
\end{aligned}$$

as required.

- If $\nu_{t,q+1} = \triangleright(\nu_{t,q})$ then for all $j, k \in \{0,1\}$ define:

$$\begin{aligned}
G_j &:= A_t(\triangleleft(\nu_{t,q}), j, i) \\
&= \{f \in I(\blacktriangleleft(\triangleleft(\nu_{t,q})), \blacktriangleright(\triangleleft(\nu_{t,q})) + 1) \mid f(\blacktriangleleft(\triangleleft(\nu_{t,q}))) = j, f(\blacktriangleright(\triangleleft(\nu_{t,q})) + 1) = i\} \,,
\end{aligned}$$

and:

$$F_{k,j} = \{f \in I(1, \blacktriangleleft(\nu_{t,q})) \mid f(1) = k, f(\blacktriangleleft(\nu_{t,q})) = j\} \,.$$

From the algorithm we have:

$$\beta_t^{\Leftarrow}(\nu_{t,q+1}, i)_a := \sum_{j \in \{0,1\}} \beta_t^{\Leftarrow}(\nu_{t,q}, j)_a \alpha_t(\triangleleft(\nu_{t,q}), j, i)_a \,.$$

From Lemma 17 we have, for all $j \in \{0,1\}$ that:

$$\alpha_t(\triangleleft(\nu_{t,q}), j, i)_a = \sum_{f \in G_j} \Omega_{t,a}(f) \,,$$

and from the inductive hypothesis we have, for all $j \in \{0,1\}$, that:

$$\begin{aligned}
\beta_t^{\Leftarrow}(\nu_{t,q}, j)_a &= \sum_{f \in I(1, \blacktriangleleft(\nu_{t,q})): f(\blacktriangleleft(\nu_{t,q}))=j} \iota_{f(1)} \Omega_{t,a}(f) \\
&= \sum_{f \in F_{0,j} \cup F_{1,j}} \iota_{f(1)} \Omega_{t,a}(f) \\
&= \sum_{k \in \{0,1\}} \sum_{f \in F_{k,j}} \iota_{f(1)} \Omega_{t,a}(f) \,,
\end{aligned}$$

so since $\blacktriangleleft(\lhd(\nu_{t,q})) = \blacktriangleleft(\nu_{t,q})$ we have, by Lemma 16, that:

$$\beta_t^{\Leftarrow}(\nu_{t,q+1}, i)_a$$

$$:= \sum_{j \in \{0,1\}} \beta_t^{\Leftarrow}(\nu_{t,q}, j)_a \alpha_t(\lhd(\nu_{t,q}), j, i)_a$$

$$= \sum_{j \in \{0,1\}} \left( \sum_{k \in \{0,1\}} \sum_{f \in F_{k,j}} \Omega_{t,a}(f) \right) \left( \sum_{f \in G_j} \Omega_{t,a}(f) \right)$$

$$= \sum_{j \in \{0,1\}} \sum_{k \in \{0,1\}} \left( \sum_{f \in F_{k,j}} \Omega_{t,a}(f) \right) \left( \sum_{f \in G_j} \Omega_{t,a}(f) \right)$$

$$= \sum_{j \in \{0,1\}} \sum_{k \in \{0,1\}} \sum_{f \in I(1, \blacktriangleright(\lhd(\nu_{t,q}))+1): f(1)=k, f(\blacktriangleleft(\nu_{t,q}))=j, f(\blacktriangleright(\lhd(\nu_{t,q}))+1)=i} \Omega_{t,a}(f)$$

$$= \sum_{f \in I(1, \blacktriangleright(\lhd(\nu_{t,q}))+1): f(\blacktriangleright(\lhd(\nu_{t,q}))+1)=i} \Omega_{t,a}(f)$$

$$= \sum_{f \in I(1, \blacktriangleleft(\rhd(\nu_{t,q}))): f(\blacktriangleleft(\rhd(\nu_{t,q})))=i} \Omega_{t,a}(f)$$

$$= \sum_{f \in I(1, \blacktriangleleft(\nu_{t,q+1})): f(\blacktriangleleft(\nu_{t,q+1}))=i} \Omega_{t,a}(f)$$

as required.

$\square$

The result then comes directly from lemmas 50 and 51.

### I.5  Proof of Lemma 19

**Result**. For all $t \in [T]$ and $a \in [K]$ we have:

$$\bar{p}_{t,a} = \sum_{U \in \mathcal{E}: u_t \in U} \bar{\kappa}_t(a, U).$$

**Proof:**

**Lemma 52.** *For all $t \in [T]$ and $a \in [K]$ we have:*

$$\bar{p}_{t,a} = \frac{1}{K} \sum_{f \in I(1, N+1): f(u_t)=1} \iota_{f(1)} \Omega_{t,a}(f).$$

*Proof.* Noting that $u_t = \nu_{t,h}$ and, since $\nu_{t,h}$ is a leaf, $\blacktriangleleft(\nu_{t,h}) = \blacktriangleright(\nu_{t,h}) = \nu_{t,h} = u_t$ we have, by Lemma 18, that for all $i \in \{0,1\}$:

$$\beta_t^{\Rightarrow}(u_t, i)_a = \sum_{f \in I(u_t+1, N+1): f(u_t+1)=i} \Omega_{t,a}(f)$$

and:

$$\beta_t^{\Leftarrow}(u_t, 1)_a = \sum_{f \in I(1, u_t): f(u_t)=1} \iota_{f(1)} \Omega_{t,a}(f).$$

Also note that by Lemma 17 we have, for all $i \in \{0,1\}$, that $\alpha_t(u_t, 1, i)_a = \tau_{1,i} \xi_{t,u_t,a}$. Hence we have, by the prediction algorithm and definition of $\Omega_{t,a}$, that:

$$K\bar{p}_{t,a} := \sum_{i \in \{0,1\}} \beta_t^{\Leftarrow}(u_t, 1)_a \alpha_t(u_t, 1, i)_a \beta_t^{\Rightarrow}(u_t, i)_a$$

$$= \beta_t^{\leftarrow}(u_t,1)_a \sum_{i\in\{0,1\}} \alpha_t(u_t,1,i)_a \beta_t^{\rightarrow}(u_t,i)_a$$

$$= \beta_t^{\leftarrow}(u_t,1)_a \sum_{i\in\{0,1\}} \tau_{1,i}\xi_{t,u_t,a} \sum_{f\in I(u_t+1,N+1):f(u_t+1)=i} \Omega_{t,a}(f)$$

$$= \beta_t^{\leftarrow}(u_t,1)_a \sum_{i\in\{0,1\}} \sum_{f\in I(u_t+1,N+1):f(u_t+1)=i} (\tau_{1,i}\xi_{t,u_t,a})\Omega_{t,a}(f)$$

$$= \beta_t^{\leftarrow}(u_t,1)_a \sum_{i\in\{0,1\}} \sum_{f\in I(u_t,N+1):f(u_t)=1,f(u_t+1)=i} \Omega_{t,a}(f)$$

$$= \beta_t^{\leftarrow}(u_t,1)_a \sum_{f\in I(u_t,N+1):f(u_t)=1} \Omega_{t,a}(f)$$

$$= \left( \sum_{f\in I(1,u_t):f(u_t)=1} \iota_{f(1)}\Omega_{t,a}(f) \right) \left( \sum_{f\in I(u_t,N+1):f(u_t)=1} \Omega_{t,a}(f) \right)$$

$$= \left( \sum_{i\in\{0,1\}} \sum_{f\in I(1,u_t):f(1)=i,f(u_t)=1} \iota_{f(1)}\Omega_{t,a}(f) \right) \left( \sum_{j\in\{0,1\}} \sum_{f\in I(u_t,N+1):f(u_t)=1,f(N+1)=j} \Omega_{t,a}(f) \right)$$

$$= \left( \sum_{i\in\{0,1\}} \iota_i \sum_{f\in I(1,u_t):f(1)=i,f(u_t)=1} \Omega_{t,a}(f) \right) \left( \sum_{j\in\{0,1\}} \sum_{f\in I(u_t,N+1):f(u_t)=1,f(N+1)=j} \Omega_{t,a}(f) \right)$$

$$= \sum_{i\in\{0,1\}} \sum_{j\in\{0,1\}} \iota_i \left( \sum_{f\in I(1,u_t):f(1)=i,f(u_t)=1} \Omega_{t,a}(f) \right) \left( \sum_{f\in I(u_t,N+1):f(u_t)=1,f(N+1)=j} \Omega_{t,a}(f) \right).$$

By Lemma 16 we then have:

$$K\bar{p}_{t,a} = \sum_{i\in\{0,1\}} \sum_{j\in\{0,1\}} \iota_i \sum_{f\in I(1,N+1):f(1)=i,f(u_t)=1,f(N+1)=j} \Omega_{t,a}(f)$$

$$= \sum_{i\in\{0,1\}} \sum_{j\in\{0,1\}} \sum_{f\in I(1,N+1):f(1)=i,f(u_t)=1,f(N+1)=j} \iota_{f(1)}\Omega_{t,a}(f)$$

$$= \sum_{f\in I(1,N+1):f(u_t)=1} \iota_{f(1)}\Omega_{t,a}(f)$$

as required. $\qquad\square$

**Lemma 53.** *Given $t \in [T]$, $a \in [K]$, $i \in \{0,1\}$ and $U \in \mathcal{E}$, then if $f \in I(1,N+1)$ is defined so that for all $u \in [N]$ we have $f(u) := [\![u \in \mathcal{E}]\!]$ and we have $f(N+1) := i$ then:*

$$\iota_{f(1)}\Omega_{t,a}(f) = K\bar{\kappa}_t(a,U)\tau_{[\![N\in U]\!],i}.$$

*Proof.* We prove by induction on $t$. For $t = 1$ we have, for all $u \in [N]$, that $\xi_{1,u,a} = 1$ so:

$$\iota_{f(1)}\Omega_{1,a}(f) = \iota_{f(1)} \prod_{u\in[N]} \tau_{f(u),f(u+1)}\xi_{1,u,a}$$

$$= \iota_{f(1)} \prod_{u\in[N]} \tau_{f(u),f(u+1)}$$

$$= \tau_{f(N),f(N+1)}\iota_{f(1)} \prod_{u\in[N-1]} \tau_{f(u),f(u+1)}$$

$$= \tau_{[\![N\in U]\!],i}\iota_{[\![1\in U]\!]} \prod_{u\in[N-1]} \tau_{[\![u\in\mathcal{E}]\!],[\![u+1\in\mathcal{E}]\!]}$$

$$= \tau_{[\![N\in U]\!],i}\sigma(U)$$

$$= \tau_{[\![N\in U]\!],i}K\bar{\kappa}_1(a,U)$$

as required. Now suppose the inductive hypothesis holds for $t = s$ (for some $s \in [T]$). We now show that it holds for $t = s + 1$, completing the proof. From the fact that $\xi_{s+1,u,a} = \xi_{s,u,a}$ for all $u \neq u_s$ we have:

$$
\begin{aligned}
\frac{\Omega_{s+1,a}(f)}{\Omega_{s,a}(f)} &= \frac{[\![f(u_s) = 0]\!] + [\![f(u_s) = 1]\!]\xi_{s+1,u_s,a}}{[\![f(u_s) = 0]\!] + [\![f(u_s) = 1]\!]\xi_{s,u_s,a}} \\
&= \frac{[\![u_s \notin U]\!] + [\![u_s \in U]\!]\xi_{s+1,u_s,a}}{[\![u_s \notin U]\!] + [\![u_s \in U]\!]\xi_{s,u_s,a}} \,,
\end{aligned}
\tag{17}
$$

so we have two cases:

- If $u_s \notin U$ then, by Equation 17, $\Omega_{s+1,a}(f)/\Omega_{s,a}(f) = 1$ so $\iota_{f(1)}\Omega_{s+1,a}(f) = \iota_{f(1)}\Omega_{s,a}(f)$ which, by the inductive hypothesis, is equal to $K\bar{\kappa}_s(a, U)\tau_{[\![N \in U]\!],i}$. From the definition of $\bar{\kappa}_{s+1}(a, U)$ we have $\bar{\kappa}_{s+1}(a, U) := \bar{\kappa}_s(a, U)$ so $\iota_{f(1)}\Omega_{s+1,a}(f) = K\bar{\kappa}_{s+1}(a, U)\tau_{[\![N \in U]\!],i}$ as required.

- If $u_s \in U$ then, by Equation 17, $\Omega_{s+1,a}(f)/\Omega_{s,a}(f) = \xi_{s+1,u_s,a}/\xi_{s,u_s,a} = \pi_{s,a}$ so $\iota_{f(1)}\Omega_{s+1,a}(f) = \pi_{s,a}\iota_{f(1)}\Omega_{s,a}(f)$ which, by the inductive hypothesis, is equal to $K\pi_{s,a}\bar{\kappa}_s(a, U)\tau_{[\![N \in U]\!],i}$. From the definition of $\bar{\kappa}_{s+1}(a, U)$ we have $\bar{\kappa}_{s+1}(a, U) := \pi_{s,a}\bar{\kappa}_s(a, U)$ so $\iota_{f(1)}\Omega_{s+1,a}(f) = K\bar{\kappa}_{s+1}(a, U)\tau_{[\![N \in U]\!],i}$ as required.

$\square$

By lemmas 52 and 53 we have:

$$
\begin{aligned}
\bar{p}_{t,a} &= \frac{1}{K} \sum_{f \in I(1,N+1):f(u_t)=1} \iota_{f(1)}\Omega_{t,a}(f) \\
&= \frac{1}{K} \sum_{i \in \{0,1\}} \sum_{f \in I(1,N+1):f(u_t)=1,f(N+1)=i} \iota_{f(1)}\Omega_{t,a}(f) \\
&= \frac{1}{K} \sum_{i \in \{0,1\}} \sum_{U \in \mathcal{E}:u_t \in U} K\bar{\kappa}_t(a,U)\tau_{[\![N \in U]\!],i} \\
&= \sum_{i \in \{0,1\}} \sum_{U \in \mathcal{E}:u_t \in U} \bar{\kappa}_t(a,U)\tau_{[\![N \in U]\!],i} \\
&= \sum_{U \in \mathcal{E}:u_t \in U} \bar{\kappa}_t(a,U)(\tau_{[\![N \in U]\!],0} + \tau_{[\![N \in U]\!],1}) \\
&= \sum_{U \in \mathcal{E}:u_t \in U} \bar{\kappa}_t(a,U)
\end{aligned}
$$

as required. $\blacksquare$

### I.6  Proof of Lemma 20

**Result**. GABA-I implements PROTOGABA with $\mathcal{E}$ and $\sigma$ defined as in Definition 10.

**Proof:**

**Definition 54.** *For all $t \in [T]$ let $\kappa_t$, $\boldsymbol{p}_t$, $z_t$, and $\lambda_t$ be as defined in the PROTOGABA algorithm when run with $\mathcal{E}$ and $\sigma$ defined as in Definition 10, and assuming that, for all $t \in [T]$, we have that $a_t$ is equal to that selected by GABA-I.*

**Lemma 55.** *For all $t \in [T]$ and all $(a, U) \in [K] \times \mathcal{E}$ we have $\bar{\kappa}_t(a, U) = \kappa_t(a, U)$ and $\bar{p}_{t,a} = p_{t,a}$.*

*Proof.* We prove by induction on $t$. For $t = 1$ it is clear, from the definitions of $\bar{\kappa}_1(a, U)$ and $\kappa_1(a, U)$, that $\bar{\kappa}_1(a, U) := \kappa_1(a, U)$ so also, by Lemma 19, we have that:

$$\bar{p}_{1,a} := \sum_{U \in \mathcal{E}: u_1 \in U} \bar{\kappa}_1(a, U) = \sum_{U \in \mathcal{E}: u_1 \in U} \kappa_1(a, U) := p_{1,a},$$

so the result holds for $t = 1$. Now suppose the result holds for $t = s$ (for some $s \in [T]$). We will now show that it holds for $t = s + 1$, which will complete the proof. By the inductive hypothesis we have, for all $b \in [K]$:

$$\begin{aligned}
c_{s,b} &:= \exp\left(-\eta[\![b = a_s]\!]\ell_{s,a_s} \|\bar{p}_s\|_1 / \bar{p}_{s,b}\right) \\
&= \exp\left(-\eta[\![b = a_s]\!]\ell_{s,a_s} \|p_s\|_1 / p_{s,b}\right) \\
&= [\![b \neq a_s]\!] + [\![b = a_s]\!] \exp\left(-\eta\ell_{s,a_s} \|p_s\|_1 / p_{s,b}\right) \\
&= [\![b \neq a_s]\!] + [\![b = a_s]\!]\lambda_s,
\end{aligned}$$

so also, by the inductive hypothesis:

$$\begin{aligned}
\bar{p}_s \cdot c_s &= \sum_{b \in [K]} \bar{p}_{s,b} \cdot c_{s,b} \\
&= \sum_{b \in [K]} p_{s,b} \cdot c_{s,b} \\
&= \sum_{b \in [K]} p_{s,b}([\![b \neq a_s]\!] + [\![b = a_s]\!]\lambda_s) \\
&= \sum_{b \in [K] \setminus \{a_s\}} p_{s,b} + p_{s,a_s}\lambda_s \\
&= (\|p_s\|_1 - p_{s,a_s}) + p_{s,a_s}\lambda_s \\
&= \|p_s\|_1 - (1 - \lambda_s)p_{s,a_s} \\
&= \frac{\|p_s\|_1}{z_s},
\end{aligned}$$

and hence, again by the inductive hypothesis:

$$\begin{aligned}
\pi_{s,a} &:= \frac{\|\bar{p}_s\|_1 c_{s,a}}{\bar{p}_s \cdot c_s} \\
&= \frac{\|p_s\|_1 c_{s,a}}{\bar{p}_s \cdot c_s} \\
&= z_s c_{s,a} \\
&= z_s([\![a \neq a_s]\!] + [\![a = a_s]\!]\lambda_s).
\end{aligned}$$

By the inductive hypothesis and definitions of $\bar{\kappa}_{s+1}(a, U)$ and $\kappa_{s+1}(a, U)$ we then have:

$$\begin{aligned}
\bar{\kappa}_{s+1}(a, U) &:= [\![u_s \notin U]\!]\bar{\kappa}_s(a, U) + [\![u_s \in U]\!]\pi_{s,a}\bar{\kappa}_s(a, U) \\
&= [\![u_s \notin U]\!]\kappa_s(a, U) + [\![u_s \in U]\!]\pi_{s,a}\kappa_s(a, U) \\
&= [\![u_s \notin U]\!]\kappa_s(a, U) + [\![u_s \in U]\!]z_s([\![a \neq a_s]\!] + [\![a = a_s]\!]\lambda_s)\kappa_s(a, U) \\
&:= \kappa_{s+1}(a, U).
\end{aligned}$$

By Lemma 19, we then have that:

$$\bar{p}_{s+1,a} := \sum_{U \in \mathcal{E}: u_{s+1} \in U} \bar{\kappa}_{s+1}(a, U) = \sum_{U \in \mathcal{E}: u_{s+1} \in U} \kappa_{s+1}(a, U) := p_{s+1,a}$$

as required. This completes the proof. $\qquad\square$

Since, by Lemma 55, we have $\bar{p}_t = p_t$ we then have, by the prediction algorithm, that $a_t$ is drawn with probability $\mathbb{P}[a_t = a] = p_{t,a}/\|p_t\|_1$ which implies the result. $\qquad\blacksquare$

## I.7 Proof of Lemma 22

**Result**. We have that:

- For all $u \in [N]$ there exists a unique pair $(a, U) \in \mathcal{A}$ with $u \in U$.
- For all $u \in [N]$ and $(a, U) \in \mathcal{A}$ with $u \in U$, we have that $a = y(u)$.

**Proof:**

For all $u \in [N]$ and $a \in [K]$ we have that $u \in \{v \in [N] \mid y(v) = a\}$ if and only if $a = y(u)$ so there exists an unique pair $(b, U) \in \mathcal{A}$ with $u \in U$ and furthermore this pair satisfies $b = y(u)$ as required. ∎

## I.8 Proof of Theorem 23

**Result**. Given $\Psi \leq (N - 1)/4$ and the parameters are tuned as:

$$\phi := 4\Psi/(K(N - 1)),$$

and:

$$\eta := \sqrt{\frac{10\Psi \ln(KN/\Psi)}{KT}},$$

we have, for GABA-I:

$$R \in \mathcal{O}\left(\sqrt{\ln\left(\frac{KN}{\Psi}\right)\Psi KT}\right).$$

**Proof:**

**Definition 56.** *Define:*

$$\Gamma := \sum_{u \in [N-1]} [\![y(u) \neq y(u+1)]\!],$$

*and for all $a \in [K]$ and $u \in [N]$ define:*

$$\epsilon(a, u) = [\![y(u) = a]\!].$$

**Lemma 57.** *We have:*

$$R \leq \frac{1}{\eta}\mathbb{E}\left[\sum_{a \in [K]} \ln\left(\frac{1}{\sigma(\{u \in [N] \mid y(u) = a\})}\right)\right] + \frac{\eta KT}{2}.$$

*Proof.* Lemmas 20 and 22 allow us to invoke Theorem 9 so, noting that $|\mathcal{A}| = K$, we have:

$$R \leq \frac{1}{\eta} \sum_{(a,U) \in \mathcal{A}} \ln\left(\frac{K}{|\mathcal{A}|\sigma(U)}\right) + \frac{\eta KT}{2}$$

$$= \frac{1}{\eta} \sum_{(a,U) \in \mathcal{A}} \ln\left(\frac{1}{\sigma(U)}\right) + \frac{\eta KT}{2}$$

$$= \frac{1}{\eta} \sum_{(a,U) \in \mathcal{A}} \ln\left(\frac{1}{\sigma(\{u \in [N] \mid y(u) = a\})}\right) + \frac{\eta KT}{2}$$

$$= \frac{1}{\eta} \sum_{a \in [K]} \ln\left(\frac{1}{\sigma(\{u \in [N] \mid y(u) = a\})}\right) + \frac{\eta KT}{2}.$$

Taking expectations then gives us the result. □

**Lemma 58.** *We have:*

$$\sum_{a \in [K]} \ln(\iota_{\epsilon(a,1)}) = \ln\left(\frac{1}{K}\right) + (K - 1)\ln\left(1 - \frac{1}{K}\right).$$

*Proof.* We have $\epsilon(y(1), 1) = [\![y(1) = y(1)]\!] = 1$ so $\iota_{\epsilon(y(1),1)} = 1/K$. Also, for all $a \in [K] \setminus \{y(1)\}$, we have $\epsilon(a, 1) = [\![y(1) = a]\!] = 0$ so $\iota_{\epsilon(a,1)} = 1 - 1/K$. Hence, we have:

$$\sum_{a \in [K]} \ln(\iota_{\epsilon(a,1)}) = \ln(\iota_{\epsilon(y(1),1)}) + \sum_{a \in [K] \setminus \{y(1)\}} \ln(\iota_{\epsilon(a,1)})$$

$$= \ln\left(\frac{1}{K}\right) + \sum_{a \in [K] \setminus \{y(1)\}} \ln\left(1 - \frac{1}{K}\right)$$

$$= \ln\left(\frac{1}{K}\right) + (K - 1) \ln\left(1 - \frac{1}{K}\right)$$

as required. $\qquad\qquad\square$

**Lemma 59.** *For all $u \in [N - 1]$ with $y(u) = y(u + 1)$ we have:*

$$\sum_{a \in [K]} \ln\left(\tau_{\epsilon(a,u),\epsilon(a,u+1)}\right) = K \ln(1 - \phi).$$

*Proof.* We have, for all $a \in [K]$, that $\epsilon(a, u) = [\![y(u) = a]\!] = [\![y(u + 1) = a]\!] = \epsilon(a, u + 1)$ so $\tau_{\epsilon(a,u),\epsilon(a,u+1)} = 1 - \phi$. This implies that:

$$\sum_{a \in [K]} \ln(\tau_{\epsilon(a,u),\epsilon(a,u+1)}) = \sum_{a \in [K]} \ln(1 - \phi) = K \ln(1 - \phi)$$

as required. $\qquad\qquad\square$

**Lemma 60.** *For all $u \in [N - 1]$ with $y(u) \neq y(u + 1)$ we have:*

$$\sum_{a \in [K]} \ln\left(\tau_{\epsilon(a,u),\epsilon(a,u+1)}\right) = (K - 2) \ln(1 - \phi) + 2 \ln(\phi).$$

*Proof.* We have that:

$$\epsilon(y(u), u) = [\![y(u) = y(u)]\!] = 1 \neq 0 = [\![y(u + 1) = y(u)]\!] = \epsilon(y(u), u + 1),$$

and that:

$$\epsilon(y(u + 1), u) = [\![y(u + 1) = y(u)]\!] = 0 \neq 1 = [\![y(u + 1) = y(u + 1)]\!] = \epsilon(y(u + 1), u + 1).$$

so $\tau_{\epsilon(y(u),u),\epsilon(y(u),u+1)} = \phi$ and $\tau_{\epsilon(y(u+1),u),\epsilon(y(u+1),u+1)} = \phi$. We also have, for all $a \in [K] \setminus \{y(u), y(u + 1)\}$, that:

$$\epsilon(a, u) = [\![y(u) = a]\!] = 0 = [\![y(u + 1) = a]\!] = \epsilon(a, u + 1),$$

so $\tau_{\epsilon(a,u),\epsilon(a,u+1)} = 1 - \phi$. Combining these equalities gives us:

$$\sum_{a \in [K]} \ln\left(\tau_{\epsilon(a,u),\epsilon(a,u+1)}\right)$$

$$= \ln\left(\tau_{\epsilon(y(u),u),\epsilon(y(u),u+1)}\right) + \ln\left(\tau_{\epsilon(y(u+1),u),\epsilon(y(u+1),u+1)}\right) + \sum_{a \in [K] \setminus \{y(u),y(u+1)\}} \ln\left(\tau_{\epsilon(a,u),\epsilon(a,u+1)}\right)$$

$$= \ln(\phi) + \ln(\phi) + \sum_{a \in [K] \setminus \{y(u),y(u+1)\}} \ln(1 - \phi)$$

$$= 2 \ln(\phi) + (K - 2) \ln(1 - \phi)$$

as required. $\qquad\qquad\square$

**Lemma 61.** *We have:*

$$\sum_{u \in [N-1]} \sum_{a \in [K]} \ln(\tau_{\epsilon(a,u),\epsilon(a,u+1)}) = 2\Gamma \ln(\phi) + (K(N - 1) - 2\Gamma) \ln(1 - \phi).$$

*Proof.* From lemmas 59 and 60 we have:

$$\sum_{u\in[N-1]}\sum_{a\in[K]}\ln(\tau_{\epsilon(a,u),\epsilon(a,u+1)})$$

$$=\sum_{u\in[N-1]:y(u)=y(u+1)}\sum_{a\in[K]}\ln(\tau_{\epsilon(a,u),\epsilon(a,u+1)})+\sum_{u\in[N-1]:y(u)\neq y(u+1)}\sum_{a\in[K]}\ln(\tau_{\epsilon(a,u),\epsilon(a,u+1)})$$

$$=\sum_{u\in[N-1]:y(u)=y(u+1)}K\ln(1-\phi)+\sum_{u\in[N-1]:y(u)\neq y(u+1)}((K-2)\ln(1-\phi)+2\ln(\phi))$$

$$=(N-1-\Gamma)K\ln(1-\phi)+\Gamma((K-2)\ln(1-\phi)+2\ln(\phi))$$

$$=((N-1-\Gamma)K+\Gamma(K-2))\ln(1-\phi)+2\Gamma\ln(\phi)$$

$$=((N-1)K-2\Gamma)\ln(1-\phi)+2\Gamma\ln(\phi)$$

as required. □

**Lemma 62.** *Given $\phi \leq 1/2$ we have:*

$$\mathbb{E}\left[\sum_{u\in[N-1]}\sum_{a\in[K]}\ln(\tau_{\epsilon(a,u),\epsilon(a,u+1)})\right]\geq 4\Psi\ln(\phi)+(K(N-1)-4\Psi)\ln(1-\phi).$$

*Proof.* Since $0 < \phi \leq 1/2$ and hence $1-\phi \geq \phi$ we have $\ln(\phi)-\ln(1-\phi)\leq 0$ so since, by Lemma 2, we have $\mathbb{E}\left[\Gamma\right]\leq 2\Psi$, we must have:

$$(\ln(\phi)-\ln(1-\phi))\mathbb{E}\left[\Gamma\right]\geq 2(\ln(\phi)-\ln(1-\phi))\Psi.$$

So, from Lemma 61, we have:

$$\mathbb{E}\left[\sum_{u\in[N-1]}\sum_{a\in[K]}\ln(\tau_{\epsilon(a,u),\epsilon(a,u+1)})\right]=\mathbb{E}\left[2\Gamma\ln(\phi)+(K(N-1)-2\Gamma)\ln(1-\phi)\right]$$

$$=\mathbb{E}\left[K(N-1)\ln(1-\phi)+2\Gamma(\ln(\phi)-\ln(1-\phi))\right]$$

$$=K(N-1)\ln(1-\phi)+2(\ln(\phi)-\ln(1-\phi))\mathbb{E}\left[\Gamma\right]$$

$$\geq K(N-1)\ln(1-\phi)+4(\ln(\phi)-\ln(1-\phi))\Psi$$

$$=4\Psi\ln(\phi)+(K(N-1)-4\Psi)\ln(1-\phi)$$

as required. □

**Lemma 63.** *Given $f, f' \in \mathbb{R}^+$, if we set $f^\dagger := f/(f+f')$ then:*

$$-f\ln(f^\dagger)-f'\ln(1-f^\dagger)\leq f\ln\left(\frac{1}{f^\dagger}\right)+f.$$

*Proof.* We recall the following standard inequality about the binary entropy of $f^\dagger$:

$$-f^\dagger\ln(f^\dagger)-(1-f^\dagger)\ln(1-f^\dagger)\leq f^\dagger\ln(1/f^\dagger)+f^\dagger.$$

Using this and the fact that $f = (f+f')f^\dagger$ and $f' = (f+f')(1-f^\dagger)$ gives us:

$$-f\ln(f^\dagger)-f'\ln(1-f^\dagger)=(f+f')(-f^\dagger\ln(f^\dagger)-(1-f^\dagger)\ln(1-f^\dagger))$$

$$\leq(f+f')(f^\dagger\ln(1/f^\dagger)+f^\dagger)$$

$$=(f+f')\left(\frac{f}{f+f'}\ln\left(\frac{1}{f^\dagger}\right)+\frac{f}{f+f'}\right)$$

$$=f\ln\left(\frac{1}{f^\dagger}\right)+f$$

as required. □

**Lemma 64.** *We have:*

$$-\sum_{a\in[K]}\ln(\iota_{\epsilon(a,1)})\le\ln(K)+1\,.$$

*Proof.* Direct from lemmas 58 and 63 with $f:=1$, $f':=K-1$ and $f^\dagger=1/K$. $\qquad\square$

**Lemma 65.** *Given $\phi:=4\Psi/(K(N-1))\le 1/2$ and $\Psi>0$, we have:*

$$-\mathbb{E}\left[\sum_{u\in[N-1]}\sum_{a\in[K]}\ln(\tau_{\epsilon(a,u),\epsilon(a,u+1)})\right]\le 4\Psi\ln\left(\frac{KN}{\Psi}\right)\,.$$

*Proof.* Direct from lemmas 62 and 63 with $f:=4\Psi$, $f':=(K(N-1)-4\Psi)$ and $f^\dagger:=4\Psi/(K(N-1))=\phi$, we have:

$$-\mathbb{E}\left[\sum_{u\in[N-1]}\sum_{a\in[K]}\ln(\tau_{\epsilon(a,u),\epsilon(a,u+1)})\right]\le 4\Psi\ln\left(\frac{K(N-1)}{4\Psi}\right)+4\Psi$$

$$\le 4\Psi\ln\left(\frac{K(N-1)}{\Psi}\right)+(1-\ln(4))4\Psi$$

$$\le 4\Psi\ln\left(\frac{K(N-1)}{\Psi}\right)$$

$$\le 4\Psi\ln\left(\frac{KN}{\Psi}\right)$$

as required. $\qquad\square$

**Lemma 66.** *Given $\phi:=4\Psi/(K(N-1))\le 1/2$ and $0<\Psi\le(N-1)/4$ we have:*

$$\mathbb{E}\left[\sum_{a\in[K]}\ln\left(\frac{1}{\sigma(\{u\in[N]\mid y(u)=a\})}\right)\right]\le 5\Psi\ln\left(\frac{KN}{\Psi}\right)\,.$$

*Proof.* For all $a\in[K]$, we have, from the definition of $\sigma$, that:

$$\ln(\sigma(\{u\in[N]\mid y(u)=a\}))$$

$$=\ln(\iota_{[\![1\in\{v\in[N]\mid y(v)=a\}]\!]})+\sum_{u\in[N-1]}\ln(\tau_{[\![u\in\{v\in[N]\mid y(v)=a\}]\!],[\![u+1\in\{v\in[N]\mid y(v)=a\}]\!]})$$

$$=\ln(\iota_{[\![y(1)=a]\!]})+\sum_{u\in[N-1]}\ln(\tau_{[\![y(u)=a]\!],[\![y(u+1)=a]\!]})$$

$$=\ln(\iota_{\epsilon(a,1)})+\sum_{u\in[N-1]}\ln(\tau_{\epsilon(a,u),\epsilon(a,u+1)})\,,$$

so by lemmas 64 and 65 we have:

$$\mathbb{E}\left[\sum_{a\in[K]}\ln\left(\frac{1}{\sigma(\{u\in[N]\mid y(u)=a\})}\right)\right]$$

$$=-\sum_{a\in[K]}\mathbb{E}\left[\ln(\sigma(\{u\in[N]\mid y(u)=a\}))\right]$$

$$=-\sum_{a\in[K]}\mathbb{E}\left[\ln(\iota_{\epsilon(a,1)})+\sum_{u\in[N-1]}\ln(\tau_{\epsilon(a,u),\epsilon(a,u+1)})\right]$$

$$=-\sum_{a\in[K]}\ln(\iota_{\epsilon(a,1)})-\mathbb{E}\left[\sum_{u\in[N-1]}\sum_{a\in[K]}\ln(\tau_{\epsilon(a,u),\epsilon(a,u+1)})\right]$$

$$\leq (\ln(K) + 1) + 4\Psi \ln\left(\frac{KN}{\Psi}\right)$$

$$= \ln(eK) + 4\Psi \ln\left(\frac{KN}{\Psi}\right) .$$

Since we have $\Psi \leq (N-1)/4 < N/e$ we also have that $KN/\Psi > eK$ so $\ln(eK) < \ln(KN/\Psi)$. Substituting this into the above inequality gives us the result. $\qquad\square$

Since $\Psi \leq (N-1)/4$ implies that $4\Psi/(K(N-1)) \leq 1/2$, combining lemmas 57 and 66 gives us the fact that if $0 \leq \Psi \leq (N-1)/4$ and $\phi := 4\Psi/(K(N-1))$, we have:

$$R \leq \frac{5}{\eta}\Psi \ln\left(\frac{KN}{\Psi}\right) + \frac{\eta KT}{2} ,$$

so, choosing:

$$\eta := \sqrt{\frac{10\Psi \ln(KN/\Psi)}{KT}}$$

we have:

$$R \leq \sqrt{10\Psi \ln\left(\frac{KN}{\Psi}\right) KT}$$

as required. $\qquad\blacksquare$

## J   GABA-II Proofs

### J.1   Proof of Lemma 29

**Result.** Given $t \in [T]$, $n \in \mathcal{C}$ and $a \in [K]$ we have:

$$\bar{\kappa}_t(a, n) = \frac{\mu_t(n)\theta_t(n, a)}{(2N-1)K} .$$

**Proof:**

We prove by induction on $t$. For $t = 1$ we have $\mu_1(n) := 1$ and $\theta_1(n, a) := 1$ so:

$$\bar{\kappa}_1(a, n) := \frac{1}{(2N-1)K} = \frac{\mu_1(n)\theta_1(n, a)}{(2N-1)K}$$

as required. Now suppose the result holds for $t = s$ (for some $s \in [T]$). We now show that it holds for $t = s + 1$, completing the proof. We first note that by the update algorithm we have, for all $n \in \Uparrow(u_s)$:

$$\mu_{s+1}(n) := \mu_s(n)\frac{\psi_s}{\psi_s - (1 - \bar{\lambda}_s)\varrho_s} = \bar{z}_s\mu_s(n) ,$$

whilst for $n \notin \Uparrow(u_s)$ we have:

$$\mu_{s+1}(n) := \mu_s(n) ,$$

so in general:

$$\mu_{s+1}(n) = (\llbracket n \notin \Uparrow(u_s) \rrbracket + \llbracket n \in \Uparrow(u_s) \rrbracket \bar{z}_s)\mu_s(n) .$$

We note that also, by the update algorithm, we have, for all $n \in \Uparrow(u_s)$, that if $a = a_t$ then

$$\theta_{s+1}(n, a) := \bar{\lambda}_t\theta_s(n, a) ,$$

whilst if $n \notin \Uparrow(u_s)$ or $a \neq a_t$ we have that:

$$\theta_{s+1}(n, a) := \theta_s(n, a) ,$$

so in general:

$$\theta_{s+1}(n, a) = (\llbracket n \notin \Uparrow(u_s) \rrbracket + \llbracket n \in \Uparrow(u_s) \wedge a \neq a_t \rrbracket + \llbracket n \in \Uparrow(u_s) \wedge a = a_t \rrbracket \bar{\lambda}_t)\theta_s(n, a) .$$

Multiplying these two equalities and invoking the inductive hypothesis gives us:

$$\mu_{s+1}(n)\theta_{s+1}(n,a)$$
$$= (\llbracket n \notin \Uparrow(u_s)\rrbracket + \llbracket n \in \Uparrow(u_s) \wedge a \neq a_t\rrbracket \bar{z}_s + \llbracket n \in \Uparrow(u_s) \wedge a = a_t\rrbracket \bar{z}_s \bar{\lambda}_t)\mu_s(n)\theta_s(n,a)$$
$$= (\llbracket n \notin \Uparrow(u_s)\rrbracket + \llbracket n \in \Uparrow(u_s) \wedge a \neq a_t\rrbracket \bar{z}_s + \llbracket n \in \Uparrow(u_s) \wedge a = a_t\rrbracket \bar{z}_s \bar{\lambda}_t)\bar{\kappa}_s(a,n)(2N-1)K \,,$$

which, by the definition of $\bar{\kappa}_{s+1}(a,n)$ is equal to $\bar{\kappa}_{s+1}(a,n)(2N-1)K$ as required. This completes the proof. ∎

## J.2 Proof of Lemma 30

**Result.** Given $t \in [T]$, $n \in \mathcal{C}$ and $m \in \mathcal{B}$ we have:

$$\theta_t(n,m) = \sum_{a \in \Downarrow(m) \cap [K]} \theta_t(n,a)\,.$$

**Proof:**

We prove by induction on $t$. For $t = 1$ we have the result directly from the initialization algorithm. Now suppose the result holds for $t = s$ (for some $s \in [T]$). We now show that the result holds for $t = s + 1$ which will complete the proof. We first consider the case that $m$ is a leaf of $\mathcal{B}$. We have two cases:

- If $m \in [K]$ then $\Downarrow(m) \cap [K] = \{m\}$ so the result is immediate.
- If $m \notin [K]$ then $\Downarrow(m) \cap [K] = \emptyset$ so by the inductive hypothesis we have:

$$\theta_s(n,m) = \sum_{a \in \Downarrow(m) \cap [K]} \theta_s(n,a)$$
$$= 0$$
$$= \sum_{a \in \Downarrow(m) \cap [K]} \theta_{s+1}(n,a)\,.$$

  From the update algorithm we have (since $m$ is not an ancestor of $a_s$) that $\theta_{s+1}(n,m) := \theta_s(n,m)$, which then gives us the result.

So the result holds in the case that $m$ is a leaf of $\mathcal{B}$. Now suppose that $m$ is an internal vertex of $\mathcal{B}$. We have two cases:

- If $m \notin \Uparrow(a_t)$ or $n \notin \Uparrow(u_t)$ we have, from the update algorithm, that $\theta_{s+1}(n,m) := \theta_s(n,m)$ and, since also either $n \notin \Uparrow(u_t)$ or both children of $m$ are not in $\Uparrow(a_t)$ we also have, from the update algorithm, that $\theta_{s+1}(n, \triangleleft(m)) := \theta_s(n, \triangleleft(m))$ and $\theta_{s+1}(n, \triangleright(m)) := \theta_s(n, \triangleright(m))$ so by the inductive hypothesis:

$$\theta_{s+1}(n,m) := \theta_s(n,m)$$
$$= \sum_{a \in \Downarrow(m) \cap [K]} \theta_s(n,a)$$
$$= \sum_{a \in \Downarrow(\triangleleft(m)) \cap [K]} \theta_s(n,a) + \sum_{a \in \Downarrow(\triangleright(m)) \cap [K]} \theta_s(n,a)$$
$$= \theta_s(n, \triangleleft(m)) + \theta_s(n, \triangleright(m))$$
$$= \theta_{s+1}(n, \triangleleft(m)) + \theta_{s+1}(n, \triangleright(m))$$
$$= \sum_{a \in \Downarrow(\triangleleft(m)) \cap [K]} \theta_{s+1}(n,a) + \sum_{a \in \Downarrow(\triangleright(m)) \cap [K]} \theta_{s+1}(n,a)$$
$$= \sum_{a \in \Downarrow(m) \cap [K]} \theta_{s+1}(n,a)$$

  as required.

- If $m \in \Uparrow(a_t)$ and $n \in \Uparrow(u_t)$ then $m = \zeta_{t,g-d}$ for some $d \in [g]$ so we prove by induction on $d$. For $d = 1$ we note that $\triangleleft(m)$ and $\triangleright(m)$ are both leaves and we proved above that the result held for leaves. Hence, by the update algorithm, we have:

$$\theta_{s+1}(n, m) := \theta_{s+1}(n, \triangleleft(m)) + \theta_{s+1}(n, \triangleright(m))$$

$$= \sum_{a \in \Downarrow(\triangleleft(m)) \cap [K]} \theta_{s+1}(n, a) + \sum_{a \in \Downarrow(\triangleright(m)) \cap [K]} \theta_{s+1}(n, a)$$

$$= \sum_{a \in \Downarrow(m) \cap [K]} \theta_{s+1}(n, a)$$

as required. Now suppose the result holds for $d = q$ (for some $q \in [g]$). We now show that it holds for $d = q + 1$ which will complete the proof. We have that one of the children of $m$ is an ancestor of $a_t$ so without loss of generality assume this is its left child. By above we proved the result holds for all internal vertices that are not ancestors of $a_s$ so specifically the result holds for $\triangleright(m)$. Noting that $\triangleleft(m) = \zeta_{t,g-q}$ we also have, by the inductive hypothesis, that the result holds for $\triangleleft(m)$ so, by the update algorithm:

$$\theta_{s+1}(n, m) := \theta_{s+1}(n, \triangleleft(m)) + \theta_{s+1}(n, \triangleright(m))$$

$$= \sum_{a \in \Downarrow(\triangleleft(m)) \cap [K]} \theta_{s+1}(n, a) + \sum_{a \in \Downarrow(\triangleright(m)) \cap [K]} \theta_{s+1}(n, a)$$

$$= \sum_{a \in \Downarrow(m) \cap [K]} \theta_{s+1}(n, a)$$

as required

This completes the proof. ∎

### J.3 Proof of Lemma 31

**Result**. For all $t \in [T]$ and $a \in [K]$ we have:

$$\mathbb{P}\left[a_t = a\right] = \frac{\bar{p}_{t,a}}{\|\bar{p}_t\|_1} .$$

**Proof:**

**Lemma 67.** *For all trials $t \in [T]$ and all $d \in [g - 1] \cup \{0\}$ we have, given $\delta_t$ and $\zeta_{t,d}$, that:*

$$\mathbb{P}\left[\zeta_{t,d+1} = m\right] = \frac{\theta_t(\delta_t, m)}{\theta_t(\delta_t, \uparrow(m))} ,$$

*for $m \in \{\triangleleft(\zeta_{t,d}), \triangleright(\zeta_{t,d})\}$.*

*Proof.* From the algorithm we have $\mathbb{P}\left[\zeta_{t,d+1} = m\right] \propto \theta_t(\delta_t, m)$ for $m \in \{\triangleleft(\zeta_{t,d}), \triangleright(\zeta_{t,d})\}$ so $\mathbb{P}\left[\zeta_{t,d+1} = m\right] = \theta_t(\delta_t, m)/Z$ where $Z := \theta_t(\delta_t, \triangleleft(\zeta_{t,d})) + \theta_t(\delta_t, \triangleright(\zeta_{t,d}))$ By Lemma 30, we then have:

$$Z := \theta_t(\delta_t, \triangleleft(\zeta_{t,d})) + \theta_t(\delta_t, \triangleright(\zeta_{t,d}))$$

$$= \sum_{a \in \Downarrow(\triangleleft(\zeta_{t,d})) \cap [K]} \theta_t(\delta_t, a) + \sum_{a \in \Downarrow(\triangleright(\zeta_{t,d})) \cap [K]} \theta_t(\delta_t, a)$$

$$= \sum_{a \in \Downarrow(\zeta_{t,d}) \cap [K]} \theta_t(\delta_t, a)$$

$$= \theta_t(\delta_t, \zeta_{t,d})$$

$$= \theta_t(\delta_t, \uparrow(m))$$

as required. ∎

**Lemma 68.** *For all $n \in \mathcal{C}$ and all $m \in \mathcal{B}$ at depth $d \in [g] \cup \{0\}$ we have:*

$$\mathbb{P}\left[\zeta_{t,d} = m \mid \delta_t = n\right] = \frac{\theta_t(n,m)}{\theta_t(n,r)} \,.$$

*Proof.* We prove by induction on $d$. For $d = 0$ we have $m = r$ and always $\zeta_{t,0} = r$ so:

$$\mathbb{P}\left[\zeta_{t,0} = m \mid \delta_t = n\right] = 1 = \frac{\theta_t(n,r)}{\theta_t(n,r)} = \frac{\theta_t(n,m)}{\theta_t(n,r)}$$

as required. Now assume the result holds for $d = q$ (for some $q \in [g-1] \cup \{0\}$). We now show it holds for $d = q+1$ which completes the proof. We have, by the inductive hypothesis and Lemma 67, that:

$$\mathbb{P}\left[\zeta_{t,q+1} = m \mid \delta_t = n\right] = \mathbb{P}\left[\zeta_{t,q+1} = m \mid \delta_t = n \wedge \uparrow(m) = \zeta_{t,q}\right] \mathbb{P}\left[\zeta_{t,q} = \uparrow(m) \mid \delta_t = n\right]$$

$$= \mathbb{P}\left[\zeta_{t,q+1} = m \mid \delta_t = n \wedge \uparrow(m) = \zeta_{t,d}\right] \frac{\theta_t(n,\uparrow(m))}{\theta_t(n,r)}$$

$$= \frac{\theta_t(n,m)}{\theta_t(n,\uparrow(m))} \frac{\theta_t(n,\uparrow(m))}{\theta_t(n,r)}$$

$$= \frac{\theta_t(n,m)}{\theta_t(n,r)}$$

as required. $\square$

By Lemma 68, the law of total probability, the prediction algorithm, and Lemma 29 we have:

$$\mathbb{P}\left[a_t = a\right] = \mathbb{P}\left[\zeta_{t,g} = a\right]$$

$$= \sum_{n \in \Uparrow(u_t)} \mathbb{P}\left[\delta_t = n\right] \mathbb{P}\left[\zeta_{t,g} = a \mid \delta_t = n\right]$$

$$= \sum_{n \in \Uparrow(u_t)} \mathbb{P}\left[\delta_t = n\right] \frac{\theta_t(n,a)}{\theta_t(n,r)}$$

$$\propto \sum_{n \in \Uparrow(u_t)} (\mu_t(n)\theta_t(n,r)) \frac{\theta_t(n,a)}{\theta_t(n,r)}$$

$$= \sum_{n \in \Uparrow(u_t)} \mu_t(n)\theta_t(n,a)$$

$$\propto \sum_{n \in \Uparrow(u_t)} \bar{\kappa}_t(a,n)$$

$$= \bar{p}_{t,a}$$

as required. $\blacksquare$

### J.4 Proof of Lemma 32

**Result**. For all $t \in [T]$ we have:

$$\bar{\lambda}_t = \exp\left(\frac{-\eta \ell_{t,a_t} \|\bar{\boldsymbol{p}}_t\|_1}{\bar{p}_{t,a_t}}\right),$$

and:

$$\bar{z}_t = \frac{\|\bar{\boldsymbol{p}}_t\|_1}{\|\bar{\boldsymbol{p}}_t\|_1 - (1 - \bar{\lambda}_t)\bar{p}_{t,a_t}}\,.$$

**Proof:**

**Lemma 69.** *We have:*

$$\varrho_t = (2N - 1)K\bar{p}_{t,a_t}\,.$$

*Proof.* From Lemma 29 and the update algorithm we have:

$$\varrho_t := \sum_{n \in \Uparrow(u_t)} \mu_t(n)\theta(n, a_t)$$

$$= \sum_{n \in \Uparrow(u_t)} \bar{\kappa}_t(a_t, n)(2N-1)K$$

$$= (2N-1)K \sum_{n \in \Uparrow(u_t)} \bar{\kappa}_t(a_t, n)$$

$$= (2N-1)K\bar{p}_{t,a_t}$$

as required. $\square$

**Lemma 70.** *We have:*

$$\psi_t = (2N-1)K\|\bar{\boldsymbol{p}}_t\|_1 \,.$$

*Proof.* From lemmas 29 and 30 and the update algorithm we have:

$$\psi_t := \sum_{n \in \Uparrow(u_t)} \mu_t(n)\theta(n, r)$$

$$= \sum_{n \in \Uparrow(u_t)} \mu_t(n) \sum_{a \in \Downarrow(r) \cap [K]} \theta(n, a)$$

$$= \sum_{n \in \Uparrow(u_t)} \mu_t(n) \sum_{a \in [K]} \theta(n, a)$$

$$= \sum_{a \in [K]} \sum_{n \in \Uparrow(u_t)} \mu_t(n)\theta(n, a)$$

$$= \sum_{a \in [K]} \sum_{n \in \Uparrow(u_t)} \bar{\kappa}_s(a, n)(2N-1)K$$

$$= (2N-1)K \sum_{a \in [K]} \sum_{n \in \Uparrow(u_t)} \bar{\kappa}_t(a, n)$$

$$= (2N-1)K \sum_{a \in [K]} \bar{p}_{t,a}$$

$$= (2N-1)K\|\bar{\boldsymbol{p}}_t\|_1$$

as required. $\square$

From lemmas 69 and 70 and the update algorithm we have:

$$\bar{\lambda}_t := \exp\left(\frac{-\eta\ell_{t,a_t}\psi_t}{\varrho_t}\right) = \exp\left(\frac{-\eta\ell_{t,a_t}\|\bar{\boldsymbol{p}}_t\|_1}{\bar{p}_{t,a_t}}\right) \,,$$

and:

$$\bar{z}_t := \frac{\psi_t}{\psi_t - (1 - \bar{\lambda}_t)\varrho_t} = \frac{\|\bar{\boldsymbol{p}}_t\|_1}{\|\bar{\boldsymbol{p}}_t\|_1 - (1 - \bar{\lambda}_t)\bar{p}_{t,a_t}}$$

as required. $\blacksquare$

### J.5   Proof of Lemma 33

**Result**. GABA-II implements PROTOGABA with $\mathcal{E}$ and $\sigma$ defined as in Definition 25

**Proof:**

**Definition 71.** *For all $t \in [T]$ let $\kappa_t$, $\boldsymbol{p}_t$, $z_t$, and $\lambda_t$ be as defined in the* PROTOGABA *algorithm when run with $\mathcal{E}$ and $\sigma$ defined as in Definition 25, and assuming that, for all $t \in [T]$, we have that $a_t$ is equal to that selected by* GABA-II.

**Lemma 72.** *For all $t \in [T]$, $a \in [K]$ and $n \in \mathcal{C}$ we have:*
$$\bar{\kappa}_t(a, n) = \kappa_t(a, \Downarrow(n)).$$

*Proof.* We prove by induction over $t$. For $t = 1$ we have:
$$\kappa_1(a, \Downarrow(n)) := \frac{\sigma(\Downarrow(n))}{K} = \frac{1}{(2N-1)K} := \bar{\kappa}_1(a, n)$$

as required. Now suppose the result holds for $t = s$ (for some $s \in [T]$). We now show that it holds for $t = s + 1$ which will complete the proof. Note first that, by the PROTOGABA algorithm and the inductive hypothesis, we have, for all $b \in [K]$, that:

$$\begin{aligned}
p_{s,b} &:= \sum_{U \in \mathcal{E}: u_s \in U} \kappa_s(b, U) \\
&= \sum_{n \in \mathcal{C}: u_s \in \Downarrow(n)} \kappa_s(b, \Downarrow(n)) \\
&= \sum_{n \in \Uparrow(u_s)} \kappa_s(b, \Downarrow(n)) \\
&= \sum_{n \in \Uparrow(u_s)} \bar{\kappa}_s(b, n) \\
&:= \bar{p}_{s,b},
\end{aligned}$$

so $\bar{\boldsymbol{p}}_s = \boldsymbol{p}_s$. By Lemma 32 we then have:
$$\bar{\lambda}_s = \exp\left(\frac{-\eta \ell_{s,a_s} \|\bar{\boldsymbol{p}}_s\|_1}{\bar{p}_{s,a_s}}\right) = \exp\left(\frac{-\eta \ell_{s,a_s} \|\boldsymbol{p}_s\|_1}{p_{s,a_s}}\right) := \lambda_s,$$

and hence, also by Lemma 32, we have:
$$\bar{z}_s = \frac{\|\bar{\boldsymbol{p}}_s\|_1}{\|\bar{\boldsymbol{p}}_s\|_1 - (1 - \bar{\lambda}_s)\bar{p}_{s,a_s}} = \frac{\|\boldsymbol{p}_s\|_1}{\|\boldsymbol{p}_s\|_1 - (1 - \lambda_s)p_{s,a_s}} := z_s,$$

so, by the PROTOGABA algorithm and the inductive hypothesis, we have:

$\kappa_{s+1}(a, \Downarrow(n))$
$:= (\llbracket u_s \notin \Downarrow(n) \rrbracket + \llbracket u_s \in \Downarrow(n) \wedge a \neq a_s \rrbracket z_s + \llbracket u_s \in \Downarrow(n) \wedge a = a_s \rrbracket \lambda_s z_s)\kappa_s(a, \Downarrow(n))$
$= (\llbracket n \notin \Uparrow(u_s) \rrbracket + \llbracket n \in \Uparrow(u_s) \wedge a \neq a_s \rrbracket z_s + \llbracket n \in \Uparrow(u_s) \wedge a = a_s \rrbracket \lambda_s z_s)\kappa_s(a, \Downarrow(n))$
$= (\llbracket n \notin \Uparrow(u_s) \rrbracket + \llbracket n \in \Uparrow(u_s) \wedge a \neq a_s \rrbracket z_s + \llbracket n \in \Uparrow(u_s) \wedge a = a_s \rrbracket \lambda_s z_s)\bar{\kappa}_s(a, n)$
$= (\llbracket n \notin \Uparrow(u_s) \rrbracket + \llbracket n \in \Uparrow(u_s) \wedge a \neq a_s \rrbracket \bar{z}_s + \llbracket n \in \Uparrow(u_s) \wedge a = a_s \rrbracket \bar{\lambda}_s \bar{z}_s)\bar{\kappa}_s(a, n)$
$:= \bar{\kappa}_{s+1}(a, n)$

as required. This completes the proof. $\square$

From the PROTOGABA algorithm and Lemma 72 we have, for all $a \in [K]$ and $t \in [T]$, that:

$$\begin{aligned}
p_{t,a} &:= \sum_{U \in \mathcal{E}: u_t \in U} \kappa_t(a, U) \\
&= \sum_{n \in \mathcal{C}: u_t \in \Downarrow(n)} \kappa_t(a, \Downarrow(n)) \\
&= \sum_{n \in \Uparrow(u_t)} \kappa_t(a, \Downarrow(n)) \\
&= \sum_{n \in \Uparrow(u_t)} \bar{\kappa}_t(a, n) \\
&:= \bar{p}_{t,a},
\end{aligned}$$

so $\bar{\boldsymbol{p}}_t = \boldsymbol{p}_t$. By Lemma 31 we then have that :
$$\mathbb{P}[a_t = a] = \frac{\bar{p}_{t,a}}{\|\bar{\boldsymbol{p}}_t\|_1} = \frac{p_{t,a}}{\|\boldsymbol{p}_t\|_1},$$

so the selections of GABA-II equal those of PROTOGABA. This completes the proof. ∎

## J.6 Proof of Lemma 35

**Result**. We have that:

- For all $u \in [N]$ there exists a unique pair $(a, U) \in \mathcal{A}$ with $u \in U$.
- For all $u \in [N]$ and $(a, U) \in \mathcal{A}$ with $u \in U$, we have that $a = y(u)$.

**Proof:**

Suppose we have some $u \in [T]$. Let $n$ be the ancestor of $u$ of least depth in $\mathcal{C}$ which satisfies $y(v) = y(u)$ for all $v \in \Downarrow(n)$. Note that such a $n$ exists as $u$ is an ancestor of $u$ with $y(v) = y(u)$ for all $v \in \Downarrow(u)$ so the set we're selecting from is non-empty. Suppose we now take some $(a, n') \in [K] \times \mathcal{C}$ with $n' \in \Uparrow(u)$. We have the following cases:

- Suppose $a \neq y(u)$. Then $u \in \Downarrow(n')$ and $y(u) \neq a$ so $(a, n') \notin \mathcal{A}^{\dagger}$.
- Suppose $a = y(u)$ and $n' \neq n$. We now have two subcases:
  - Suppose $n'$ is a descendant of $n$. Then $\uparrow(n')$ is also a descendant of $n$ so since then $\Downarrow(\uparrow(n'))$ is a subset of $\Downarrow(n)$ we have, by definition of $n$, that for all $v \in \Downarrow(\uparrow(n'))$ we have $y(v) = y(u) = a$. By definition of $\mathcal{A}^{\dagger}$ we must then have $(a, n') \notin \mathcal{A}^{\dagger}$.
  - Suppose $n'$ is an ancestor of $n$. Then $n'$ is of lower depth that $n$ so, by definition of $n$, we must have that there exists some $v \in \Downarrow(n')$ with $y(v) \neq y(u) = a$ and so, by definition of $\mathcal{A}^{\dagger}$, that $(a, n') \notin \mathcal{A}^{\dagger}$.

  So in either subcase we have $(a, n') \notin \mathcal{A}^{\dagger}$.
- Suppose $a = y(u)$ and $n' = n$. Then we have two subcases:
  - Suppose $n$ is the root of $\mathcal{C}$. By definition of $n$, we have, for all $v \in \Downarrow(n) = \Downarrow(n')$, that $y(v) = y(u) = a$ and $n'$ is the root of $\mathcal{C}$. By definition of $\mathcal{A}^{\dagger}$ we must then have $(a, n') \in \mathcal{A}^{\dagger}$.
  - Suppose $n$ is not the root of $\mathcal{C}$. Then $\uparrow(n')$ is the parent of $n$ so is at lower depth than $n$ and hence, by definition of $n$, there exists some $v \in \Downarrow(\uparrow(n'))$ with $y(v) \neq y(u) = a$. By definition of $n$, we also have that all $v \in \Downarrow(n')$ satisfy $y(v) = y(u) = a$. By definition of $\mathcal{A}^{\dagger}$ we must then have $(a, n') \in \mathcal{A}^{\dagger}$.

  So in either subcase we must have $(a, n') \in \mathcal{A}^{\dagger}$.

We have hence shown that $(a, n') \in \mathcal{A}^{\dagger}$ if and only if $a = y(u)$ and $n' = n$ so, by definition of $\mathcal{A}$ we have that $(a, \Downarrow(n')) \in \mathcal{A}$ if and only if $a = y(u)$ and $n' = n$ which implies the result. ∎

## J.7 Proof of Lemma 36

**Result**. We have:
$$\mathbb{E}\left[|\mathcal{A}|\right] \leq 4\Psi \log_2\left(\frac{eN}{\Psi}\right).$$

**Proof:**

**Definition 73.** *Let:*
$$\Gamma := \sum_{u \in [N-1]} [\![y(u) \neq y(u+1)]\!].$$

**Lemma 74.** *For all $n \in \mathcal{C}$ there exists at most one $a \in [K]$ such that $(a, n) \in \mathcal{A}^{\dagger}$.*

*Proof.* Suppose $(a, n) \in \mathcal{A}^{\dagger}$ for some $a \in [K]$ and take any $b \in [K] \setminus \{a\}$. Then by definition of $\mathcal{A}^{\dagger}$ we have $y(u) = a \neq b$ for all (and hence since $\Downarrow(n) \neq \emptyset$, for some) $u \in \Downarrow(n)$ so by definition of $\mathcal{A}^{\dagger}$ we have $(b, n) \notin \mathcal{A}^{\dagger}$. This implies the result. □

**Lemma 75.** *For all $(a, n) \in \mathcal{A}^{\dagger}$ such that $n$ is not the root of $\mathcal{C}$ there exists some $u \in \Downarrow(\uparrow(n))$ with $y(u) \neq y(u+1)$.*

*Proof.* Suppose, for contradiction, the contrary: that $y(u) = y(u+1)$ for all $u \in \Downarrow(\uparrow(n))$. Then since $\Downarrow(\uparrow(n))$ is a complete interval of natural numbers we have, by a simple induction, that there exists $b \in [K]$ such that for all $u \in \Downarrow(\uparrow(n))$ we have $y(u) = b$. By definition of $\mathcal{A}^\dagger$ we must have that all $u \in \Downarrow(n)$ satisfy $y(u) = a$ and hence, as $\emptyset \neq \Downarrow(n) \subseteq \Downarrow(\uparrow(n))$, we have that there exists $u \in \Downarrow(\uparrow(n))$ with $y(u) = a$. So $b = a$ and hence all $u \in \Downarrow(\uparrow(n))$ satisfy $y(u) = a$. But this contradicts the fact that $(a,n) \in \mathcal{A}^\dagger$ which completes the proof. $\qquad\square$

**Lemma 76.** *Given $d \in [h]$, the cardinality of the set of all $n \in \mathcal{C}$ at depth $d$, such that there exists $a \in [K]$ with $(a,n) \in \mathcal{A}^\dagger$, is bounded above by $2\Gamma$.*

*Proof.* Given some $n' \in \mathcal{C}$ at depth $d-1$ with a child $n \in \mathcal{C}$ and some $a \in [K]$ with $(a,n) \in \mathcal{A}^\dagger$ we have, from Lemma 75, that there exists some $u \in \Downarrow(n')$ with $y(u) \neq y(u+1)$. Now, since all such $\Downarrow(n')$ are pairwise disjoint, we must have that the cardinality of the set of all such $n'$ is bounded above by $\Gamma$. Since each such $n$ has only two children the result follows. $\qquad\square$

Suppose we have $\Gamma \geq 1$. Since there are no more than $4\Gamma$ vertices at depth at most $\log_2(\Gamma) + 1$ we have, by Lemma 74, that the number of pairs $(a,n) \in \mathcal{A}^\dagger$ in which $n$ is at depth at most $\log_2(\Gamma) + 1$ is no more that $4\Gamma$. Also, by lemmas 74 and 76, we have that for all $d \in [h]$ there are at most $2\Gamma$ pairs $(a,n) \in \mathcal{A}^\dagger$ in which $n$ is at depth $d$, which implies that there are at most

$$2\Gamma(h - \log_2(\Gamma)) = 2\Gamma(\log_2(N) - \log_2(\Gamma)) = 2\Gamma \log_2\left(\frac{N}{\Gamma}\right)$$

pairs $(a,n) \in \mathcal{A}^\dagger$ in which $n$ is at depth greater than $\log_2(\Gamma) + 1$. So the total cardinality of $\mathcal{A}^\dagger$ is bounded above by:

$$|\mathcal{A}^\dagger| \leq 2\Gamma \log_2\left(\frac{N}{\Gamma}\right) + 4\Gamma = 2\Gamma\left(\log_2\left(\frac{N}{\Gamma}\right) + \log_2(4)\right) = 2\Gamma \log_2\left(\frac{4N}{\Gamma}\right) < 2\Gamma \log_2\left(\frac{2eN}{\Gamma}\right).$$

Since the function $x \to 2x \log_2(2eN/x)$ is concave and monotonic increasing for $x \leq 2N$ and $2\Psi < 2N$ we then have, by definition of $\mathcal{A}$, Jenson's inequality, and Lemma 2, that:

$$\mathbb{E}[|\mathcal{A}|] = \mathbb{E}[|\mathcal{A}^\dagger|] \leq \mathbb{E}\left[2\Gamma \log_2\left(\frac{2eN}{\Gamma}\right)\right] \leq 2\mathbb{E}[\Gamma] \log_2\left(\frac{2eN}{\mathbb{E}[\Gamma]}\right) \leq 4\Psi \log_2\left(\frac{eN}{\Psi}\right)$$

as required. $\qquad\blacksquare$

## J.8 Proof of Theorem 37

**Result**. Given $\Psi \leq N/2$ and setting:

$$\eta := \sqrt{\frac{8\Psi \log_2(eN/\Psi) \ln(3KN/2\Psi)}{KT}},$$

we have, for GABA-II:

$$R \in \mathcal{O}\left(\sqrt{\ln\left(\frac{N}{\Psi}\right) \ln\left(\frac{KN}{\Psi}\right) \Psi KT}\right).$$

**Proof:**

Lemmas 33 and 35 allow us to invoke Theorem 9 giving us, by definition of $\sigma$:

$$R \leq \frac{1}{\eta} \sum_{(a,U) \in \mathcal{A}} \ln\left(\frac{K}{|\mathcal{A}|\sigma(U)}\right) + \frac{\eta KT}{2}$$

$$= \frac{1}{\eta} \sum_{(a,U) \in \mathcal{A}} \ln\left(\frac{K(2N-1)}{|\mathcal{A}|}\right) + \frac{\eta KT}{2}$$

$$\leq \frac{1}{\eta} \sum_{(a,U) \in \mathcal{A}} \ln\left(\frac{2KN}{|\mathcal{A}|}\right) + \frac{\eta KT}{2}$$

$$= \frac{1}{\eta}|\mathcal{A}| \ln\left(\frac{2KN}{|\mathcal{A}|}\right) + \frac{\eta KT}{2}$$

$$< \frac{1}{\eta}|\mathcal{A}| \ln\left(\frac{6KN}{|\mathcal{A}|}\right) + \frac{\eta KT}{2}. \tag{18}$$

By the change of base rule for logarithms we have:

$$4\Psi \log_2\left(\frac{eN}{\Psi}\right) = \frac{4\Psi}{\ln(2)} \ln\left(\frac{eN}{\Psi}\right) = \frac{4eN}{\ln(2)}\left(\frac{\Psi}{eN}\right) \ln\left(\frac{eN}{\Psi}\right),$$

so since the function $x \ln(1/x)$ has a maximum value of $1/e$ we have that $4\Psi \log_2\left(\frac{eN}{\Psi}\right)$ is no greater than $4eN/(\ln(2)e) < 12N/e \leq 6KN/e$. Note also that by definition $|\mathcal{A}| \leq N < 6KN/e$.

Hence, since the function $x \to x\ln(6KN/x)$ is concave and monotonic increasing for $x \leq 6KN/e$ we have, from Jenson's inequaltiy and Lemma 36, that:

$$\mathbb{E}\left[|\mathcal{A}| \ln\left(\frac{6KN}{|\mathcal{A}|}\right)\right] \leq \mathbb{E}\left[|\mathcal{A}|\right] \ln\left(\frac{6KN}{\mathbb{E}\left[|\mathcal{A}|\right]}\right) \leq 4\Psi \log_2\left(\frac{eN}{\Psi}\right) \ln\left(\frac{3KN}{2\Psi \log_2(eN/\Psi)}\right).$$

Substituting into Equation 18 and noting that since $\Psi \leq N$ we have $\log_2(eN/\Psi) \geq \log_2(e) > 1$ we then have, by taking expectations:

$$R < \frac{1}{\eta}4\Psi \log_2\left(\frac{eN}{\Psi}\right) \ln\left(\frac{3KN}{2\Psi \log_2(eN/\Psi)}\right) + \frac{\eta KT}{2} \leq \frac{1}{\eta}4\Psi \log_2\left(\frac{eN}{\Psi}\right) \ln\left(\frac{3KN}{2\Psi}\right) + \frac{\eta KT}{2}$$

so setting:

$$\eta := \sqrt{\frac{8\Psi \log_2(eN/\Psi) \ln(3KN/2\Psi)}{KT}}$$

gives us:

$$R < \sqrt{8\Psi \log_2\left(\frac{eN}{\Psi}\right) \ln\left(\frac{3KN}{2\Psi}\right) KT}.$$

Enforcing $\Psi \leq N/2$ ensures that $N/\Psi$ and $KN/\Psi$ never fall below a positive constant so:

$$\log_2\left(\frac{eN}{\Psi}\right) = \log_2(e) + \frac{1}{\ln(2)} \ln\left(\frac{N}{\Psi}\right) \in \mathcal{O}\ln\left(\frac{N}{\Psi}\right),$$

and

$$\ln\left(\frac{3KN}{2\Psi}\right) = \ln\left(\frac{3}{2}\right) + \ln\left(\frac{KN}{\Psi}\right) \in \mathcal{O}\left(\ln\left(\frac{KN}{\Psi}\right)\right),$$

which implies the result. $\blacksquare$