# OpenReview forum: "A Gang of Adversarial Bandits"
_NeurIPS.cc/2021/Conference — NeurIPS 2021 Poster_

### Official Review · Reviewer_A1FJ · 2021-07-13

**Rating:** 6
**Confidence:** 3

**Summary:**

This paper studies a new multi-armed bandit setting termed "Gang of Adversarial Bandits", in which a learner solves an instance of the adversarial MAB problem for each of $N$ different users. Without further assumptions on the $N$ users, essentially one can only solve the instances independently (e.g., running $N$ copies of the EXP3 algorithm), and incur a loss that scales as $\sqrt{KNT}$, where $K$ is the number of actions and $T$ is the time horizon.

In contrast, the authors assumes that the connections between the users are specified by a known weighted graph $G$, and assumes that nearby vertices in the graph (measured in terms of the effective resistance) are more likely to have the same action in the "benchmark" mapping $y: [N] \to [K]$. Both algorithms proposed in the paper are instantiations of the "specialists" framework, using graph $G$ to define certain initial/prior distributions over a carefully-chosen family of "specialists".

This approach involves sampling from the (exponentially large) space of mappings from users to actions, so a naive implementation would be computationally costly. The authors show that by sampling a random spanning tree and then embedding the users to a binary tree, the running time per step can be significantly reduced to either $O(K\log N)$ or $O(\log K\log N)$.

**Main Review:**

The paper is generally well-written, except that some of the technical parts (e.g., Figures 3 and 4 can be hard to parse). The authors provided adequate motivation for the problem setting, as well as the specific approach that they adopt (e.g., why resistance weighted cutsize can be favorable, and the distinction between $\Phi$ and $\Psi$).

I have a question about Corollaries 3 and 4: the choice of the learning rate $\eta$ in both algorithms depends on $\Psi(y)$, where $y$ is the "benchmark" to be compared to, and should be unknown to the learner. So why can we set $\eta$ in this way? (My guess is that we can apply a doubling trick on this quantity, e.g., by considering $\log N$ copies of the algorithm with $\Psi(y)$ being $1, 2, 4, 8, \ldots, N$ and then running another EXP3 on top of these. Even if this were true, some explanation seems to be needed.)

Overall, I feel that the authors proposed a new setting that is interesting and well-motivated. The proposed solutions follow a relatively standard framework, but still require non-trivial ideas as well as tools from other fields to obtain computationally efficient algorithms. I would recommend accept.

Minor comments:
- Equation (2): Some justification/explanation would be helpful for this claim.
- Displayed equation below Line 138: The denominator seems a bit off after the colon---is it $x_u - x_v = 1$, and should it be a constraint of the $\min$?

**Time Spent Reviewing:**

2

---

> ### Author Response · Authors · 2021-08-11
> **Response to Reviewer A1FJ**
>
> Thank you for your review and comments. Please find our specific responses to your comments below.
>
> > "I have a question about Corollaries 3 and 4: the choice of the learning rate $\eta$ in both algorithms depends on $\Psi(y)$ , where
>  is the "benchmark" to be compared to, and should be unknown to the learner. So why can we set $\eta$
>  in this way? (My guess is that we can apply a doubling trick on this quantity, e.g., by considering $\log N$
> copies of the algorithm with $\Psi(y)$ being $1,2,4,8,\ldots,N$  and then running another EXP3 on top of these. Even if this were true, some explanation seems to be needed.)"
>
> This is a good point. It was also raised by reviewer GuKU, and we paste our response to GuKU for your convenience here.
>
> GABA-I depends on two parameters, $\phi$ and $\eta$, while GABA-II depends only on $\eta$.  In the following, we will 1) sketch how to autotune $\phi$ at little cost and 2) autotune $\eta$, however at essentially the cost of moving $\Psi(y)$ outside of the square-root.
>
>
> We first describe how to automatically tune the parameter $\phi$ that appears in GABA-I. To do this, we make $\log(N)$ copies of each specialist whose weights are derived (as in the current paper) from exponentially increasing values of $\phi$. Each specialist weight is then divided by $\log(N)$ so that all weights add up to one, noting that the additional $\log(N)$ factor to the weights has only an extremely small effect on the regret bound. Implementing this efficiently is similar to the implementation in the paper, except, we now have $\log(N)$ copies of the BST, each with a different value of $\phi$. The computed values from the different copies of the BST are summed to find the prediction vector and the normaliser on each trial.  This comes at the cost of an additional $\log N$ factor in the computation time.
>
>
> Now that we have shown how to automatically tune $\phi$, we are left with the learning rate $\eta$ in both algorithms. We first note that, with any $\eta$, the regret of both algorithms is $\Upsilon/\eta + \eta KT/2$, where $\Upsilon$ is the robustified resistance weighted cutsize $\Psi(y)$ multiplied by logarithmic terms (one in GABA-I and two in GABA-II). By setting $\eta=\sqrt{2/KT}$ we get a regret of $(\Upsilon+1)\sqrt{KT/2}$. In addition, if $T$ is unknown then a doubling trick can be performed with this result to get a regret bound of order $\Upsilon \sqrt{KT}$ with no parameters needed. We compare this to the regret bound of order $\sqrt{\Upsilon KT}$ that comes from the optimal tuning of $\eta$. In general note that, for any $c>0$, we have, by setting $\eta=c \sqrt{2/KT}$, a regret of $(\frac{\Upsilon}{c}+c)\sqrt{KT/2}$ where the parameter $T$ can then be learnt by a doubling trick.
>
> Thus at this present time we can autotune to obtain $O(\Upsilon\sqrt{KT})$ regret.  We leave it as an open problem to bring $\Upsilon$ inside the square-root.
>
> ---
>
> > "Equation (2): Some justification/explanation would be helpful for this claim."
>
> The result and its proof are contained in, e.g., MAB survey  [BW12, Theorem 4.1].  We will add a clarifying reference to the paper.
>
> [BW12] Regret Analysis of Stochastic and Nonstochastic Multi-armed Bandit Problems, Bubeck & Cesa-Bianchi 1992
>
> ---
>
> > "Displayed equation below Line 138: The denominator seems a bit off after the colon---is it
> and should it be a constraint of the min?"
>
> Thank you.  Yes there is a typo.  The eq. after the colon is the constraint which should be over u,v rather than i,j.  Thus the proper definition is $r(u,v) := \frac{1}{\min_{x\in \Re^N} \\{\sum_{i<j}^N \omega_{i,j} (x_i -x_j)^ 2 :x_u -x_v = 1\\}}$

---

### Official Review · Reviewer_GuKU · 2021-07-14

**Rating:** 6
**Confidence:** 3

**Summary:**

This paper considers the problem of contextual bandit problem with adversarial loss and finite number of contexts. Specifically, the authors consider the case where the contexts are related in some sense such that the action generated by a policy seeing one context may be related to the one seeing another similar context. The similarity is measured by a weighted undirected graph G. The authors then design two efficient algorithms which achieves $\widetilde{O}(\sqrt{\Psi(y) K T})$ regret bound in this setting where $y$ is the benchmark policy and $\Psi(y)$ is a dispersion measure of $y$ over the context. This result is better than $\widetilde{O}(\sqrt{NKT})$, where $N$ is the number of contexts.

The algorithm is mainly based on classic EXP3/EXP4 algorithm. As generally EXP4 is inefficient, the authors provide two efficient implementation by using online belief propagation over a complete binary tree for contexts, which is generated from the graph G. The first algorithm GABA-I has $O(K\ln(N))$ computation complexity per round while the second algorithm GABA-II has $O(\ln(K)\ln(N))$ computation complexity but with a factor of $\log(N)$ worse regret bound.

**Ethical Concerns:**

None.

**Ethics Review Area:**

["I don’t know"]

**Limitations And Societal Impact:**

No negative societal impact is shown in the work. Suggestions are written in Cons. in the Main Review.

**Main Review:**

Pros:
1. The problem proposed in this paper is well motivated and closer to the application, which considers the similarity among the contexts instead of considering them independently.


2. The authors propose algorithms with theoretical guarantees on the regret bound and analyze the computational cost of the algorithm, which is efficient and implementable in practice. I didn't check the proof in very details, especially the second algorithm GABA-II but overall it looks sound to me.


3. The related works are adequately cited.

Cons:
1. One of my main concern of this work is about the tuning issue. As the authors mentioned in the paper, the obtained bound is an non-uniform bound which gives different bounds for different comparator policy $y$. I was assuming this is a parameter-free algorithm but it seems that $\phi$ and $\eta$ in the algorithm depend on $\Psi(y)$, which may not be known during learning as we do not know which $y$ we are comparing to. Is my understanding correct? If so, then the result may be less attractive and interesting as if we have one policy $y_0$ whose $\Psi(y_0)$ is large, then the regret will scale with this policy, even though $y_0$ is a suboptimal policy.


2. Also I wonder whether there are some better intuitions on the social network graph G defined in the paper, especially on the weight of the edge? It seems that this weight is only used in describing the property of a uniformly generated spanning tree and giving an upper bound on the quantity $\mathbb{E}\left[\sum_{i=1}^{N-1} \mathbb{I}[y(u)\ne y(u+1)]\right]$ in Lemma 1, which looks more like a technical design. Although in Section 2.1, the authors say that the larger the weight is, the more paths there are between two contexts, the less penalty (on $\Phi(y)$ or $\Psi(y)$) will be paid if $y$ has different output actions on these two contexts, I still could not have a very intuitive idea on this weight. Or is there any other metrics to measure the similarity among contexts?


3. About the size of $\Phi(y)$. Although I understand $\Phi(y)\leq N$ and for some specific graph and policy pairs (as shown from line 156 to line 166) this value is much less than $N$, I wonder whether in some more realistic applications, this value could be much better than the worst case? I think this will show more contributions to the proposed algorithms.


4. For the design of the two algorithms, although the runtime each round is $O(K\ln (N))$, which is efficient, the space complexity is still $O(KN)$, which is linear in the number of contexts and is not that efficient. I was supposing that by utilizing the social network graph G, we can reduce the size of the context in some sense and lead to a better space complexity. I think reducing this space complexity should also be important.


5. Some minor typos:
- line 56 $[1,n]$ -> $[1,N]$


======= After rebuttal =========
During the rebuttal, the authors addressed my issues and I updated my score.

**Time Spent Reviewing:**

7 hours

---

> ### Author Response · Authors · 2021-08-11
> **Response to Reviewer GuKU**
>
> Thank you for your review and comments. Please find our specific responses to your comments below.
>
> >"1.  One of my main concern of this work is about the tuning issue.  [...] is my understanding correct? "
>
> Your understanding is correct; however, we offer the following as a (partial) remedy.
>
> GABA-I depends on two parameters, $\phi$ and $\eta$, while GABA-II depends only on $\eta$.  In the following, we will 1) sketch how to autotune $\phi$ at little cost and 2) autotune $\eta$, however at essentially the cost of moving $\Psi(y)$ outside of the square-root.
>
>
> We first describe how to automatically tune the parameter $\phi$ that appears in GABA-I. To do this, we make $\log(N)$ copies of each specialist whose weights are derived (as in the current paper) from exponentially increasing values of $\phi$. Each specialist weight is then divided by $\log(N)$ so that all weights add up to one, noting that the additional $\log(N)$ factor to the weights has only an extremely small effect on the regret bound. Implementing this efficiently is similar to the implementation in the paper, except, we now have $\log(N)$ copies of the BST, each with a different value of $\phi$. The computed values from the different copies of the BST are summed to find the prediction vector and the normaliser on each trial.  This comes at the cost of an additional $\log N$ factor in the computation time.
>
>
> Now that we have shown how to automatically tune $\phi$, we are left with the learning rate $\eta$ in both algorithms. We first note that, with any $\eta$, the regret of both algorithms is $\Upsilon/\eta + \eta KT/2$, where $\Upsilon$ is the robustified resistance weighted cutsize $\Psi(y)$ multiplied by logarithmic terms (one in GABA-I and two in GABA-II). By setting $\eta=\sqrt{2/KT}$ we get a regret of $(\Upsilon+1)\sqrt{KT/2}$. In addition, if $T$ is unknown then a doubling trick can be performed with this result to get a regret bound of order $\Upsilon \sqrt{KT}$ with no parameters needed. We compare this to the regret bound of order $\sqrt{\Upsilon KT}$ that comes from the optimal tuning of $\eta$. In general note that, for any $c>0$, we have, by setting $\eta=c \sqrt{2/KT}$, a regret of $(\frac{\Upsilon}{c}+c)\sqrt{KT/2}$ where the parameter $T$ can then be learnt by a doubling trick.
>
> Thus at this present time we can autotune to obtain $O(\Upsilon\sqrt{KT})$ regret.  We leave it as an open problem to bring $\Upsilon$ inside the square-root.
>
> ---
>
> > "2.  Also I wonder whether there are some better intuitions on the social network graph $\mathcal{G}$ defined in the paper, especially on the weight of the edge? It seems that this weight is only used in describing the property of a uniformly generated spanning tree and giving an upper bound on the quantity
> $\mathbb{E}\left[\sum_{u=1}^{N-1} \mathbb{I}[y(u) \ne y(u+1)]\right]$ in Lemma 1, which looks more like a technical design. Although in Section 2.1, the authors say that the larger the weight is, the more paths there are between two contexts, the less penalty (on or $\Phi(y)$ will be paid if  $\Psi(y)$ has different output actions on these two contexts, I still could not have a very intuitive idea on this weight. Or is there any other metrics to measure the similarity among contexts?"
>
> Regarding the weights of the edges, we said informally (125-126) "our homophilic bias can be stated as follows: users strongly connected w.r.t. the link weights $\omega$ tend to be associated with the same label."  Thus a large $\omega$ weight *directly* expresses the  bias that those two users/contexts share the same label.
> Formally, we said (134-135): "More precisely, viewing the graph as an electrical circuit,
> where each edge weight $\omega_{u,v}$ corresponds to a $\frac{1}{\omega_{u,v}}$-resistor, [..]"  I.e., the edge weight is electrically equivalent to the inherent conductance of that edge.
>
> To gain some intuition about this formal analogy to an electric network, let us build an illustrative example.  Consider the following 6 user network with edges $(1,2),(1,3),(2,4),(3,4),(4,5),(5,6)$, and assume each edge is unit weight.  Although user 4 is not directly connected to user 1 or 6 (i.e., $\omega_{1,4}=\omega_{4,6}=0$), in some sense user 4 is more connected to user 1 than to user 6.  This is because although they are both two edges distant, there is only one path of length two between 4 and 6, while there are two edge disjoint paths of length two between 1 and 4.  This is reflected in their effective resistances $r(4,6) =2$ because there are two unit "resistors" in series and $r(1,4) =1$ by applying the law of resistors in series and then the law of resistors in parallel.  Also for insight observe that $r(u,v) \le \frac{1}{\omega_{u,v}}$.
>
> Thus the set of $\\{\omega_{u,v}\\}$ as given to the algorithm determines a resistive circuit.  This set then induces the effective resistances $r(u,v)$, which in fact defines a *metric* over $[N]$ i.e., $r(a,b) + r(b,c) \ge r(a,c)$.
>
> Finally, the key term in our bound is the effective resistance weighted cutsize $\Phi(y)$  (and its robustifcation $\Psi(y)$).These quantities depend on the sum of products of $\omega_{u,v} r(u,v)$. This implies that $\Phi(y)$ is *scale-free* with respect to the weights, i.e., if we multiply *all* of the $\omega$'s by a constant $c$ then $\Phi(y)$ ($\Psi(y)$)  is unchanged.
>
> ---
>
> >"3. About the size of $\Phi(y)$ . Although I understand $\Phi(y)\le N$
>  and for some specific graph and policy pairs (as shown from line 156 to line 166) this value is much less than $N$, I wonder whether in some more realistic applications, this value could be much better than the worst case? I think this will show more contributions to the proposed algorithms."
>
> Yes, this a very good point.  Reviewer Dc7h also made essentially the same point.  Below we will provide the same application that we provided them.
>
>
>
> We provide the workings of the example suggested by reviewer Dc7h: "$N$ contexts can be clustered into $G$ groups" as "corresponding to some social network graph composed of $G$ highly-weighted cliques,"
>
>  In  summary, we will show that $N$ may be replaced by $G$ up to logarithmic factors.
>
>
> [Single clique].  First consider an unweighted $N$-vertex clique.  Between any two vertices there are $N-2$ edge-disjoint paths of length 2 and 1 edge disjoint path of length 1 thus $r(u,v) \le 1/(1 + ((N - 2)/2) = 2/N$  (using the law of resistors series/parallel and Rayleigh's monotonicity principle), for any pair of vertices.
>
> [A labeled "clique of cliques"]. Next consider that we have $G$, $n$-cliques (thus $N= G n$), and assume that $G = o(n)$, with $n/G$ inter-clique edges between each of the cliques. Thus the total number of inter-clique edges is $(G-1)n/2$.  (Exactly how the inter-edges connect any pair of cliques does not matter).   Observe  if associated with each clique is some action in $[K]$, then in the worst case the cutsize (def. line 129) is the number of inter-clique edges $(G-1)n/2$.  Furthermore, the effective resistance of an inter-clique edge is also $O(1/n)$. This follows since leaving each clique is $\Theta(n)$ edges, and, like we did with the computation of intra-clique effective resistance given any two points on differing cliques, we can find $\Theta(n)$ edge disjoint of $\Theta(1)$ length.   Therefore the resistance-weighted cutsize $\Phi(y)$ (see (4)) is bounded by $(G-1)n/2 \times O(1/n) = O(G)$.  Substituting into (8) we have  $O(\sqrt{K \log(K n) G T})$.
>
> Thus as you ask "does the dependence on $N$ decrease to a dependence on $G$?" we can answer in the affirmative as $G$ has replaced $N$ except for a logarithmic factor.  Finally, note this result is robust in that we could apply "edge noise" to create more "realistic" graphs, as well as increase the number of inter-clique edges by a constant factor, and still the bound will scale smoothly.
>
> ---
>
> >"4. For the design of the two algorithms, although the runtime each round is $O(K \ln N)$,
> which is efficient, the space complexity is still $O(K N)$ , which is linear in the number of contexts and is not that efficient. I was supposing that by utilizing the social network graph $G$, we can reduce the size of the context in some sense and lead to a better space complexity. I think reducing this space complexity should also be important."
>
> This is an interesting point for further research.  It does not seem straightforward to us that we can maintain the $O(\log n)$ time complexity for either GABA-I or GABA-II without order $O(KN)$ space complexity, as both algorithms essentially obtain their speed-ups by re-using cached values in the binary support tree.  If we were to create a GABA-III to minimise space complexity we believe we could do so with $O(\\min(T, KN))$ space complexity by just storing the data sequence and then using an adapted "classical" belief propagation.  This would lead to $O(K N)$ prediction time, thus losing the faster prediction times of $O(K \\log N)$ [GABA-I] or $O(\\log K \\log N)$ [GABA-II].
>
> ---
>
> >"5. Some minor typos:"
>
> Thank you for these corrections.

---

> > ### Comment · Reviewer_GuKU · 2021-08-23
> > **On the dependence of \Psi**
> >
> > Thanks for the authors response!
> >
> > My concerns are mostly addressed but it seems that now the regret bound is suboptimal in the dependence of the number of context $N$? If we consider the worst case bound, then the bound could be $N\sqrt{KT}$? The bound with $\Psi(y)$ outside seems to make the bound worse. I wonder whether there are some robustness such as $\min({\Phi(y)\sqrt{KT}, \sqrt{NKT}})$?

---

> > > ### Author Response · Authors · 2021-08-27
> > > **Re: dependence of \Psi**
> > >
> > > Thank you for your reply!
> > >
> > > For the sake of simplicity in the following discussion we ignore logarithmic factors and $K$ factors.
> > >
> > > >My concerns are mostly addressed but it seems that now the regret bound is suboptimal in the dependence of the number of contexts $N$?  If we consider the worst case bound, then the bound could be $N\sqrt{KT}$?
> > >
> > > Your understanding is correct.  Without a tuning oracle, we can obtain via a "defensive tuning" $\widetilde{O}(\Psi(y)\sqrt{KT})$.  With a tuning oracle, we have $\widetilde{O}(\sqrt{\Psi(y) KT})$ and in the worst case $\Psi(y) = N$.  However, in the following we discuss a very natural scenario where the baseline bound of $\widetilde{O}(\sqrt{NKT})$ is always dissatisfying but the defensively tuned bound $\widetilde{O}(\Psi(y)\sqrt{KT})$ is valuable.
> > >
> > >
> > > Consider a very large social network where the bandit problem is to show 1-of-K advertisements at nodes.  Now consider the case that each node  serves **at most one** advert (as will be the case in many applications). Since now $N\geq T$  the bound Eq (1) of the baseline competitor that runs EXP3 at each node is now an *always vacuous* $\Theta(T)$.  We can intuitively see this analysis is correct since the baseline algorithm is now just picking a single "uniformly at random" advertisement for each user independently.  Alternatively, consider GABA-I/II with our defensive "knowledge-free" tuning where we obtain $\widetilde{O}(\Psi(y)\sqrt{KT})$ regret.  Hence, if $\Psi(y) \in o(\sqrt{T})$ we then obtain non-vacuous regret. So in this case the "knowledge-free" bound of GABA-I/II is at least as good and often much better than that of the baseline.  Intuitively, GABA-I/II achieves this result since algorithmically it is exploiting the network structure.
> > >
> > > >The bound with $\Psi(y)$ outside seems to make the bound worse. I wonder whether there are some robustness such as $\min({\Phi(y)\sqrt{KT}, \sqrt{NKT}})$}?
> > >
> > > This is a good question!  If we were in the *full information setting* ("Hedge"), we could auto-tune to obtain $\widetilde{O}(\sqrt{\Phi(y) KT})$.
> > > As far as we are aware, none of these techniques transfer to the *bandit* setting to enable either the previous bound or the proposed "min" bound.
> > >
> > >  One way of framing this problem is as a nested model selection problem for the bandit setting.  Define $\Pi_c =${$ y : \Psi(y) \le c$}  observing that $\Pi_1 \subseteq \Pi_2 \subseteq \cdots \subseteq \Pi_N$.  The aim here is to obtain a bound in terms of some $f(\Pi_i)$ instead of  $f(\Pi_N)$, where $i$ indexes the smallest set that contains the "best" $y$.  Various special cases of this problem are discussed in [1,2] and why known approaches fail, in particular see [2, Appendix A].  Finally, we note the special case raised in [2] was posed as open problem with prize in COLT 2020 see [3].
> > >
> > > Thus in summary, although we do not know how to obtain the best bound without a tuning oracle, there exists natural scenarios where GABA-I/II with a "knowledge-free" tuning improves on the baseline competitor.  This is since the baseline is always vacuous but in a common case GABA-I/II is non-vacuous.
> > >
> > >
> > > [1] *Corralling a Band of Bandit Algorithms*, A. Agarwal, H. Luo, B. Neyshabur, R. Schapire
> > >
> > > [2] *Model Selection for Contextual Bandits*, D. Foster, A. Krishnamurthy, H. Luo.
> > >
> > > [3] *Open Problem: Model Selection for Contextual Bandits*, D. Foster, A. Krishnamurthy, H. Luo

---

> > > > ### Comment · Reviewer_GuKU · 2021-08-27
> > > > **Re: Re: dependence of \Psi**
> > > >
> > > > Thanks for the authors response and I understand that there are some difficulties in obtaining better bounds. I think though there are some improvements can be made, there are some interesting ideas in the work and I updated my score after the rebuttal.

---

### Official Review · Reviewer_7145 · 2021-07-16

**Rating:** 7
**Confidence:** 4

**Summary:**

In this paper the authors consider the problem of adversarial contextual bandits, where the contexts (finite) and the reward function are adversarial, but there exists a graph structure among the various context vectors. The goal of the online learning algorithm is to select a mapping from the context to an action that minimizes the total regret. Here, the aim is to achieve results better than naively treating the cross product of contexts and actions as an arm and running a EXP3 type algorithm. The authors present regret bounds that depend on an interesting quantity of the graph: weighted effective resistance. This quantity improves the regret to logartihmic when the max-cut size in the graph is small and recovers the worst-case bound (treating the number of context times the number of actions as the arms).

**Limitations And Societal Impact:**

The primary limitation of this paper is that although mathematically interesting, they do not make good case for motivating examples. In particular, the paper would have become much more stronger if they had a concrete motivating application where the losses are (necessarily) adversarial while the graph structure is randomized. Along those lines, the paper makes claims about improving computational complexity (asymptotically) even for the belief propagation problem. It would have been interesting to see some simulations to see how this translates for reasonable sized but smaller instances.

Societal impact
This is a theoretical paper and the impact on society is negligible. It is as good/bad as the entire area of theoretical online learning has had on society and thus, this paper itself does not pose any issues.

**Main Review:**

Originality: The setting considered in this paper is a new for the (oblivious) adversarial setting. Moreover, the notion of complexity used in this paper is new for the bandit literature (although well known in the graph theory and theoretical computer science communities) and it is interesting connection this paper brings about. The authors need to device new algorithms to exploit this graph. Underlying the two algorithms is a key data structure that embeds the graph G into a fully height binary tree such that the required complexity measure only increases by a constant factor in this new tree. Then they exploit prediction with expert advice type algorithm to update weights. In particular, they combine the standard belief propagation algorithm determined on the graphical model induced by this BST. This induces a distribution over the actions at time $t$ given the context $u_t$. They then sample using this distrubtion and perform a EXP3 style update of the parameters of the belief propagation algorithm. As far as I can tell both the algorithms presented in this paper (they differ in a small nuance) are original.


Quality/Clarity: The paper is excellently written with the main results, the algorithmic ideas and the construction of the required data-structures are well-communicated (even if many of them are from prior work, they introduce the necessary components in this paper). I did not have trouble understanding the main ideas of the paper. Although some of the notations were clumsy, I think that it is unavaiodable given the nature of the graph algorithm.

Significance: The results in this paper add to the literature of learning with expert advice and also learning on social networks. Each of these areas themselves have had and have wide range of applications. Oblivious adversary goes beyond the stochastic versions that is usually made in the learning on social networks which typically makes it non-applicable in real-world settings. This paper shows that even in the adversarial setting, assuming randomness in the social graph structure as opposed to losses, one can retrieve similar bounds, which is interesting.


**Time Spent Reviewing:**

4

---

> ### Author Response · Authors · 2021-08-11
> **Response to Reviewer 7145**
>
> Thank you for your review and comments. Please find our specific responses to your comments below.
>
> > "Originality: [...] In particular, they combine the standard belief propagation algorithm determined on the graphical model induced by this BST. This induces a distribution over the actions at time $t$ given the context $u_t$ . They then sample using this distribution and perform a EXP3 style update of the parameters of the belief propagation algorithm."
>
> I would like to emphasize that, as a part of our novel contribution, we do not in fact use *standard* belief propagation, nor do we have a *standard graphical model* defined on the BST.  In lines 199-202, we say this explicitly.  If as an alternate approach we used a standard graphical model, we could then apply batch belief propagation per trial requiring $O(K N)$, or we could use the online adaptation of belief propagation (with caching), using either [63,76] to obtain  $O(K^3 \log N)$ per trial.  The difference comes  from  the fact that our "specialist models" (see Sec. 3)  do not correspond in any straightforward way to a graphical model.  Finally,  we note that for GABA-I we do use online belief propagation [63,76] but as a subroutine (see 254-256).
>
> ---
>
> >"The primary limitation of this paper is that although mathematically interesting, they do not make good case for motivating examples. In particular, the paper would have become much more stronger if they had a concrete motivating application where (1) the losses are (necessarily) adversarial while the (2) graph structure is randomized."
>
> (1) Adversarial losses may arise naturally in a setting in which users may change their preference to products over time in an uncontrolled way, so that it would not be realistic to assume a stochastic model for the users.
>
> (2) In the model the user graph $\mathcal{G}$ is not randomized, and is in fact deterministic.  For the sake of computational efficiency only, we build a BST $\mathcal{C}$ (see Figure 1) which is now a random object.  Our regret bounds, however, depend on the topological properties of the original graph $\mathcal{G}$ and the context mapping $y$ via the robustified resistance weighted cutsize $\Psi(y)$. They do not depend on the properties of the particular sampled $\mathcal{C}$.
>
> ---
>
> >"Along those lines, the paper makes claims about improving computational complexity (asymptotically) even for the belief propagation problem. It would have been interesting to see some simulations to see how this translates for reasonable sized but smaller instances."
>
> We agree that simulations would be interesting here to see the speed-up crossover point.  We note in the case of this algorithm that (unlike, say, matrix multiplication improvements), the speed-up of $O(n)$ to $O(\log n)$ is an exponential improvement. And we note that, in our experience with similar tree-based algorithms,  such a crossover point occurs at "relatively small" integers ($n \le 50$).

---

### Official Review · Reviewer_Dc7h · 2021-07-25

**Rating:** 7
**Confidence:** 4

**Summary:**

This paper studies the following contextual variant of the adversarial multi-armed bandit problem. Each round t (for T rounds), the adversary begins by choosing a context (“user”) u_t in [N] and a loss function l_t from [K] -> R, and reveals the context to the learning algorithm. Based on this context, the learner must select an action a_t in [K]. The adversary then reveals the value of l_t(a_t) to the learner, which the learner suffers as loss. By running a separate instance of EXP3 for each context, it is possible for the learner to achieve a total loss of O(sqrt(NKT)). This paper asks the question of whether we can do better if we believe that “similar” users should be subject to “similar” losses.

The authors answer this in the following way. They assume there is some given graph structure on the N users (“social network graph”), where users i and j are connected with a link of weight w_{i, j} (and higher weights mean users are more likely to have similar losses). They then give the following non-uniform regret bound: against the fixed policy y : [N] -> [K], their algorithms get regret of the form ~ O(sqrt(KT f(y))), where f(y) is a function that measures how “consistent” y is with the social network graph. For example, if y(s) is a constant function (doesn’t depend on user), then this is very consistent with any graph structure, and f(y) ~= log(N). On the other hand, for arbitrary y, y(s) can be as large as ~O(N), recovering the original ~O(sqrt(KTN)) bound. Formally, f(y) is something like the average effective resistance between users with different values of y(s).

The author’s algorithms are based on a technique they call “predicting with specialists” (extending the work of Freund et al.), where one maintains a distribution over functions mapping contexts to either actions or the decision to abstain. In particular, it’s possible to get a non-uniform regret bound for such an algorithm that only depends on the starting distribution and the policy to which the algorithm is being compared (similar in flavor to the regret bounds above). The authors then define two different starting distributions which lead to two algorithms (GABA-I and GABA-II) with good regret bounds for their setting. The first starting distribution has a higher support size (and larger eventual runtime), but lower overall regret than the second. The second algorithm is based off a clever embedding of the social network graph into a full binary tree that partially captures the connectedness of the original network graph.

Finally, the authors demonstrate how to implement their algorithms efficiently. In particular, even though the first distribution has an exponentially large support size (and thus naively would require exponential time / space to run), they show how to implement it in only O(K log N) time per round via a clever online belief propagation algorithm. Similarly, they show how to implement GABA-II in only O(log K log N) time per round (exponentially faster than GABA-1).



**Limitations And Societal Impact:**

No concerns here. The authors explicitly and adequately address this in their paper.

**Main Review:**


I enjoyed reading this paper -- the high-level setting is very natural (adversarial contextual bandits with graph-based similarity information), the technical content was non-trivial and reasonably novel (at least to me), and the paper was well-written.

The only downside I really see is that it is a little hard for me to parse what the regret bounds mean in settings that might naturally come up -- in particular, what sort of uniform regret bounds do these results imply if we impose constraints on the adversary’s behavior (which is the more common way to think about these types of bandit problems). For example, if the N contexts can be clustered into G groups, and the losses are generated stochastically in such a way that losses of actions in the same group are the same (presumably corresponding to some social network graph composed of G highly-weighted cliques), does the dependence on N decrease to a dependence on G? I think some examples like this would be helpful for conveying intuition for what the results say.

I think the paper could also benefit by citing some of the literature on adversarial contextual bandits (and mentioning that their paper falls into this setting). There has even been some work on adversarial contextual bandits with similarity between contexts: see e.g. “Contextual Bandits with Similarity Information” by Slivkins et al., or “Contextual Bandits with Cross-Learning” by Balseiro et al. I don’t think any of these papers come anywhere close to subsuming these results (the graph-based model in this paper seems quite different), but some discussion would be helpful.

Overall, I think this paper will be of interest to the NeurIPS audience.


**Time Spent Reviewing:**

3

---

> ### Author Response · Authors · 2021-08-11
> **Response to Reviewer Dc7h**
>
> Thank you for your review and comments. Please find our specific responses to your comments below.
>
> > "The only downside I really see is that it is a little hard for me to parse what the regret bounds mean in settings that might naturally come up -- in particular, what sort of uniform regret bounds do these results imply if we impose constraints on the adversary's behaviour (which is the more common way to think about these types of bandit problems). For example, if the $N$ contexts can be clustered into $G$ groups, and the losses are generated stochastically in such a way that losses of actions in the same group are the same (presumably corresponding to some social network graph composed of $G$ highly-weighted cliques), does the dependence on $N$ decrease to a dependence on $G$? I think some examples like this would be helpful for conveying intuition for what the results say."
>
> Thank you for the suggestion on the regret bounds with respect to the graph.  We will include such an example in the next edit.  Here we sketch the workings of the example you suggest .
>
> >"$N$ contexts can be clustered into $G$ groups" as "corresponding to some social network graph composed of $G$ highly-weighted cliques."
>
>  In  summary, we will show that $N$ may be replaced by $G$ up to logarithmic factors.
>
>
> [Single clique].  First consider an unweighted $N$-vertex clique.  Between any two vertices there are $N-2$ edge-disjoint paths of length 2 and 1 edge disjoint path of length 1 thus $r(u,v) \le 1/(1 + ((N - 2)/2) = 2/N$  (using the law of resistors series/parallel and Rayleigh's monotonicity principle), for any pair of vertices.
>
> [A labeled "clique of cliques"]. Next consider that we have $G$, $n$-cliques (thus $N= G n$), and assume that $G = o(n)$, with $n/G$ inter-clique edges between each of the cliques. Thus the total number of inter-clique edges is $(G-1)n/2$.  (Exactly how the inter-edges connect any pair of cliques does not matter).   Observe  if associated with each clique is some action in $[K]$, then in the worst case the cutsize (def. line 129) is the number of inter-clique edges $(G-1)n/2$.  Furthermore, the effective resistance of an inter-clique edge is also $O(1/n)$. This follows since leaving each clique is $\Theta(n)$ edges, and, like we did with the computation of intra-clique effective resistance given any two points on differing cliques, we can find $\Theta(n)$ edge disjoint of $\Theta(1)$ length.   Therefore the resistance-weighted cutsize $\Phi(y)$ (see (4)) is bounded by $(G-1)n/2 \times O(1/n) = O(G)$.  Substituting into (8) we have  $O(\sqrt{K \log(K n) G T})$.
>
> Thus as you ask "does the dependence on $N$ decrease to a dependence on $G$?" we can answer in the affirmative as $G$ has replaced $N$ except for a logarithmic factor.  Finally, note this result is robust in that we could apply "edge noise" to create more "realistic" graphs, as well as increase the number of inter-clique edges by a constant factor, and still the bound will scale smoothly.
>
> ---
>
> > "I think the paper could also benefit by citing some of the literature on adversarial contextual bandits (and mentioning that their paper falls into this setting)  [...]  e.g., *Contextual Bandits with Similarity Information by Slivkins* et. al., or *Contextual Bandits with Cross-Learning* by Balseiro et. al. I don't think any of these papers come anywhere close to subsuming these results (the graph-based model in this paper seems quite different), but some discussion would be helpful."
>
>
> Thank you for making us aware of these references.  We checked them and would like to confirm that these do not subsume the current results.  Furthermore,  Appendix B in the supplementary material contains an expanded literature review, which covers _Contextual Bandits with Similarity Information_ [83] although the Balseiro reference is not discussed.  Should the paper be accepted we hope to re-incorporate some of the expanded literature review to the body of the paper so that we can have a fuller discussion on contextual bandits.

---

### Decision · Program_Chairs · 2021-09-27

**Decision:**

Accept (Poster)

**Comment:**

Some concerns from the reviewers were addressed during the discussion phase.
Despite the tuning issues and the suboptimal bounds brought up in the discussion,
reviewers reached a consensus that the paper studies a well-motivated problem with
some interesting results.
Please do incorporate all the suggestions/discussions from the reviews into the final version.